# Network community structure of substorms using SuperMAG magnetometers

L. Orr [1✉], S. C. Chapman [1], J. W. Gjerloev[2,3] & W. Guo[4,5]

Geomagnetic substorms are a global magnetospheric reconfiguration, during which energy is abruptly transported to the ionosphere. Central to this are the auroral electrojets, large-scale ionospheric currents that are part of a larger three-dimensional system, the substorm current wedge. Many, often conflicting, magnetospheric reconfiguration scenarios have been proposed to describe the substorm current wedge evolution and structure. SuperMAG is a worldwide collaboration providing easy access to ground based magnetometer data. Here we show application of techniques from network science to analyze data from 137 SuperMAG ground-based magnetometers. We calculate a time-varying directed network and perform community detection on the network, identifying locally dense groups of connections. Analysis of 41 substorms exhibit robust structural change from many small, uncorrelated current systems before substorm onset, to a large spatially-extended coherent system, approximately 10 minutes after onset. We interpret this as strong indication that the auroral electrojet system during substorm expansions is inherently a large-scale phenomenon and is not solely due to many meso-scale wedgelets.

[1] Centre for Fusion, Space and Astrophysics, University of Warwick, Coventry, UK. [2] Applied Physics Laboratory—John Hopkins University, Laurel, MD, USA. [3] Birkeland Centre, University of Bergen, Bergen, Norway. [4] School of Aerospace, Cranfield University, Cranfield, UK. [5] Alan Turing Institute, London, UK. ✉email: l.orr1@lancaster.ac.uk

During substorms, Earth's magnetosphere undergoes a global reconfiguration during which stored energy accumulated from solar wind driving is abruptly transported to the ionosphere where it is dissipated[1]. This process generates auroral electrojet currents; electrical currents that flow in the ionosphere. These are the ionospheric segment of a global-scale three-dimensional current system, known as the substorm current wedge (SCW), linking the ionosphere and the magnetosphere across spatial scales of several tens of earth radii. Ground-based magnetometers provide decades worth of time series, with near global-coverage, to address the spatio-temporal coupling of the ionosphere and magnetosphere[2] and have been used in a long list of publications to study the auroral electrojet system, e.g., ref. [3,4]. Since the first schematic picture of the large-scale SCW system[5], there has been a steady stream of competing models[6–11]. In the classic scenario[5], the ionospheric segment of the SCW is illustrated as an intense westward electrojet located in the midnight region with post-midnight feeding and pre-midnight drainage. There have been many variations of this classic scenario[6,9,11] with most focusing on the large-scale structure.

Recently there has been a renewed interest in the meso-scale structure with the scenario of the SCW comprising of individual wedgelets gaining attention[12–20]. In this scenario, the wedgelets are small 3D wedge-like current systems associated with magnetospheric dipolarizing flux bundles[21], which are magnetic structures associated with bursty bulk flows (BBFs)[21,22] hypothesized that the ionospheric segments of a series of individual wedgelets, separated in local time, would appear as a single extensive electrojet from ground magnetometers. The picture by ref. [13] is somewhat different as they suggested the individual BBF's lead to a pressure pileup in the inner magnetosphere, which then result in a large-scale SCW. Merkin et al.[20] demonstrated that magnetic flux accumulation in the inner magnetosphere in the expansion phase of a substorm was dominated by azimuthally localized plasma flows; this process was also accompanied by a plasma pressure build-up. However, they did not directly address the question of the SCW composition by wedgelets. Coupling the magnetospheric process of BBF's to the SCW is appealing as it points to a fundamental property of magnetospheric convection in that BBFs are widely considered as a process that can resolve the pressure balance inconsistency in the magnetotail[23]. Although, simulations[13,20,24–26] provide a powerful means to link ground and space-born magnetometer observations to magnetospheric processes a convincing comparison remains elusive[14].

In ref. [27], we recently made the first application of directed networks to the set of 100+ ground-based magnetometers collated by SuperMAG, which allowed us to test these ideas on a large set of isolated substorm events. To objectively test the hypothesized large-scale SCW models, we used the raw network properties to resolve timings of propagation/expansion of the current wedge within and between three predefined spatial regions and found timings of a consistent sequence in which the classic SCW forms. This required the use of POLAR-VIS data[28] to determine the region boundaries for each event, and necessarily makes the assumption that these regions will unambiguously demarcate the various features of the dynamically evolving current system.

Network science[29–31] provides an extensive tool-set to analyze the properties of networks. Network analysis does not introduce spatial correlation, require any a priori assumptions for variation in ground conductivity or formal categorization of features seen in the spatial field, but allows for quantification of spatio-temporal patterns across sets of multiple, spatially distributed observations. In the context of ionospheric current systems, communities are subsets of the 100+ ground-based magnetometer observations that are more strongly correlated with each other than they are with the rest of the network.

Here we perform community detection on the time-varying networks constructed from all magnetometers collaborating with the SuperMAG initiative. The method can identify whether one or more community (i.e., current system) exists and how this changes in time, and importantly, once the raw network has been constructed, it does not rely on any assumptions or other inputs. We perform this analysis across 41 isolated substorm events at 1-min temporal resolution, focusing on the spatial and temporal characteristics of the SCW and we find a robust and consistent configuration of ionospheric substorm currents evolving as the substorm progresses. Multiple discrete current systems that are present before onset are found to progressively transition into a coherent SCW. Since our methodology quantifies cross-correlation between spatially separated magnetometers, this transition is to a coherent large-scale spatially extended structure, rather than solely a flux accumulation of small-scale wedgelets or flow bursts that are not coherent with each other. This transition occurs over 10–20 min following onset and characterizes the peak expansion phase of the substorm. Our analysis reveals, across many events, an extensive coherent structure, which is consistent with a spatially extended correlated current system. It establishes that a large-scale SCW is an essential part of substorm evolution, but does not exclude the co-existence of smaller scale structures. Our results underline the central role of time-dynamics in models for the substorm current system, since an observational snapshot of the system after onset, but before the SCW has fully formed, would suggest multiple current systems, which may indeed differ from one event to another. This may resolve much of the controversy surrounding models for the substorm ionospheric current system.

## Results

**Community structure of a single substorm.** Figure 1 is an example of how the community structure of the network varies throughout a substorm for a single event. We present this example event as an illustration of the methodology that we will use to compare multiple events. The network is calculated for a substorm on the 16th March 1997, which has excellent magnetometer spatial coverage of the entire nightside, 18–6 h of magnetic local time (MLT), between 60 and 75° magnetic latitude (MLAT) (see "Methods," "Data and event," as well as "Calculating the directed network"). Throughout the substorm there are always ≥7 magnetometers in each 4 h of MLT (i.e., in each of MLT windows 18–22, 22–02, and 02–06), as well as ≥4 magnetometers in each 3 h of MLT (i.e., in each of MLT windows 18–21, 21–00, etc.). In all panels of Fig. 1, the abscissa plots normalized time (see "Methods," "A single normalized time-base for the events" and Eq. (1)) so that the onset of the substorm is at $t' = 0$ (green dashed line) and the time at which the auroral bulge has reached its maximum expansion is $t' = 30$ (red dashed line).

Figure 1a, b visualizes the overall importance in the network, and physical location, of the network communities as a function of time, see "Methods," "Community detection and network parameters." The edge-betweenness algorithm does not pre-define how many communities there should be so the number of communities (plotted as circles) at each time is completely unconstrained and changes throughout the substorm. The network has been constructed by identifying connections using the canonical cross-correlation (CCC) lag at which the CCC between each pair of magnetometers is at its peak so that each connection has an associated lag, $\tau_c$. The magnitude of the lag $|\tau_c|$ indicates whether connections within a given community are

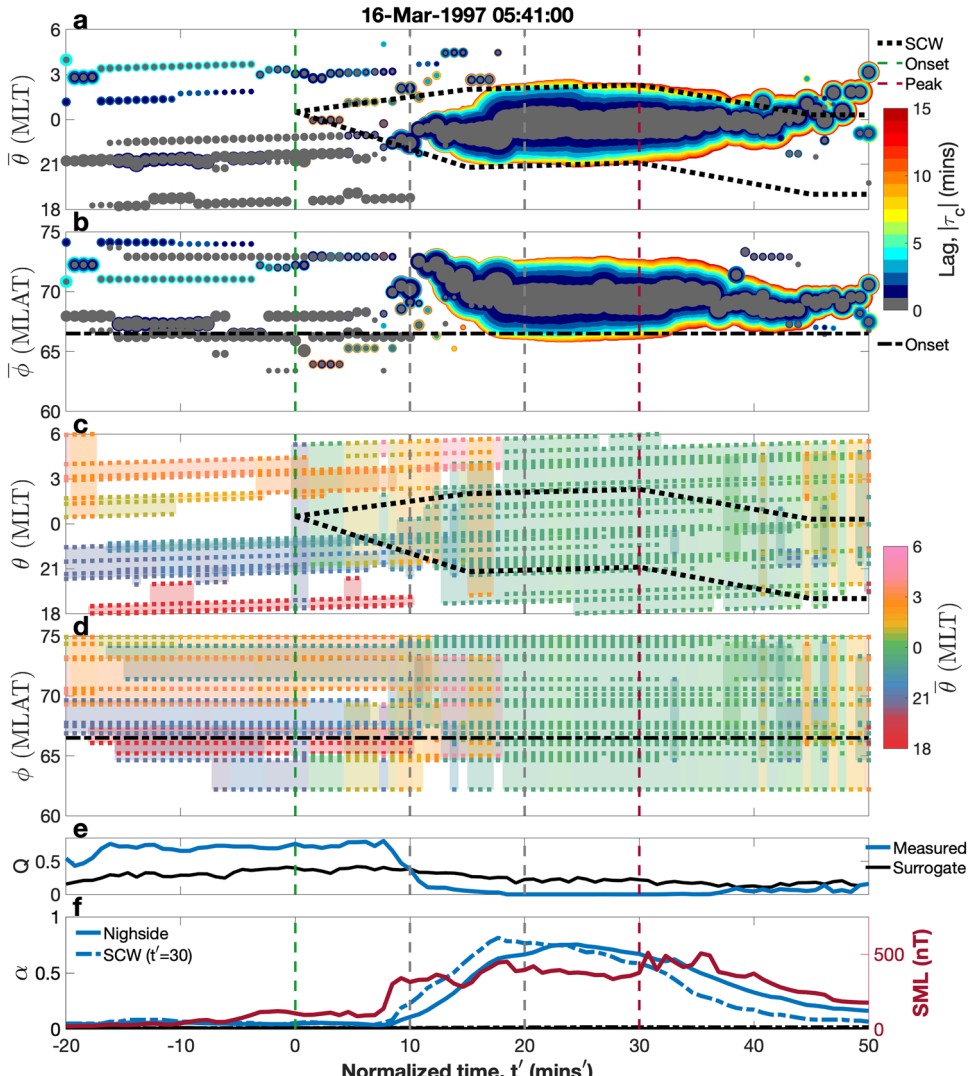

**Fig. 1 The community structure of a substorm on the 16/03/1997.** The abscissa of all panels is normalized time ($t' = 0$ is onset (dashed green line) and $t' = 30$ (dashed purple line) is the time of maximum auroral bulge expansion). Vertical gray dashed lines show ten normalized minute intervals within the expansion phase. **a**, **b** Plots individual communities as circles where the size of the circle reflects the number of connections within the community. The ordinate plots the mean magnetic local time/latitude (MLT/MLAT in h/degrees) of the community, $\bar{\theta}_x(t')$ and $\bar{\phi}_x(t')$, and the color indicates the proportion of connections with each time lag, $|\tau_c|$. The dashed lines overplotted are the edges of the auroral bulge (MLT) and the onset location (MLAT), found from auroral images. **c**, **d** Show the spatial extent of each community, where the dots are the magnetometer locations and the shading is the extent. Color represents the mean MLT of the stations contained within each community, $\bar{\theta}_x(t')$. **e** Plots the modularity, $Q$, (blue line) and the random phase surrogate (black line). **f** Plots the normalized number of connections, $\alpha(t')$, both within the nightside (solid blue) and within the SCW (dashed blue), as well as their surrogates (solid and dashed black, respectively, both near-zero throughout). The right ordinate plots (negative) SML (red).

formed rapidly (zero CCC lag i.e., $\tau_c = 0$ (gray)) or whether they are associated with propagation and/or expansion (non-zero CCC lags from 1 to 15 min (blue-red))[27], see "Methods," "Calculating the direct network."

From before onset until ~10 normalized minutes after there are ~5 small communities. Each community contains few connections (few magnetometers are highly correlated) and are mostly instantaneous (zero lag, there is no delay of the signal between pairs of magnetometers). These communities are spread spatially throughout the nightside at all MLT. These persist beyond onset until $t' \sim 10$ where there is a clear transition in structure. The small communities are replaced by one large community. By $t' \sim 20$, the small communities have been completely replaced by a single large structure containing many highly connected magnetometers (the circle radius is now large representing a large

fraction of the available network). This single community contains about half of connections at <2-min lag and half at longer lags, consistent with expansion and/or propagation of the structure. The MLT, $\bar{\theta}$, of the centroid of this single large community is located well within the auroral bulge. At $t' \sim 40$, this large structure begins to breakdown as the auroral bulge shrinks.

Figure 1c, d plots the physical location of all individual magnetometers that are connected to the network at each time. Each dot is a single magnetometer and each color indicates a distinct community, the color used to label each community corresponds to the MLT of the centroid ($\bar{\theta}(t')$) of that community at each time. There are few, if any, magnetometers in the network below 65° MLAT ($\phi$) throughout the substorm. Before onset we see that the multiple small communities are mostly spatially separated in MLT ($\theta$) and each only involves 3–4 magnetometers.

The communities are slightly more overlapping in MLAT ($\phi$). By $t' \sim 20$, the entire network is one community, spanning the nightside.

Figure 1e plots the modularity, $Q$, a measure of what proportion of the network connections go between communities compared to connections contained within communities (Eq. (2)). If $Q$ is large, the network is densely connected within communities and sparsely connected between them, whereas if $Q$ is low, the system is globally densely connected, that is, dominated by one main community. The modularity for the event (blue line) is compared to that of its random phase surrogate (black line) and we can see that in the event, there is a clear drop in modularity at $t' \sim 10$ from spatially separated localized communities ($Q \sim 0.8$) to one dense global community ($Q \sim 0$) which is clearly distinct from the behavior of the surrogate, which shows little change throughout the event.

Figure 1f shows the (negative) SML time series, an index of substorm electrojet enhancement, alongside the normalized number of connections, $\alpha(t')$ (Eq. (3)) contained both within the nightside (solid blue) and within the auroral bulge boundaries at the time of maximum expansion, $t' = 30$. Both SML and $\alpha(t')$ begin to increase just before $t' = 10$, just as the network begins to transition from small scale to large scale and drop in modularity, and maximizes at $t' \sim 20$, when the entire network is a single highly connected global structure. The random phase surrogate remains $\sim 0$ throughout the substorm.

The detailed evolution of the spatio-temporal current system from onset to peak expansion is found to vary between events. We show several examples in the supplementary information (see Supplementary Figs. 2, 5, 8, 11 and Supplementary Note 1) and the modularity before onset varies across these five examples (e.g., $\sim 0.4$ in Supplementary Fig. 2, Supplementary Note 1 and $\sim 0.8$ in Fig. 1). However, we consistently find a transition from many distinct communities to a single coherent community, which spans the entire nightside at peak expansion. In contrast, Supplementary Figs. 29–32, Supplementary Note 6, show the community structure of two flow burst events[32,33], neither of which transition into a large-scale community with spatial extent across the nightside.

We can see this transition in the physical maps of substorm individual events. Figure 2 contains eight snapshots of the same substorm, corresponding to the times in Fig. 1 (see also Supplementary Figs. 3, 6, 9, 12 and Supplementary Note 1). The snapshots are SuperMAG polar plots of the nightside overplotted with images of Polar VIS[28]. The polar plots are overlayed with the magnetometer locations and their associated magnetic field perturbation vectors ($B_N$ and $B_E$). The magnetometers, which are not part of any community, at each time, are colored black whilst the magnetometers contained in a community are colored by the the MLT of the centroid ($\bar{\theta}(t')$) of each community. The colors match those in Fig. 1c, d. Each community is also shaded and surrounded with a dashed line. The full network is shown in Supplementary Fig. 1 and Supplementary Note 1. We see from Fig. 2a that just before onset there are five communities, all of which are small and spatially separate, with the largest community only containing six magnetometers. These communities only begin to incorporate larger numbers of magnetometers at about $t' = 10$ (Fig. 2d). At this time there is much more activity around the auroral bulge (brightening of polar VIS), ten magnetometers with dense connectivity form a community centered at $\sim 22.5$ MLT (blue) and a second community formed of nine magnetometers spans from $\sim 21.5$ to $\sim 5.5$ MLT (orange). Magnetometers that are not included in these communities may still be included in the network (as is the case here, see Supplementary Fig. 1, Supplementary Note 1, which plots all network connections).

At this time the modularity can be seen in Fig. 1 to be dropping significantly. At $t' = 15$ (Fig. 2e), all magnetometers that are within the network are within these two communities. There are still three magnetometers, which are not connected to the network. At the peak of the substorm expansion, $t' = 20$ (Fig. 2f), all available magnetometers are connected to the network and are now contained in one single community, suggesting the entire global system is highly connected. This configuration is unchanged across the maps at $t' = 25$ and $t' = 30$ (Fig. 2g, h), consistent with Fig. 1.

The overall dynamics of this event is then a coalescing of multiple, small communities into two, and then one single global community. We have analyzed other events (see Supplementary Figs. 2–13 and Supplementary Note 1 for a further four examples) and find that there is always a transition from initially many small, spatially separated communities at onset to globally correlated system at the peak of the expansion phase. However, there is a great deal of variety in how these small communities coalesce depending on the substorm. For examples see Supplementary Figs. 3, 6, 9, 12 and Supplementary Note 1 where all begin as several communities, sometimes overlapping in MLT, and tend toward a spatially extended system.

**Community structure of multiple substorms**. We can see how robust this transition is by comparing the time evolution of the modularity (see "Methods," "Community detection and network parameters") of the networks of the set of 41 substorms, which have been selected using the above criteria to be quiet before onset and to have good magnetometer spatial coverage of the nightside, each has at least two magnetometers in each 3-h MLT sector (see Supplementary Figs. 26, 27, Note 4 for plots of the magnetometer coverage and "Methods," "Data and event"). Figure 3 plots the normalized modularity, $Q_N$ for these 41 events. We use normalized modularity, $Q_N$, as a parameter for community structure, where $Q_N \rightarrow 1$ when the network has multiple, separate sub-networks of magnetometers with many connections within but few between and $Q_N \rightarrow 0$ when the network is globally dense (either a single community or several large-scale communities with many connections between them), see "Methods," "Community detection and network parameters." There is a value of the modularity for each substorm at each minute in time and panel a plots as a function of time the modularity probability (count of substorms with $Q_N$/total number of substorms), that is, the fraction of the substorms, which have normalized modularity within each $Q_N = 0.05$ bin, indicated by color. We have not included data where there was less than five connections in the network, i.e., we imposed a $m \geq 5$ criteria on the modularity data to avoid $Q = 0$ simply because there are no connections. The criteria make little difference to Fig. 3 but when the network has few connections (i.e., Supplementary Fig. 24 and Supplementary Note 2 where the correlation threshold is much higher) it has a noticeable effect.

There is a clear pattern of high modularity before onset, which drops to low modularity from $t' \sim 20$. This is also shown in Fig. 3b, which plots an overlay of the modularity time series for each substorm. The median modularity and the 25th and 75th quantiles are overplotted on this panel. This again shows a clear transition from high to low modularity that takes place in the (normalized) time between onset and expansion peak. Finally, Fig. 3c plots histograms of the probabilities of modularity from all 41 events aggregated across 10–11 normalized minute time windows. There is a clear transition from a right-shifted distribution, with median $\sim 0.7$ before onset, to an approximately uniform distribution from $0 \leq t' \leq 9$, to a highly peaked left-shifted distribution from $t' \sim 10$, with $\geq 50\%$ of substorms having $Q_N \leq 0.1$. By the

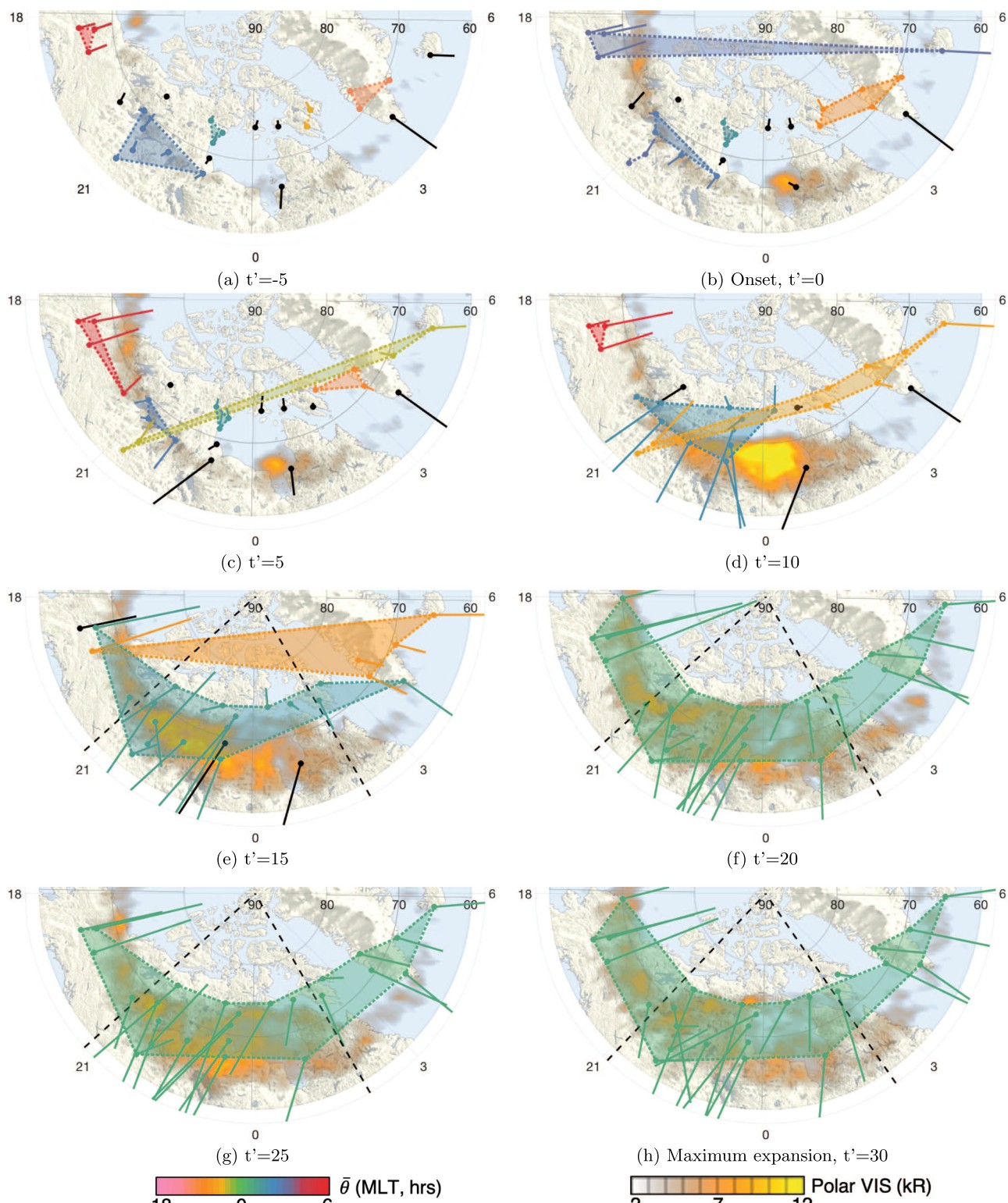

**Fig. 2 Community structure snapshots of an example substorm in normalized time, _t′_.** Polar plots are in magnetic coordinates centered at the magnetic pole, where magnetic local time (MLT, h) increases clockwise, with midnight located at the bottom (MLT = 0 h). Maps show the nightside from dusk (MLT = 18 h) to dawn (MLT = 6 h) and 60–90° magnetic latitude. Plotted are the magnetic field perturbation vectors (North and East components, $B_{N,E}$, measured in nT) for a substorm on 16/03/1997. The colorbars at the bottom of the figure represent the MLT of the centroid ($\bar{\theta}(t')$) of each community, and polar VIS data from left to right, respectively. The vectors are color-coded using the left and match those of **c**, **d** in Fig. 1. Each subplot (**a**–**h**) represents a snapshot of the community structure in intervals of five normalized minutes from before onset (**a**, $t' = -5$) to the time of maximum expansion (**h**, $t' = 30$), corresponding to the times in Fig. 1. The circles represent ground magnetometers with the line representing the $B_{N,E}$ vector. Black magnetometers are not part of a community. The dashed lines are the locations of the auroral bulge found from auroral images. The vectors are overplotted on maps provided by superMAG[2] containing polar VIS data[28] in kR, matching the right colorbar.

**5**

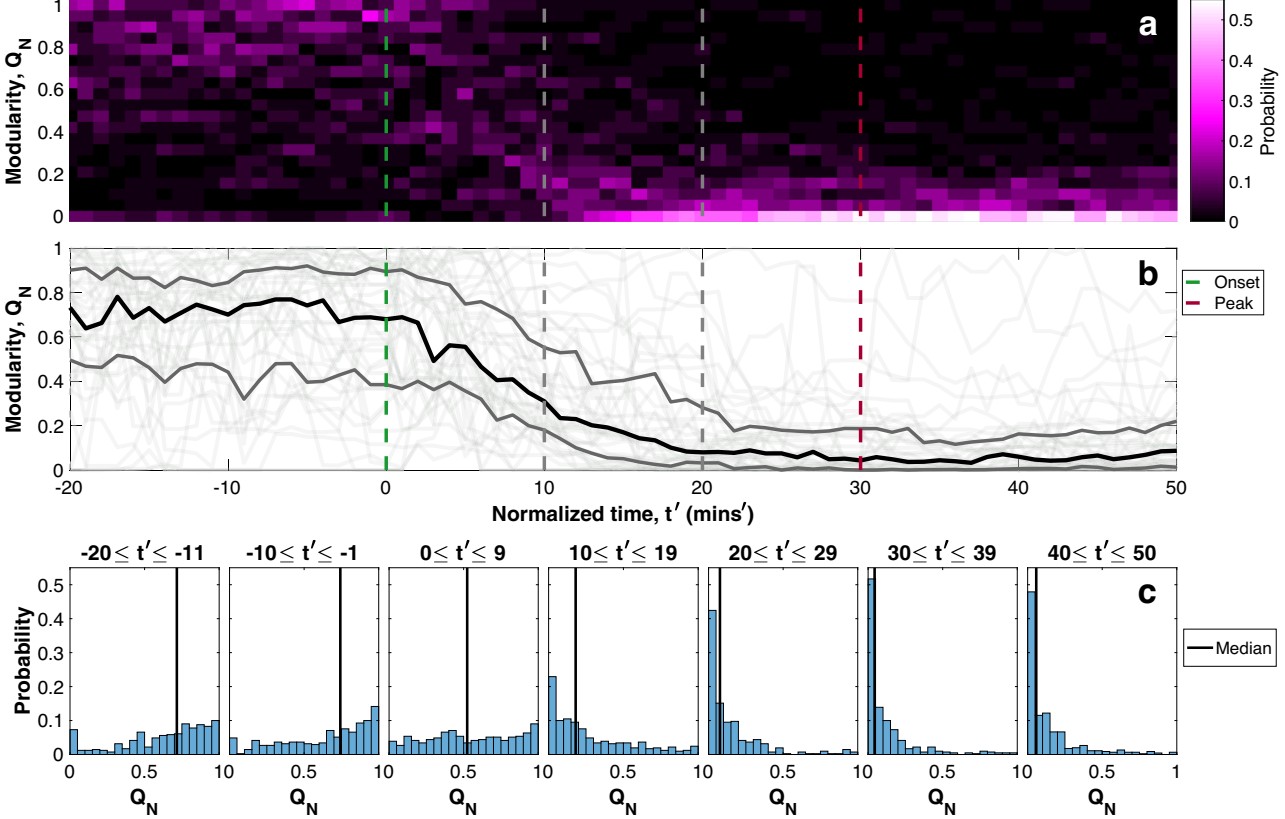

**Fig. 3 Community structure of multiple substorm events.** The normalized modularity, $Q_N$, of the set of 41 substorms that have two or more magnetometers in four even local time sectors of the nightside and are quiet before onset. **a, b** Share normalized time as the abscissa. **a** Ordinate bins $Q_N$ at each normalized time and the color indicates the probability (count of substorms with $Q_N$/total number of substorms). **b** Plots $Q_N$ of each of the 41 substorms as a function of normalized time, $t'$, as thin light gray lines. The median is overplotted in black and the 25 and 75% quantiles in darker gray. **c** Plots the normalized histograms of $Q_N$ of the events aggregated over 10-min intervals as time progresses. The median is overplotted.

time of maximal auroral bulge expansion less than a quarter of the substorms have modularity over 0.2. This suggests that whilst before the substorm, and up to 10 min following onset, there can be a variety of community structures, which give a broad spread in modularity values, once the expansion phase is reached, almost all the events are at low modularity, consistent with a large-scale spatially coherent community.

We find the same overall behavior looking over a wider set of events (see Supplementary Figs. 19, 20 and Supplementary Note 2), however more active conditions before the substorm make the pattern less clear. If we only consider substorms with large values of SML in the 127-min preceding onset, we can no longer isolate such a clear transition in the modularity distribution (see Supplementary Fig. 22 and Supplementary Note 2). Supplementary Figures 15–19 and Supplementary Note 2 show the same figure but using five alternative algorithms for community detection. The same pattern is repeated in all cases. Further, the coherent behavior is repeated across a range of cross-correlation thresholds (see Supplementary Figs. 23, 24 and Supplementary Note 2). If the threshold is set too high (<1% of magnetometers connected within a month) there would not be enough network connections to see any pattern (hence Supplementary Fig. 23, Note 2 contains more outliers). Supplementary Figure 25 and Supplementary Note 3 shows that the modularity, $Q$, derived from the substorm events has more structure (higher $Q$) than random networks and for a given degree distribution, the data explores a broad range of modularity values, $Q$, and vice versa. We have included that these Supplementary Figs. to show our results are not simply an artifact of choice of algorithm or threshold.

## Discussion

We have used well-established network science techniques to analyze data from >100 ground-based magnetometers operated by contributors to the SuperMAG collaboration[2], see "Methods," "Data and event." We translated this data into a time-varying directed network, based on CCC of the vector magnetic field perturbations measured at each magnetometer pair[34,35] and performed community detection on the network, see "Methods," "Calculating the directed network," and "Community detection and network parameters." Communities are locally dense but globally sparse groups of connections in the network[36], identifying emerging coherent patterns in the current system as the substorm evolves. We consistently find robust structural change from many small, uncorrelated groups of magnetometers before substorm onset, to a large spatially extended correlated system during the expansion phase.

The spatial structure and time evolution of the magnetopsheric SCW is fundamental to our understanding of how earth's magnetosphere releases and dissipates energy accumulated by solar wind driving. The various proposed models for the SCW imply different and conflicting magnetospheric reconfiguration scenarios[5,6,9,16,37]. We have shown that the SCW consistently, over 40+ events, displays large-scale correlated behavior and this is inconsistent with the recent hypothesis that this current system consists solely of multiple distinct meso-scale wedgelets[12,13,15–20].

As our technique is based on correlated magnetometer data, we cannot resolve structures that are on spatial scales smaller than that of the inter-magnetometer spacing, nor on short temporal scales of <1 min, thus although we are able to resolve flow bursts

(see Supplementary Figs. 29–32 and Supplementary Note 6 for examples) we cannot resolve short-timescale, small-scale events such as individual BBFs or wedgelets if they are below this resolution. However, unless the wedgelets are spatially and temporally correlated with each other, we would not expect to see the spatially coherent signature of global cross-correlation seen across the auroral bulge. We cannot rule out the scenario in which there are two or more individually correlated spatial structures that result in perturbations on the ground that are spatially overlapping. A two component system where region A is correlated with region B, and region B is correlated with region C, cannot be distinguished from a single component system where A is directly correlated with C (as both are correlated with B). Nonetheless, the spatially extended communities we observe cannot be obtained by having many, small, spatially localized wedgelets, which are each internally correlated, but not cross-correlated with each other. All 40+ substorms analyzed here ultimately form a large-scale coherent current wedge structure. The structural shift from multiple to a spatially extended current system occurs ~10 min after onset. It excludes models in which the current system is solely comprised of individual, uncorrelated wedgelets and indicates that the auroral electrojet system is inherently a large-scale phenomenon.

The structural transition from multiple to a large-scale correlated system is not instantaneous, it occurs approximately over the 10 min after onset. We have found examples where this transition is a direct coalescence of multiple small communities into a single global community. We also found examples where multiple small communities first coalesce into two or more large communities, which then transition into the single global community at expansion peak. This emphasises the need for an understanding of the dynamical evolution of substorms, which may resolve the controversy surrounding models for the substorm current system. If we were to only look at a snapshot of the nightside's magnetic activity within the first 10 min of a substorm we may indeed see multiple structures but as the substorm evolves we clearly see an underlying spatially extended coherent system. Our network analysis is built on linear cross correlation and as such does not identify non-linear relationships. Communities are a relatively non-formal way (despite unbiased estimates) of quantifying interactions. This could be addressed in future work using more advanced methods, provided there is sufficient observations to make this viable; an essential limitation here is that the substorm timescale is a few hours and the data are at 1-min time resolution.

This work introduces a parameter for the spatio-temporal pattern captured by the full set of ground-based magnetometers— the network modularity. From ~20 min into the substorm expansion phase over 75% of the substorms analyzed here have low normalized modularity (<0.2), indicating a highly correlated, large-scale global system. The modularity provide a quantitative spatio-temporal response benchmark for MHD simulations and SCW models.

## Methods

**Data and event section**. The SuperMAG database[2] collates the full set of magnetometer observations at 1-min time resolution with a common baseline removal technique and standardized coordinate system. We study a subset of events drawn from the list of isolated substorms between 1997 and 2001 established by ref. [37] and also described in ref. [11]. They determined the substorm timing (onset and peak expansion) solely from global auroral images[37] rather than magnetometer data or magnetic indices. Earth camera data were supplemented with additional data (visible imaging cameras and ultraviolet imager[38]) to eliminate pseudo-onsets. The substorms were included if they (i) were optically and magnetically isolated events, (ii) had a spatially defined onset location, (iii) were bulge-type auroral events, (iv) had a single expansion and recovery phase (or the event ended at the time of a new expansion), (v) had the entire auroral bulge region in darkness, and (vi) had $|Dst| < 30$ nT (not during magnetic storms) or prolonged magnetic

activity. Excluding daylit stations avoids large differences in ground conductivity between the stations, which would otherwise dominate the analysis. The analysis is fully explained below, after the data are introduced.

Magnetometers within the nightside, from 18 to 6 h of MLT, between 60 and 75° MLAT, are chosen to best observe the auroral electrojets[37,39]. To maintain good magnetometer coverage of the nightside during the substorm, we require two or more magnetometers in each 3-h window of the nightside (e.g., ≥2 magnetometers in each segment of 18–21, 21–24, 00–03, and 03–06 h of local time). A subset of 75 substorms fulfill this criteria. We also require that the nightside is quiet for at least one CCC window (127 min) before the substorm onset so that the network calculated at the time of substorm onset is not contaminated with previous activity, rejecting events where the SML index[40] exceeded ~25% of its maximum value (at the peak of the substorm) in the 127 min before the start of the substorm. Forty-one substorms fulfill all specified criteria and are analysed in this paper. A list of the 41 substorms, with their onset and peak expansion times, is given in Supplementary Table 1 and Supplementary Note 8. Substorms with more activity before onset, from the list of 75 events, are analysed in the supplementary information (Supplementary Figs. 20–22 and Supplementary Note 2) for comparison. We will present as an example a substorm with excellent magnetometer spatial coverage alongside an epoch analysis of the normalized modularity of the 41 substorms. Figures plotting the spatial distribution and geodetic magnetometer separation, for each of these 41 substorms, are given in the Supplementary information (Supplementary Figs. 26, 27 and Supplementary Note 4). Supplementary Figure 26 shows that the vast majority of the chosen substorms exceed this minimum criteria for spatial distribution and have good local time coverage. We find that the inclusion of substorms with lower coverage does not alter our result and their inclusion avoids bias that could arise by hand picking events. Further substorm examples (Supplementary Figs. 2–13 and Supplementary Note 1) and an epoch analyses of other substorms (Supplementary Figs. 20–22 and Supplementary Note 2) are included, alongside a list of their onset and peak expansion times, in Supplementary Tables 2, 3 and Supplementary Note 8.

**A single normalized time-base for the events**. To compare multiple substorms, we first map each event onto a common normalized time-base such that, once normalized, all substorms share a common onset time and take 30 normalized minutes to develop from onset to the maximum expansion. The method for time normalization, $t'$, developed in ref. [37] is as follows:

$$t' = \frac{T_E \times (t - t_{onset})}{t_{peak} - t_{onset}} \tag{1}$$

where $T_E = 30$ min, approximately the mean length of substorm expansion. Onset is defined at $t' = 0$ and the time of peak expansion at $t' = 30$. The critical timings for this normalization, onset and peak, can be explicitly identified in these isolated substorm events. The network and all associated parameters are calculated at the resolution of the magnetometer data (1 min) before being rescaled onto the normalized time-base.

**Calculating the directed network**. Networks have been utilized as a useful mathematical analysis tool in the social sciences over a number of years[29–31] and have more recently found application in geophysical data[41–43]. A diagram and description of a network are included in the glossary, Supplementary Fig. 33 and Supplementary Note 7. The first step in any network analysis is to calculate the raw time-varying network, here from the full set of ground-based magnetometer observations. A number of approaches to infer links exist, such as cross-correlation[44] or causal inference between temporal signals on nodes[45]. The details of the underpinning methodology for forming the raw network are detailed in refs. [27,34] and we summarize it here. The magnetometers are the nodes of the network and a pair of nodes are connected (have an edge between them) if the CCC[46] of their vector magnetic field perturbation time series exceeds a station and event-specific threshold. A station and event dependent threshold is calculated from the data for each event to determine the threshold CCC for each station to be connected into the network for 5% of the month surrounding the event[34]. This ensures that each station has the same likelihood of being connected to the network, normalizing for their individual sensitivities to overhead current perturbations and for variations in ground conductivity. The time-varying network is calculated at minute resolution, using a sliding leading edge 128-minute window for the CCC (i.e., the time, $t$, will refer to the last/latest time point on the window). The network calculated at the time of substorm onset will then involve observations over 127 min of pre-substorm activity. By focusing on isolated substorms for this study, where preceding conditions are quiet, we can compare substorm network properties with a quiet-time baseline for each event. For each windowed time series, we linearly detrend to remove any slow trends such as seasonal trends or modes associated with the enhanced DP-2 current. Supplementary Figure 28 and Supplementary Note 5 models the typical DP-1 and DP-2 responses and shows our detrending distinguishes between the two currents. In order to give sufficient accuracy in the computed cross-correlation function, whilst still capturing the large-scale spatio-temporal current system behavior, a 128-min window is chosen. Dods et al.[35] previously demonstrated that changes can be resolved on finer timescales than that of the window length, specifically capturing a step change in

activity, such as when the SCW forms. Supplementary Figure 28 and Supplementary Note 5 demonstrates likewise, using a 128-min CCC window. In ref. [34], the technique was applied using zero-lag CCC and trialled to obtain the undirected network for a small set of isolated substorm events. This was sufficient to capture the initial spatially coherent response at onset.

Importantly, to recover the full dynamics that occurs on multiple timescales, the CCC at all time lags is needed. Orr et al.[27] pioneered this approach, obtaining the first directed networks for the response to substorms seen in the full SuperMAG data set. Including time-lagged correlations means that a magnetometer pair, which were not connected to the network (i.e., correlated above the threshold) at zero lag may be connected at non-zero time lag. The directed network is formed by using the peak value of the CCC (considered for lags $-15 \leq \tau_c \leq 15$ min) between two magnetometers to determine whether there is a network connection between them. A station pair is connected in the network if the peak CCC value at time lag $\tau_c$ exceeds the connection threshold for the station pair. A non-zero lag, $\tau_c$, indicates temporal information flow, so that the connection now has an associated direction (indicated by the sign of $\tau_c$) and timescale (magnitude of $\tau_c$) of propagation/expansion of the observed signal between the two magnetometers. A schematic showing how the network is constructed by identifying connections between pairs of magnetometers is included in the Supplementary Fig. 34 and Supplementary Note 7. We use this method to calculate the raw time-varying directed networks for this study.

**Community detection and network parameters.** Community detection is a method of locating groups of nodes within the network, which are locally dense with connections but which have sparse inter-group connections. Communities characterize the meso-scale topology of the network[36] and provide insight on its formation and functionality[47]. A diagram is provided in the Supplementary Fig. 35 and Supplementary Note 7, as an example of a network with three communities. We apply community detection algorithms to the raw directed time-varying networks obtained from SuperMAG observations of substorm events to characterize the network structure in terms of communities. There are many community detection algorithms that identify optimal dense subgraphs in directed or undirected graphs. We used a variety of community detection algorithms in the igraph package in R[48] to determine community structure.

Results using the edge-betweenness algorithm[36] are highlighted here but we have verified that our results are robust against the choice of algorithm; results using different algorithms, including the "optimal"[49], walk trap[50], information mapping[51], leading eigenvector[50], and label propagation[52] algorithms are reported in the supplementary information (Supplementary Figs. 15–19 and Supplementary Note 2). The ability to validate community structure is limited[53] but we aim to preform a statistical study without focusing on individual details within individual substorms, hence our conclusions should be robust against detection algorithm. The edge-betweenness algorithm from the igraph package[48] considers the direction of the edge when dividing into communities. All other algorithms tested treat the network as undirected. The edge-betweenness community detection algorithm may be summarized as follows: consider a network comprised of two communities (A and B), each containing highly connected nodes, with only one connection (a bridge) between the two communities. If we consider all possible shortest paths needed to travel between the nodes in A and the nodes in B, the edge that connects the communities will always be crossed, therefore the bridge edge will have the highest edge-betweenness. The same logic corresponds to multiple communities; the edges between the communities will carry the majority of shortest paths and therefore have the highest edge betweenness. The edge-betweenness algorithm[36] identifies and successively removes these edges, which have the highest edge betweenness, leaving behind sub-networks that are the individual communities. Like Stochastic Block Models, it can be argued that this is an unbiased algorithm, which does not predetermine how many communities the network should divided into nor prescribe any other parameter, enabling multi-scale community structures to be recovered by varying the number of edges removed.

Modularity is a measure that has been widely used to evaluate how well the community structure has been captured[50,54]. To measure the robustness of the communities formed, the network modularity measures how separate the nodes within the different communities are[36,55]. The modularity treats all edges as undirected and is expressed as follows. We first divide the network into q communities, which defines a $q \times q$ symmetric matrix $\mathbf{e}$ whose elements $e_{xy}$ are the fraction of all edges in the network that link nodes in community x to nodes in community y. Then $e_{xx}$ is the fraction of the network contained within community x and $e_{xy}$ is the fraction of the network that connects between communities x and y. The modularity elements are provided as an example in the network from Supplementary Fig. 35 and Supplementary Note 7. The fraction of network edges that connect nodes in community x to the rest of the network is $f_x = \sum_{y=1,q} e_{xy}$. The modularity, Q, is then given by:

$$Q = \sum_{x=1,q} \left( e_{xx} - f_x^2 \right) \tag{2}$$

The modularity parameterizes to what extent the network is characterized by many, separate communities or one dominant community. If the magnetometer signals become more strongly correlated to each other, forming a single community, the modularity will tend to zero as $Q \to e_{xx} - e_{xx}^2 \to 0$ as $0 < e_{xx} < 1$. For a network comprised of smaller, interconnected communities, Q will be finite. In order to compare many events, we then normalize the modularity such that in

the quiet interval before each individual substorm, the maximum value of $Q = 1$. For the networks formed from the SuperMAG set of ground-based magnetometers, the network communities identify the spatially coherent perturbations from ionospheric current systems. The change in the modularity as the substorm progresses identifies how these current systems are changing. The modularity will be maximum when spatially localized coherent perturbations cause a group/community of magnetometers to be internally correlated, but not cross-correlated with other groups/communities of magnetometers. If all auroral latitude magnetometers are highly correlated due to a large-scale ionospheric current system overhead, the modularity will be near-zero. A schematic of the current system models we would expect from high or low modularity is contained in the Supplementary Fig. 36 and Supplementary Note 7.

The overall network response is parameterized by the normalized number of connections, $\alpha(t)$ (see also refs. [27,34,35])

$$\alpha(t) = \frac{\sum_{i \neq j}^{N(t)} \sum_{j \neq i}^{N(t)} A_{ij}}{N(t)(N(t) - 1)} \tag{3}$$

where $\mathbf{A}$ is the adjacency matrix ($A_{ij} = 1$ if magnetometers i and j are connected and $A_{ij} = 0$ otherwise) and $N(t)$ is the number of active magnetometers.

We have performed two tests of the significance of our results. We checked the statistical significance of the modularity of a given substorm by constructing a random phase surrogate for that event. The random phase surrogate provides an estimate of the network properties that could arise "by chance" in a given set of data. For each event, the time series from each magnetometer are Fourier-transformed, the phase spectrum is randomized whilst preserving the amplitude spectrum and this is then inverse Fourier-transformed to give a surrogate time series with the same power spectrum as the original signals, but with no time correlation (for an example see ref. [56]). We used an iterated amplitude-adjusted Fourier transform[57] method with the Matlab code supplied by ref. [58]. The surrogate time series of each magnetometer is then used to calculate the random phase surrogate of the network parameters using the method as described in "Methods" subsections "Calculating the directed network" and "Community detection and network parameters". For each event, we repeated this ten times to obtain an average value of the random phase surrogate modularity. The random phase surrogate is indicated on plots of our results. We also compared the modularity with that of random networks generated with the same total number of connections and degree distributions as the networks derived from the substorms, which varies with time. This is plotted in Supplementary Fig. 25, Note 3, which shows that there is a clear upper bound to the value of modularity per degree distribution observed in random networks. However, the modularity patterns observed throughout this text exceed those observed in a random network and therefore are not simply a result of the increasing numbers of connections.

## Data availability
The SuperMAG ground magnetometer station data and Polar Vis Earth Camera images analysed during the current study are available in the SuperMAG repository, http://supermag.jhuapl.edu/. Substorm timings used for analyse during this study are included in this published article and its supplementary information files. Source data are provided with this paper.

## Code availability
All code[59] used in this study is available from https://github.com/laurenorr/substorm-community/tree/nat_comm/.

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

## Acknowledgements

We acknowledge use of the SuperMAG ground magnetometer station data and Polar Vis Earth Camera images. S.C.C. and L.O. acknowledge AFOSR Grant FA9550-17-1-0054 and ST/P000320/1. S.C.C. and J.W.G. acknowledge ISSI team 455.

## Author contributions

L.O. performed the bulk of the analysis with assistance from S.C.C. and W.G., and created the figures. J.W.G. provided Polar VIS timings and the list of substorm events. All authors contributed to the manuscript and approved the final draft.

## Competing interests

The authors declare no competing interests.
