## [Peer Review File · Nature Communications]

Reviewers' comments:

Reviewer #1 (Remarks to the Author):

This paper uses 'community detection', which has not been used in space physics, to study the evolution of magnetospheric substorms as measured by ground magnetometers in the form of the substorm current wedge (SCW). I believe pulling in new techniques from other disciplines is worthwhile, and from all indications the concept of 'communities' has the potential for great application in multipoint space physics measurements. So there is a lot to like about this paper, and the authors did a nice job of explaining a complicated topic. Below I'm going to list a number of criticisms with the paper; but there are some very nice aspects to the paper.

My primary criticism, which I detail below, is that I don't believe the conclusion that substorms do not proceed solely by small-scale wedgelets is supported by the analysis. In fact, if I understand the analysis correctly – and I may not be! – I believe the study is incapable of answering the question as posed. In addition to the comments below, I think it would be very helpful to run this technique on one of the published events that definitely sees a wedgelet current system. E.g. Grocott et al. [2004]; there are several. Demonstrate that the technique can identify a wedgelet system. Without that confirmation, it is difficult to determine what a negative result really means.

The paper describes a wedgelet/SCW scenario in which the ionospheric segment of individual wedgelets appear as an extensive electrojet (lines 52-57). But I don't believe this is the commonly accepted viewpoint of how BBFs/flows create the SCW. The many simulation papers of Birn, and the recent one by Merkin [2019] (all referenced), argue that BBFS enter the SCW generation region, and contribute to an overall larger region of enhanced pressure, and very quickly the mesoscale structure of the BBF has disappeared, consumed within the larger pressure gradient in the inner magnetosphere. This overall enhanced pressure is what drives the SCW, not the summation of distinct individual wedgelets that remain. I think this is the widely accepted scenario. The picture of Liu et al. [2018] I think is what the authors are taking issue with (I would take issue with it as well), as this is expanded upon in lines 422-425. But I don't believe it correct to lump the 2 pictures together. So I believe the paper needs to do a better job of cleanly motivating the study, in that the authors are testing (I think) the idea that wedgelets remain distinct within the overall SCW picture. Which, I agree with the authors here, is probably incorrect.

That said, within the merging/pressure gradient scenario, the current study cannot address the contribution of wedgelets. But it's also unclear to me if the current study would be able to pick out individual wedgelets of the Liu et al. scenario, as I have a number of questions about the methodology below.

Questions/concerns about the methodology:

1) “good” magnetometer coverage is defined as 2 or more stations in each 3 hour window of the nightside (lines 226-229). I think this would actually lead to poor coverage. For example, 2 stations could be at the duskward edge of the pre-midnight window, 2 could be at the dawnward edge of the postmidnight window, and the separation between the pairs could be almost 6 hours of LT. That’s a lot. In practice, how often does this happen? I don’t know, but this definition does not constrain coverage very well, and quick flip through the examples shows many cases where this is true.

2) There are a couple of aspects of the study that I think emphasize picking out large scale features while minimizing the small scale features the study wants to identify. First, the time normalization is a good approach for the superposed aspect of the study, but introduces a bias against short duration structures. For example, the 16-Mar-1997 example in the paper had onset at 05:41 UT, and going by the SML index peak expansion was about 06:30 UT. That’s a 50 minute substorm, which was contracted into a 30 minute time-series. A wedgelet current system that may last only a few minutes in ‘real’ time, is contracted further, making it difficult to pull out in a 128 minute cross-correlation. Relatedly, in a 128 minute cross correlation, I have a hard time believing that it would key off of a temporally short and importantly small relative amplitude signal of a wedgelet. More likely is it’ll key off of the temporally and dynamically larger integrated SCW system. I’d like to see an example of what a cross-correlation looks like. Even in the Liu et al. scenario, how long could one reasonably expect a wedgelet system to remain intact, and would it be picked out by this technique? The authors need to make that case.

3) The spatial coverage is key to making this technique work for the intended purpose. Figure 2b was really nicely done, but highlights the issue. Onset in auroral imagery is just post midnight, where there is 1 station. So the technique will not pick out a community until the coherence expands to the next station, which is ~2 hours of LT duskward, which is larger than the scale size of wedgelet systems. In other words, unless you had a dense network of magnetometers, this current study would miss the wedgelet. And the criterion of 2 stations in 3 MLT hours I think is insufficient (see e.g., any number of papers that have ‘imaged’ the current system of BBFs/wedgelets using inversion techniques). Further, the study limits stations to mlat > 65°, which explicitly limits measurements to the auroral zone – and often to well poleward of the auroral zone. The reasoning for this was not mentioned, but presumably this was because mid latitude stations would not observe local wedgelet systems. But auroral zone magnetometers during substorms are dominated by local variations, so it’s not surprising to not get a ‘community’ until later in the substorm.

4) Figure 2 also highlights that I think the technique is pulling out the WTS/auroral bulge, at least initially (Figure 2c). I think this explains why the Q factor drops 1/3 – 2/3 of the way into a substorm expansion. You don’t get largescale coherence until the WTS forms. It also explains why, in Figure 1, you don’t see much indication that the substorm has occurred until T=10.

Finally, at several points the paper notes that the substorm starts off initially as non coherent/distinct communities that eventually merge into a single coherent community at substorm

peak expansion. I think that is in agreement with the widespread understanding of how BBFs/flows accumulate to create and sustain the SCW, as I first described above. However, because of the issues listed above, I'm not sure the study would have reached any other conclusion. That is, because of the sparse ground network and the use of 128 minute windows, I think the study could only come to the conclusion that the SCW is not spatially coherent until expansion is well underway.

Also, if the authors are really arguing specifically against a specific model of distinct wedgelets that remain intact, that should be made explicit in the beginning. And the authors should then detail what they would expect to see if this was really happening. I think the authors are arguing that because the SCW eventually becomes coherent – 2/3 of the way into the substorm – the model cannot be correct. But let's say that the model is correct. Would the model show a set of communities within the auroral bulge, each representing a distinct wedgelet? I think that's what the authors are arguing, but it isn't very clear. But, again, based on the issues above, I don't believe there are sufficient magnetometer points to see this, even if it was occurring.

Reviewer #2 (Remarks to the Author):

SUMMARY

This paper examines an important question. What is the system of ionospheric currents associated with the substorm current wedge? Is it possible that this system is the superposition of multiple wedgelets caused by multiple bursty bulk flows in the tail? The paper utilizes the magnetic variations recorded by more than 100 auroral zone magnetometers during isolated substorms to examine this topic. The authors apply a new approach which they describe as a “time-varying directed network”. Each magnetometer is considered as a possible node in the network. If the correlation of this magnetometer with another exceeds a “connection threshold” the pair is said to be “connected”. A time delay in the correlation indicates the direction of propagation of the signal. If there is a region with locally dense connections that do not extend beyond the region the collection of nodes is called a “community”. The paper utilizes new concepts including: canonical correlation analysis, edge-betweenness, modularity, adjacency matrix, normalized connection number, and normalized time base.

The authors provide a list of 76 isolated substorms from which they use 41 to define and characterize network development. They conclude that prior to the formation of the current wedge the auroral zone currents form isolated communities with no large scale coherent structure. However, once the current wedge begins to grow the isolated communities combine to a single coherent structure indicating that it is unlikely that the ionospheric currents are produced by multiple unrelated events, i.e. the superposition of wedgelets idea is rejected.

EVALUATION

I do not find fault with the mathematics or statistical procedures used although I was unable to follow their description of the analysis procedure for lack of detailed explanation of several concepts. I also agree with their conclusions as I have also criticized the multiple and simultaneous wedgelet explanation of the current wedge. Instead I have advocated for a coherent source that is localized in space but produced by the accumulation of multiple flow bursts. My main objections to this paper involve the absence of a clear physical description of how the magnetic perturbations used in the analysis are produced by processes in the magnetotail, and how their method would detect them. I am also concerned with details of the substorm selection and onset timing. I am unable to determine what effects these details have on their analysis or conclusion. Finally I do not feel the authors make the case that their approach disproves the wedgelet idea. My recommendation is significant revisions are needed.

HISTORICAL NOTES

In the early days of the space age a very small number of magnetic observatories was present in the auroral zone and Polar Regions. These magnetometers recorded photographically and it was difficult to determine the nature of the current systems causing the magnetic variations. Various studies resulted in the recognition of two distinct current systems in the ionosphere during auroral activity. These were the DP-2 (disturbance polar of second type) and DP-1 (*Nishida*, 1968a; *Nishida*, 1968b; *Nishida*, 1971). DP-2 is a two cell system with foci near the terminators in the auroral zone, and DP-1 had a single cell centered near midnight. The two systems occur

simultaneously but are uncorrelated. Some authors suggested that the currents were confined to the ionosphere (*Akasofu et al.*, 1965) while others suggested that the currents were three dimensional originating in space (*Bonnevier et al.*, 1970; *Bostrom*, 1964; *Fukushima*, 1970). Definitive proof that the DP-1 current system originated in space was provided by data from the ATS-1 satellite (*McPherron*, 1972). This 3-D current system was later studied using midlatitude magnetometer data by (*Clauer and McPherron*, 1974; *Horning et al.*, 1974). The work by Clauer established that isolated substorms produce a coherent pattern of midlatitude variations in both the north and east magnetic field components that grows systematically with time. Subsequently it was established that DP-2 is the ionospheric manifestation of the 3-D Region 1- Regio-2 current system (*Iijima and Potemra*, 1976; *Iijima and Potemra*, 1978) driven by magnetospheric convection and DP-1 is driven by the current wedge (*McPherron et al.*, 1973).

GENERAL COMMENTS

My first question is if you take account of past work why would you expect to find correlations between stations averaged over two hours (128 min) and the entire nightside auroral zone to reveal the presence of a wedgelet perturbation localized in local time (<1 hour), magnetic latitude, and in time (< 15 min)? My second question is why use auroral zone data to study the current wedge? The current wedge is most clearly seen in midlatitude data where the systematic growth of specific patterns in X and Y components is seen over the entire nightside. My third concern is the absence of any discussion of the distinction between DP-2 and DP-1. DP-2 is present throughout the substorm driven by the R-1 & R-2 currents. It is globally coherent as shown by previous studies such as AMIE (assimilative modeling of ionospheric dynamics). DP-1 (the substorm current wedge) is an independent current added to the pre-existing currents. The analysis does not include a procedure for separation of these two systems so it is difficult to see how a network of correlations proves anything about how the current wedge is formed. See for example the use of principal component analysis to carry out this task (*Sun et al.*, 1998).

A major concern on which I can comment is related to the list of substorm onsets provided in the Supporting Information. This list has 38 rows and two columns totaling 76 events. These were presumably timed using a combination of SML index records and Polar Satellite images. The authors say they used 41 events for their analysis but do not specify which events. When I examined the first event used in Figure 1 of this paper (05:41 on March 16, 1997) I found I did not agree with the timing of the event. Figure r1 of this review shows why. The stated onset is 05:41 UT but the sharp break in SML index occurs at 05:50 and in the MPB index (midlatitude positive bay) at 05:52. The MPB index is specifically designed to reveal the onset and development of the current wedge (*Chu et al.*, 2015; *McPherron et al.*, 2018; *McPherron and Chu*, 2017; *McPherron and Chu*, 2018). These times are respectively 9 and 11 minutes later than the time in their list. This seems significant as they discuss the initial 10 minute delay after their onset as a time of multiple small communities. The later times correspond to the start of a single coherent community so the idea suggested that many localized current systems merge to form a single system seems suspect.

Because of the above I decided to convert their list into a time ordered sequence and created similar plots for all 76 events. In doing this I interactively selected the start and stop times for the pulses in SML and MPB. These are evident in Figure r1 as a colored circles on SML and diamonds on MPB. The selection process also illustrates a second problem in the author's

analysis. The end of the expansion (peak of the current wedge) are often ambiguous. I used the highest point in MPB, not the end of its flat top (this is not typical), and the lowest point in SML just before recovery begins, but an earlier time might be appropriate. Note the two times are quite different giving an MPB duration of 19 min and a negative bay duration of 39 minutes. This is also significant as the SML duration was used to scale events to a common length as explained in Section 2.4. In fact, since it is the current wedge that is being studied it seems more appropriate to use timing from MPB index than the SML index. The authors do not describe their timing procedure or list the times determined in the scaling of their time series.

In addition to plotting and timing I also performed a superposed epoch analysis of SML and MPB using the author's list of 76 events. The results are shown in Figure r2. It is apparent that their requirement that the substorms are isolated is not satisfied for all events. It may be the case that they discarded the bad events, but this point is not discussed in the manuscript and there is no indication of which events were actually used. An examination of the 76 plots reveals a number of other problems. Some negative bays do not have a corresponding positive bay and vice versa. Some events have multiple positive bays in sequence while some have multiple dips in SML. In some cases the SML index simply decreases gradually without a sharp onset.

The times determined by interactive selection of sharp onsets were then used to examine the time delays between events in the three lists. Figure r3 contains three panels showing probability distributions for these three lists. Panel (a) presents the difference between the author's SML list (Orr) and the interactive SML selection list. The peak is at 1-minute and the average is at 6 minutes. The distribution has a standard deviation of 10.2 min. Positive delay means the interactive onset is later than the author's onset. Actually, in my experience, this is quite good agreement between two independent lists of substorm onsets. Panel (b) shows the distribution of MPB onset delays relative to the author's list. It is broader ($\sigma = 16.7$); has a more positive average of 13.5 minutes; and is peaked at 4 minutes. The fact that the average delay is more than 10 minutes suggests that their topic of study, the substorm current wedge, does not start at the time assumed in their analysis. Panel (c) shows the delay between the interactive MPB and SML onsets. The distribution is more tightly constrained ($\sigma = 13.3$), its average is lower (7.5) and its peak closer to zero delay (2.0), than is the case using the author's list. These facts suggest that there are small errors in their timing of SML onsets, and that typically perturbations at midlatitudes are delayed relative to those at high latitudes. If this is true it poses an interesting scientific question, why is the auroral zone closure of the current wedge seen before the onset of midlatitude effects? One possible explanation is that a few auroral zone stations record the initial closure before the wedge is wide enough to create significant midlatitude perturbations.

The significance of including bad events and timing errors is illustrated by additional analysis. First we standardized the time series for every event by subtracting event mean and dividing by event standard deviation. This forces all events to have approximately equal size. Then we performed superposed epoch analysis to obtain the average pulse shapes. The results are shown in Figure r4. In panel (a) the three MPB pulses correspond to ensembles created from the three lists shown in the legend. The blue curve represents the author's list. It has the lowest peak amplitude, broadest peak, and longest delay to peak. The average with respect to the interactive SML onsets is shown by the red curve. It is slightly higher, narrower, and has shorter delay than the author's list indicating better phasing of the events in the average. The black curve is the

result for the average of MPB using MPB onsets. As expected it is the highest, narrowest, and least delay to peak.

Panel (b) presents similar results for the SML index. The pulse height and widths are very similar, but the time to minimum is ordered in the same way as in MPB results. Note that the red curve for the interactive list has a very sharp onset consistent with the procedure for selecting the onset. The blue curve for the author's list is not quite as sharp. The black curve for averaging relative to the MPB onsets is very rounded and not so sharp because the time delay between MPB onset and SML onset is so variable.

The consequence of inclusion of bad events with improper timing is illustrated in Figure r5. To produce this figure we correlated the average pulses for SML determined with the authors' onset list. Each event in the ensemble of 76 events was correlated with the average pulse determining the maximum correlation and its lag relative to the onset time used. This lag provides a correction that could be added to the interactive onset to improve the correspondence between the shapes of the event and the model pulse. This will smooth the sharp onset but gives an onset dependent on the complete waveform. However, it is not clear that this is the proper thing to do. We then sorted the original list into descending order of correlation. The 38 MPB wave forms shown in panel (a) have the highest correlating ranging from ??? t ???., while those in panel (b) have the lowest correlation (/// to ????). Panels (c) and (d) show similar results for the SML index. It is interesting and perhaps significant that as soon as we included 41 events in the plot we began to observe distortion produced by activity prior to or following the current wedge interval.

In panel (a) is evident that there is a spread in onset times and time of peak activity when compared to the average curve. In panel (b) the average MPB pulse (a) nearly disappears. The problem seems to be serious timing errors as well as multiple wedge signatures. Similar statements can be made about the SML curves in panels (c) and (d). In the next figure we illustrate how these results change when the MPB onset list and average pulse is used for the MPB correlation and sorting, and the same is done for SML.

The effect of using the interactive onset lists and average pulses determined from these ensembles is demonstrated in Figure r6. Panels (a) and (b) are MPB events superposed by the interactive MPB onset list and correlated with the pulse determined with this list. Panels (c) and (d) are the same except SNL is organized by the interactive SML list. The superposed results in panel (a) are much cleaner than obtained using the authors' list. The individual MPB pulses are more similar to the average, the peak of the average is higher, the pulse is narrower, all pulses start at the same relative time, and the time of peaks line up better. There is even an improved signature for the 38 events with the worst correlation plotted in panel (b). The behavior of the SML index is very similar.

CONCLUSIONS

The analysis I have described above clearly shows that there are errors in event timing, and that the choice of events may not have been optimum. Because they did not specify which events were used I can't tell if they used bad events. I did find that the agreement between their list of SML onsets and mine was relatively good but with my events averaging 6 minutes later.

However, I suspect that there may be more disagreement about the times of the end of the expansion as their procedure for this determination was not described and their times were not provided in the supporting information. This is serious as the difference between the two determines their “normalization” of the time series. Another problem is the fact that my determination of the onset of the midlatitude signature of the current wedge has an average delay of 13.5 min (7.5 min if I use my list of SML onsets). This makes me wonder if there is real variability in this delay. It could happen if it takes different lengths of time for the current wedge to open enough to create a midlatitude signature.

SPECIFIC COMMENTS

Abstract

The paper appears to be directed to the magnetospheric research community as it begins with a discussion of the substorm current wedge and the question of the structure of the ionospheric currents at the end of the current wedge. However, the bulk of the presentation emphasizes the technique of network structure and barely discusses the geophysical issues. I am not convinced that the technique used is capable of answering the geophysical question posed in the abstract. I note that the abstract reads more like an Introduction complete with citations to the geophysical literature.

Section 1: At the end of this section it is claimed that there are multiple discrete current systems present prior to expansion onset, i.e. during the substorm growth phase. I would expect that this would be the time when the global DP-2 system is present. Studies of this system show that this is globally coherent. That this does not show up in the analysis suggests that its magnetic perturbations are swamped by localized noise which is correlated only over a few stations. If there are localized communities of connections at this time then one should look at the local magnetic perturbations to determine if they have an obvious physical cause. After the current wedge forms I would expect its signal to grow to the point it overcomes the local noise and a single coherent system develops. What is the evidence that these merge rather than just disappear?

Section 2: This section introduces a number of mathematical/statistical concepts that are unlikely to be familiar to most space physicists. More description of these are needed. These include:

- Network
- Node
- Canonical correlation
- Direction of propagation
- Community detection
- Edge betweenness
- Modularity
- Normalized number of connections
- Adjacency matrix
- Time series normalization

Line 84: Which substorms? The list provided contains two columns with 38 rows or a total of 76 events. These events are not in any obvious order. Also the time for the end of each expansion is not included.

Line 113: The procedure for canonical correlation is not described. A search of the space physics literature shows only a few papers that describe this technique. I suspect that this procedure obtains two pseudo time series, one for each station, that is some linear combination of the three field components that have maximum correlation between stations. A fuller explanation is needed here if anyone is going to reproduce this work.

Line 126: The typical duration of a substorm is 150-180 minutes. The duration of a current wedge is less than 70 minutes. It appears that the correlation function cannot change much once the full pulse is inside the moving window of two-hour length. I can imagine that the correlation would not change much for at least one hour as the window passes over the pulse related to current wedge. Please explain how a 2-hour window is able to detect short-term temporal variations. In my experience the correlation functions tends to be dominated by the largest disturbance present in the window so I don't see how it could detect a small and short term perturbation at two stations if a global system is also present.

Line 132: The concept of "directed" is defined here as delays due to propagation. In the study of current patterns I would think that the sign of the correlation coefficient is also important. In the current wedge the east component (Y or D) is positive pre-midnight and negative post-midnight. It is not clear that this would be true if components are being combined in a canonical correlation. It is true that this requires knowledge of the actual pattern of the magnetic perturbations to prove that the correlations have the correct sign.

Line 164: I have no idea of what the "edge-betweenness" algorithm does. References to the statistical literature are not sufficient for the typical space physics reader. Please give some simple summary of what is done in this process.

Section 2.3: The description of event selection is inadequate as I discuss at length above. I am concerned that timing errors in the selection of expansion and recovery onset times has introduced significant errors in the analysis. Also it cannot be determined if all the substorms used are truly isolated.

Line 274: In Figure 1 there is a proliferation of small communities with few connections until 10 minutes into expansion. This is likely noise as they vanish as the wedge develops. There is a more important issue here. I showed above that the timing used in the study is likely erroneous. The stated onset is 05:41, but my scaling of the SML index shows it is actually 05:50 in the magnetometer data, and the MPB index shows the current wedge did not start until 05:52. Thus the first 10 minutes do not appear to be related to the actual wedge formation.

Line 319: There are only 4 dashed lines at 10 min intervals in Figure 1. Figure 2 uses 5-in intervals.

Figure 1: Can you explain why you used theta for magnetic local time and phi for magnetic latitude? This is the opposite of the usual definitions of spherical coordinates.

RECOMMENDATION

I believe this manuscript requires major revisions to address the issues I have raised above. Frankly I don't believe that better timing will change any of the results in a significant way. I am not convinced that the technique used really establishes the non-existence of wedgelet signatures. I am not even convinced that they have studied the current wedge (DP-1) as distinct from DP-2 (global convection). They co-exist during a substorm and are both global and coherent. The paper must first establish that they are studying DP-1. Second they must show what their technique can respond to wedgelet signatures. I am not sure what these are in the SuperMag data. My guess is that wedgelets cause a small isolated negative bay at one or two stations. Typically I would expect these to occur at high latitudes and in only one station. The signature should move equatorward. I can't see how these could appear in the network analysis when two other global current systems are present.

SUPPORTING INFORMATION FOR REVIEW

In addition to the following figures I have prepared two text files. The first file "reftimes.txt" contains five columns containing date and time in the format 'yyyymmddHHMM '. Column 1 is the Orr list sorted chronologically. The next two columns are my lists of SuprMAG SML index times, expansion onset and recovery onset for each event in the Orr list. The last two columns are the same for the MPB index.

```
199701070303 199701070306 199701070331 199701070325 199701070335
199701071729 199701071727 199701071740 199701071730 199701071743
```

The second file "sortidx.txt" are the indices that will sort each of the lists into order of decreasing correlation with the ensemble average pulse. Col 1 sorts Orr list (Col 1 in above file), Col 2 sorts Col 2 & 3) of previous file (SML), and Col 3 sorts MPB (col 4 and 5 of previous file.

```
52 34 45
34 52 10
```

Matlab the code is:

```
% Create sorted ensembles for MPB using Interactive MPB onset list
modMIM = nanmean(ensMIM); %Create a model MPB pulse using Orr list
[ZensMIM, mxixMIM, imMIM] = standccens(ensMIM,modMIM,30);
smxixMIM = mxixMIM(imMIM,:);
sZensMIM = ZensMIM(imMIM,:);
nmodMIM = nanmean(sZensMIM(1:38,:)); %model MPB pulse with sorted Orr list
```

The unction "`= standccens`" does the correlation and sorting. It returns the columns such as "imMIM" (indices im for MPB sorted by Interactive MPB nsets).

REFERENCES

- Akasofu, S.I., S. Chapman, and C.-I. Meng (1965), The polar electrojet, *Journal of Atmospheric and Terrestrial Physics*, 27(11/12), 1275-1305, doi:10.1016/0021-9169(65)90087-5.
- Bonnevier, B., R. Bostrom, and G. Rostoker (1970), A 3-Dimensional Model Current System For Polar Magnetic Substorms, *Journal of Geophysical Research*, 75(1), 107-&, doi:10.1029/JA075i001p00107.
- Bostrom, R. (1964), A model of the auroral electrojets, *J. Geophys. Res*, 69(23), 4983-4999.
- Chu, X., R.L. McPherron, T.-S. Hsu, and V. Angelopoulos (2015), Solar cycle dependence of substorm occurrence and duration: Implications for onset, *Journal of Geophysical Research-Space Physics*, 120(4), 2808-2818, doi:10.1002/2015ja021104.
- Clauer, C.R., and R.L. McPherron (1974), Mapping the Local Time-Universal Time Development of Magnetospheric Substorms Using Mid-Latitude Magnetic Observations, *Journal of Geophysical Research*, 79(19), 2811-2820, doi:10.1029/JA079i019p02811.
- Fukushima, N., (1970), Electric current-system for polar substorms and its magnetic effect below and above the ionosphere, in *Proceedings of Upper Atmosphere currents and electric fields symposium. Boulder, CO*.
- Horning, B., R.L. McPherron, and D. Jackson (1974), Application of Linear Inverse Theory to a Line Current Model of Substorm Current Systems, *J. Geophys. Res.*, 79(34), 5202-5210, doi:10.1029/JA079i034p05202.
- Iijima, T., and T. Potemra (1976), The Amplitude Distribution of Field-Aligned Currents at Northern High Latitudes Observed by Triad, *Journal of Geophysical Research*, 81(13), 2165-2174, doi:10.1029/JA081i013p02165.
- Iijima, T., and T.A. Potemra (1978), Large-Scale Characteristics Of Field-Aligned Currents Associated with Substorms, *Journal of Geophysical Research-Space Physics*, 83(NA2), 599-615, doi:10.1029/JA083iA02p00599.
- McPherron, R.L. (1972), Substorm related changes in the geomagnetic tail: the growth phase, *Planetary and Space Science*, 20(9).
- McPherron, R.L., B.J. Anderson, and X.N. Chu (2018), Relation of Field-Aligned Currents Measured by the Network of Iridium (R) Spacecraft to Solar Wind and Substorms, *Geophysical Research Letters*, 45(5), 2151-2158, doi:10.1002/2017gl076741.
- McPherron, R.L., and X. Chu (2017), The Mid-Latitude Positive Bay and the MPB Index of Substorm Activity, *Space Science Reviews*, 206(1-4), 91-122, doi:10.1007/s11214-016-0316-6.

- McPherron, R.L., and X. Chu (2018), The Midlatitude Positive Bay Index and the Statistics of Substorm Occurrence, *Journal of Geophysical Research: Space Physics*, 123(4), 2831-2850, doi:doi:10.1002/2017JA024766.
- McPherron, R.L., C.T. Russell, and M. Aubry (1973), Satellite studies of magnetospheric substorms on August 15, 1968, 9. Phenomenological model for substorms, *J. Geophys. Res.*, 78(16), 3131-3149.
- Nishida, A. (1968a), Coherence Of Geomagnetic Dp 2 Fluctuations With Interplanetary Magnetic Variations, *Journal of Geophysical Research*, 73(17), 5549-&, doi:10.1029/JA073i017p05549.
- Nishida, A. (1968b), Geomagnetic DP-2 Fluctuations And Associated Magnetospheric Phenomena, *Journal of Geophysical Research*, 73(5), 1795-&, doi:10.1029/JA073i005p01795.
- Nishida, A. (1971), DP 2 and polar substorm, *Planetary and Space Science*, 19(2), 205-221.
- Sun, W., W.Y. Xu, and S.I. Akasofu (1998), Mathematical separation of directly driven and unloading components in the ionospheric equivalent currents during substorms, *Journal of Geophysical Research-Space Physics*, 103(A6), 11695-11700, doi:10.1029/97ja03458.

FIGURES

Figure r1: A plot showing solar wind coupling function (a), MPB (midlatitude positive bay) index (b), and SML index (c) for an event in Figure 1 of the manuscript. The vertical dashed line indicates the expansion onset quoted in text. The colored circles show the expansion onsets and ends as determined interactively. Both the SML and MPB onsets are delayed relative to that of the authors. The end of the expansion is often ambiguous.

Figure r2. Superposed epoch analysis of SML (a) and MPB index (b) using authors' (Orr) list of SML onsets. A number of the events display activity prior to the expansion onset (see elevated values prior to dashed line). The MPB index suggests that some events have multiple current wedge signatures.

Figure r3. Panels (a) and (b) show distributions of time delays between the authors' onset list and the interactive determination of SML and MPB. Panel (c) contains the delay between interactive MPB and SML onsets. In all distributions positive delay indicating interactive onsets are later than the authors' onsets. For both SML onset lists the MPB onset follows the SML onset.

Figure r4. Ensemble average pulses in the MPB and SML indices caused by substorm onset. Note the dependence of amplitude, width, and time delay on which list is used for analysis. The differences are caused by small errors in the onset times in each list. These average pulses were used in correlation analysis to detect the best examples of correlated signatures of the current wedge.

Figure r5. The average pulses were correlated with all events and then the event lists were sorted according to decreasing quality of the correlation. The 38 events with best correlation are shown in left column, panels (a) and (c), and the 38 worst in the right column (b) and (d). Panel (a) shows the consequences of the variable relation of the start of the MPB pulse and the authors' list. Panel (c) reveals a slight spread in SML onset and peak times. The right column displays problems caused by lack of isolation of the substorms from other activity.

Figure r6. The top panels (a) and (b) shows the difference in ensemble of events when interactive MPB onset is used to correlate and sort MPB events. The bottom panels show similar results for SML. Even the events with lower correlation seem to be better organized.

199701070303	199701070306	199701070331	199701070325	199701070335
199701071729	199701071727	199701071740	199701071730	199701071743
199701220505	199701220506	199701220527	199701220508	199701220525
199701271014	199701271014	199701271033	199701271015	199701271037
199701312021	199701312019	199701312035	199701312022	199701312038
199702021606	199702021607	199702021637	199702021619	199702021636
199702240820	199702240833	199702240909	199702240852	199702240913
199702271417	199702271417	199702271454	199702271419	199702271449
199703020329	199703020328	199703020410	199703020343	199703020411
199703031621	199703031618	199703031711	199703031622	199703031639
199703160541	199703160549	199703160627	199703160552	199703160609
199703241030	199703241037	199703241056	199703241047	199703241114
199709060457	199709060501	199709060524	199709060501	199709060518
199709300815	199709300817	199709300832	199709300818	199709300833
199711060401	199711060352	199711060417	199711060406	199711060418
199712162129	199712162134	199712162156	199712162136	199712162205
199801021339	199801021339	199801021407	199801021416	199801021442
199801041237	199801041251	199801041307	199801041253	199801041317
199801060249	199801060251	199801060259	199801060249	199801060305
199801101557	199801101633	199801101636	199801101615	199801101644
199801110705	199801110706	199801110751	199801110716	199801110758
199801200342	199801200348	199801200407	199801200354	199801200411
199801291602	199801291600	199801291624	199801291602	199801291612
199802061641	199802061655	199802061703	199802061654	199802061754
199802211953	199802211955	199802212032	199802211955	199802212008
199803171057	199803171059	199803171105	199803171101	199803171113
199809150640	199809150640	199809150707	199809150640	199809150656
199811021848	199811021911	199811021944	199811021916	199811021934
199811160228	199811160249	199811160305	199811160253	199811160315
199811162135	199811162144	199811162233	199811162144	199811162236
199811260810	199811260823	199811260850	199811260832	199811260901
199811261616	199811261625	199811261645	199811261629	199811261641
199812030711	199812030704	199812030722	199812030710	199812030710
199812051142	199812051143	199812051159	199812051146	199812051207
199812051351	199812051357	199812051434	199812051359	199812051411
199812071137	199812071138	199812071229	199812071154	199812071208
199812071612	199812071610	199812071614	199812071613	199812071626
199812122031	199812122027	199812122120	199812122037	199812122107
199812161032	199812161040	199812161106	199812161045	199812161106
199812200406	199812200448	199812200517	199812200449	199812200517
199901010452	199901010455	199901010501	199901010452	199901010509
199901082003	199901082037	199901082042	199901082036	199901082054
199901100846	199901100847	199901100910	199901100854	199901100920
199901181301	199901181314	199901181319	199901181307	199901181336
199902032155	199902032204	199902032220	199902032200	199902032217
199902150151	199902150152	199902150211	199902150200	199902150217
199903260856	199903260915	199903260939	199903260925	199903260946
199910211555	199910211603	199910211623	199910211559	199910211613
199910291906	199910291911	199910291947	199910291938	199910291958
199911020013	199911020006	199911020028	199911020010	199911020046
199911061120	199911061124	199911061216	199911061200	199911061221
199911070145	199911070144	199911070158	199911070155	199911070210
199912250517	199912250558	199912250621	199912250602	199912250619
199912291541	199912291607	199912291704	199912291650	199912291709
200001030308	200001030308	200001030346	200001030311	200001030331

200001031934	200001031935	200001031959	200001031923	200001031952
200001071605	200001071605	200001071625	200001071611	200001071619
200001081031	200001081029	200001081043	200001081039	200001081058
200001092132	200001092132	200001092148	200001092141	200001092203
200001251739	200001251744	200001251756	200001251746	200001251804
200002091620	200002091628	200002091651	200002091636	200002091708
200002161205	200002161215	200002161253	200002161319	200002161349
200002161205	200002161216	200002161253	200002161319	200002161349
200002191013	200002191012	200002191027	200002191013	200002191028
200002200646	200002200645	200002200655	200002200646	200002200658
200002251153	200002251154	200002251229	200002251157	200002251211
200002260601	200002260602	200002260656	200002260606	200002260634
200002290727	200002290725	200002290750	200002290726	200002290750
200003060445	200003060450	200003060511	200003060506	200003060520
200003170936	200003170932	200003171011	200003170956	200003171027
200012181606	200012181612	200012181638	200012181615	200012181636
200012290339	200012290340	200012290408	200012290341	200012290405
200101031244	200101031245	200101031254	200101031246	200101031255
200109060627	200109060628	200109060659	200109060630	200109060701
200110210715	200110210714	200110210729	200110210719	200110210737
200110310730	200110310730	200110310741	200110310733	200110310746

52	34	45
34	52	10
70	13	59
13	31	38
38	33	11
31	12	4
67	38	16
12	28	56
74	11	37
21	65	67
7	74	26
17	67	2
33	18	43
50	70	72
71	16	32
11	69	36
69	50	3
28	59	19
20	19	29
18	71	12
58	20	13
8	17	33
51	4	47
16	58	8
30	7	62
65	51	63
9	5	65
19	21	18
5	39	66
59	26	7
60	72	34
6	30	6
4	8	31
72	43	76
26	9	44
10	76	9
43	60	42
36	22	22
22	6	64
66	32	70
1	10	41
39	36	51
76	75	74
75	66	25
40	25	46
32	44	5
25	49	28
49	1	20
24	57	53
57	53	71
44	24	21
53	47	30
2	2	15
47	40	23
62	56	14

63	54	50
37	37	58
15	46	54
54	68	60
68	3	24
56	63	17
46	62	39
3	15	75
45	45	40
29	64	69
64	27	68
27	29	27
48	42	55
14	48	49
41	14	61
23	41	1
42	23	52
35	35	48
73	73	57
61	61	73
55	55	35

Reviewer #3 (Remarks to the Author):

The paper presents an interesting piece of research whereby isolated substorms were analysed collectively using network science. Magnetic properties of the substorms were used to construct networks and community detection was applied and benchmarked with null models. Results demonstrated that the previously unconnected substorms merged into a large scale SCW.

1) Results presented here are very interesting. With regards to the network analysis, two areas would affect the overall results.

Firstly, it is dependent on how the networks were constructed. The networks were constructed using readings from over 100 ground-based magnetometers whereby the magnetometers (physical locations) are nodes and a link exists between two locations when the cross-correlation of their vector magnetic field exceeds a pre-defined value - i.e. when two locations exhibit similar magnetic patterns. Given that the networks were conjectured based on pre-defined assumptions, it would be also be useful to say something about what is actually forming the communities. While the results show that a large-scale structure has emerged in the "constructed" network, what does this actually mean in terms of the behaviour of substorms. Magnetic activities from different locations become more similar/correlated/synchronous?

The authors have assigned directions to links in the construction of the networks (page 3 line 65 and In 401). The original edge-betweenness algorithm (Newman & Girvan, 2004) ignores the direction and treats networks as undirected graphs. Is this also the case here? If so, this needs to be made clear. If the direction is included in the actual analysis, then please clarify why and how this has been taken into account when the algorithm was applied.

Secondly, for the null comparison, the randomisation is done on the phase spectrum. Could you provide more information why this was adopted? A common way to perform null comparison is to preserve certain property of the network, e.g. degree sequence, and generate a large number of ensemble networks (e.g. 100 or 1000). Also, the empirical networks are benchmarked against 10 randomly generated networks. It is not clear why 10 networks would be sufficient.

In addition, I have a number of suggestions on the manuscript and hope they would help improve the overall readability.

3) It would be useful to emphasise more on the contribution of the research in various parts of the manuscript. In the abstract, Ln 15-17, “imply different and conflicting scenarios” - it would be useful to be more precise about the research question/problem being address here and therefore help highlight the novelty of the work. There are also points in various parts of the manuscript that need to be linked to make the argument here more coherent.

Ln 47 - suggests large-scale structure has its weaknesses and then in Ln 433 recommends a model for correlated large-scale SCW is needed. It seems contradictory?

Are the issues being substorms have only been studied as isolated object? ... i.e. in Ln 20 “a series of mesoscale wedgelets”? or been only focusing on the large scale structure (Ln 47). does that mean that this study is the FIRST to examine SCWs and their time varying properties collectively?

If this is the case, I don't think it has been made 100% clear in the manuscript. Also, if this is the case, the use of network science is a mean to achieve this goal, suggest to emphasise more on the objective.

3) In the methods section, suggest you move 2.3 and 2.4 to the beginning and introduce your data before describing the methods

4) Ln 255 - 271 - this paragraph is repetitive of the caption, suggest the authors to make this more concise or use this space to help readers to interpret panel 1 and panel 2 of figure 1. At present, these subfigures (as well as the rest of the subfigures) contain a lot of information, it would be help to provide further description and explanation to help the readers.

In Panel 6 (fig 1), it is not so easy to see the results from the surrogate - presumably there should be two lines close to zero there.

NCOMMS-20-02005-T 'Network Community Structure of Substorms using SuperMAG Magnetometers' by Orr et al

Response to Reviewers

We are grateful to the Reviewers for their careful reading of our manuscript and respond to all of their comments point-by-point below. As this is an extensive report we first offer a summary of the main concerns. Both Reviewers #1 and #2 raise various concerns about the spatial and temporal resolution of our methodology, specifically, whether or not it could resolve the signatures of smaller scale wedgelets or flow bursts. We have edited the end of our abstract read:

"All 40+ substorms analysed exhibit robust structural change from many small, uncorrelated current systems before substorm onset, to a large spatially-extended coherent system, ~10 minutes after onset. We interpret this as strong indication that the auroral electrojet system during substorm expansions is inherently a large-scale phenomenon and is not solely due to many meso-scale wedgelets."

This is our key result. It relies on quantitatively establishing that in all of these events there is the formation of an extended spatial region encompassing many ground-based magnetometers, all of which are spatially and temporally correlated. It does not rely on being able to distinguish small scale, uncorrelated structures, i.e wedgelets. Wedgelets may indeed be present and may indeed precede, and may even be involved in, the formation of the large-scale electrojet system.

We have emphasised this point throughout this response letter and made many changes to the manuscript in an attempt to clarify. That being said we have followed the advice of Reviewers #1 and #2 and ran the technique on several published flow burst events. We now demonstrate that our method can indeed detect flow bursts. We now include analysis of two events: that mentioned by reviewer #1 from Grocott et al. [2004]; and one other from Gallardo-Lacourt et al. [2014]. The events are plotted below. These events are included in the SI and text has been added to the main text as follows:

Lines 232-235

"We consistently find a transition from many distinct communities to a single coherent community which spans the entire nightside at peak expansion. In contrast, Figures S29-S32 from the SI show the community structure of two flow burst events (Grocott et al. [2004]; Gallardo-Lacourt et al. 2014), neither of which transition into a large-scale community with spatial extent across the nightside."

Figure 29. The community structure of a flow burst on 07/09/2001 identified in⁹. The abscissa of all panels is universal time. The times and locations highlighted in Figure 2 of⁹ are indicated here by pink boxes (panels 1-4) and dashed lines (panels 5-6). Panels 1-2 plot the community structure where the size of the circle reflects the normalized number of connections within the community, the ordinate plots the mean MLT/MLAT of the community, $\bar{\theta}_x(t')$ and $\bar{\phi}_x(t')$, and the color indicates the proportion of connections with each lag, $|\tau_c|$. Panels 3-4 show the spatial extent of each community, where the dots are the specific location of the magnetometers and the shading is the extent. Color represents the mean MLT of the stations contained within each community, $\bar{\theta}_x(t')$. Black magnetometers are not connected to the network. Panel 5 plots the modularity, Q . Panel 6 plots the normalized number of connections, $\alpha(t')$, within the nightside. The right ordinate plots (negative) SML.

Figure 30. The community structure of a flow burst on 07/09/2001 (the event shown in Figure 29), is plotted in the same format as Figure 1 (which shows a different event, 16/03/1997). Connections are colour coded to each community, black connections are inter-community connections. Magnetometers noted in⁹, Figure 2 are labelled here for reference.

Figure 31. The community structure of a flow burst on 05/02/2008, identified in ¹⁰ is plotted in the same format as Figure 29. The times and locations highlighted in Figure 4 of ¹⁰ are indicated here with pink boxes (panels 1-4) and dashed lines (panels 5-6).

Figure 32. The community structure of a flow burst on 05/09/2001 (the event shown in Figure 31) is plotted in the same format as Figure 1 (which shows a different event, 16/03/1997). Connections are colour coded to each community, black connections are inter-community connections.

Point-by point response to the Reviewers

Reviewer #1:

R1 C1: *This paper uses ‘community detection’, which has not been used in space physics, to study the evolution of magnetospheric substorms as measured by ground magnetometers in the form of the substorm current wedge (SCW). I believe pulling in new techniques from other disciplines is worthwhile, and from all indications the concept of ‘communities’ has the potential for great application in multipoint space physics measurements. So there is a lot to like about this paper, and the authors did a nice job of explaining a complicated topic. Below I’m going to list a number of criticisms with the paper; but there are some very nice aspects to the paper.*

My primary criticism, which I detail below, is that I don’t believe the conclusion that substorms do not proceed solely by small-scale wedgelets is supported by the analysis. In fact, if I understand the analysis correctly – and I may not be! – I believe the study is incapable of answering the question as posed. In addition to the comments below, I think it would be very helpful to run this technique on one of the published events that definitely sees a wedgelet current system. E.g. Grocott et al. [2004];

there are several. Demonstrate that the technique can identify a wedgelet system. Without that confirmation, it is difficult to determine what a negative result really means.

Response: As stated above, we have edited the end of our abstract to read:

“All 40+ substorms analysed exhibit robust structural change from many small, uncorrelated current systems before substorm onset, to a large spatially-extended coherent system, ~10 minutes after onset. We interpret this as strong indication that the auroral electrojet system during substorm expansions is inherently a large-scale phenomenon and is not solely due to many meso-scale wedgelets.”

This is our key result. It relies on quantitatively establishing that in all of these events there is the formation of an extended spatial region encompassing many ground-based magnetometers, all of which are spatially and temporally correlated. It does not rely on being able to distinguish small scale, uncorrelated structures, i.e wedgelets. Wedgelets may indeed be present and may indeed precede, and may even be involved in, the formation of the large scale electrojet system.

This point is already discussed in detail in paragraph 3 of the Conclusions, which we have edited in an attempt to clarify, and we now add text in the Introduction which summarizes it to make it absolutely clear as to what the ‘question posed’ is.

Lines 58-60

“Our analysis reveals, across many events, an extensive coherent structure which is consistent with a spatially-extended correlated current system. It establishes that a large-scale SCW is an essential part of substorm evolution, but does not exclude the co-existence of smaller scale structures”

Lines 307-318

“As our technique is based on correlated magnetometer data we cannot resolve structures that are on spatial scales smaller than that of the inter-magnetometer spacing, nor on short temporal scales of less than 1 minute, thus although we are able to resolve flow bursts (see SI for examples) we cannot resolve short-timescale, small-scale events such as individual bursty bulk flows or wedgelets if they are below this resolution. However, unless the wedgelets are spatially and temporally correlated with each other, we would not expect to see the spatially coherent signature of global cross-correlation seen across the auroral bulge. We cannot rule out the scenario in which two or more spatio-temporally correlated currents would appear as one community because they would not be able to be resolved by ground-based magnetometers. Nonetheless, the spatially-extended communities we observe cannot be obtained by having many, small, spatially localized wedgelets which are each internally correlated, but not cross-correlated with each other. All 40+ substorms analyzed here ultimately form a large-scale coherent current wedge structure. The structural shift from multiple to a spatially-extended current system occurs approximately 10 minutes after onset. It excludes models in which the current system is solely comprised of individual, uncorrelated wedgelets and indicates that the auroral electrojet system is inherently a large-scale phenomenon.”

R1 C2: *The paper describes a wedgelet/SCW scenario in which the ionospheric segment of individual wedgelets appear as an extensive electrojet (lines 52-57). But I don’t believe this is the commonly accepted viewpoint of how BBFs/flows create the SCW. The many simulation papers of Birn, and the recent one by Merkin [2019] (all referenced), argue that BBFS enter the SCW generation region, and contribute to an overall larger region of enhanced pressure, and very quickly the mesoscale structure of the BBF has disappeared, consumed within the larger pressure gradient in the inner magnetosphere. This overall enhanced pressure is what drives the SCW, not the summation of distinct individual wedgelets that remain. I think this is the widely accepted scenario. The picture of Liu et al. [2018] I think is what the authors are taking issue with (I would take issue with it as well), as*

this is expanded upon in lines 422-425. But I don't believe it correct to lump the 2 pictures together. So I believe the paper needs to do a better job of cleanly motivating the study, in that the authors are testing (I think) the idea that wedgelets remain distinct within the overall SCW picture. Which, I agree with the authors here, is probably incorrect.

Response: Indeed, the 'summation' of individual uncorrelated wedgelets would not produce a single large scale spatially correlated structure, and so we have falsified this scenario. Our results confirm a scenario where a large scale SCW forms. This may indeed follow on and be driven by multiple wedgelets/BBF. We have rewritten the text around these simulation papers (paragraph 2 of the Introduction) to distinguish these scenarios.

Lines 23-36

"Recently there has been a renewed interest in the meso-scale structure with the scenario of the SCW comprising of individual wedgelets gaining attention (Birn & Hesse, 2014; Birn et al., 2019; Forsyth et al., 2014; Liu et al., 2015, 2018; Malykhin et al., 2018; Palin et al., 2016; Panov et al., 2016; Merkin et al., 2019). In this scenario, the wedgelets are small 3D wedge-like current systems associated with magnetospheric dipolarizing flux bundles (Liu et al., 2013) which are magnetic structures associated with bursty bulk flows (BBFs) (Angelopoulos et al., 1992). Liu et al. [2013] hypothesized that the ionospheric segments of a series of individual wedgelets, separated in local time, would appear as a single extensive electrojet from ground magnetometers. The picture by Birn et al. [2019] is somewhat different as they suggested the individual BBF's lead to a pressure pileup in the inner magnetosphere which then result in a large-scale SCW. Merkin et al. [2019] demonstrated that magnetic flux accumulation in the inner magnetosphere in the expansion phase of a substorm was dominated by azimuthally localized plasma flows; this process was also accompanied by a plasma pressure build-up. However, they did not directly address the question of the SCW composition by wedgelets (private communication with Dr. Merkin). Coupling the magnetospheric process of BBF's to the SCW is appealing as it points to a fundamental property of magnetospheric convection in that BBFs are widely considered as a process that can resolve the pressure balance inconsistency in the magnetotail (Pontius Jr & Wolf, 1990). Although, simulations (Birn et al., 2019; Cramer et al., 2017; Sorathia et al., 2018; Ukhorskiy et al., 2018; Merkin et al., 2019) provide a powerful means to link ground and space-born magnetometer observations to magnetospheric processes a convincing comparison remains elusive (Forsyth et al., 2014)."

R1 C3: *That said, within the merging/pressure gradient scenario, the current study cannot address the contribution of wedgelets. But it's also unclear to me if the current study would be able to pick out individual wedgelets of the Liu et al. scenario, as I have a number of questions about the methodology below.*

Response: We have now included examples of our analysis method applied to flow bursts in the SI- these show quite unambiguously that we can detect these structures. However it is not the stated aim of this paper to address the contribution of wedgelets, or to pick out individual wedgelets as in Liu et al. Whilst our methodology does resolve small scale structures, inevitably, there can be structures that are smaller than the inter-ground station spacing as we have discussed in the conclusions. The essential point is that we do confirm that a large-scale spatially correlated SCW is generally an essential part of substorm evolution. We do not need to detect small scale wedgelets in order to confirm that a large-scale structure is present. We need, and are able, to test for a signature -spatially extended correlation- quantifiable across many events, that cannot be reproduced by uncorrelated wedgelets alone. This is what our methodology is designed to do.

Reviewer: *Questions/concerns about the methodology:*

R1 C4: 1) “good” magnetometer coverage is defined as 2 or more stations in each 3 hour window of the nightside (lines 226-229). I think this would actually lead to poor coverage. For example, 2 stations could be at the duskward edge of the pre-midnight window, 2 could be at the dawnward edge of the postmidnight window, and the separation between the pairs could be almost 6 hours of LT. That’s a lot. In practice, how often does this happen? I don’t know, but this definition does not constrain coverage very well, and quick flip through the examples shows many cases where this is true.

Response: Below is a plot of the distribution of the magnetometers at the time of onset for the 41 substorms in MT. Most substorms have many more than the required 2 magnetometers per 3 hours of local time. Further, a rough estimation of 6 hours of local time is $\sim 4000\text{km}$ in geodetic distance. Here we have included a histogram with the geodetic distances between each magnetometer and its two nearest neighbours (from the 41 substorms used in the MT). Separations of $>1000\text{km}$ are rare. We have added both figures to the SI and include the following text:

Lines 88-89

“Figures plotting the spatial distribution and geodetic magnetometer separation, for each of these 41 substorms. are given in the SI (Figures S26-S27).”

Lines 263-264

“(see SI Figures S26-S27 for plots of the magnetometer coverage).”

Figure 26 describes the distribution of magnetometers at the time of onset (from polar VIS images) for each of the 41 substorms included in main text. Figure 27 is a histogram of the minimum separation distances between magnetometers.

Figure 26. The magnetic local time (top panel) and magnetic latitude (bottom panel) of magnetometers at the time of onset (time determined from polar VIS images⁸) for each of the 41 substorms included in main text. The individual substorm events are arranged along the x axis. Horizontal grey lines separate the nightside into three hour segments in MLT. $\sim 90\%$ of 3 hour segments contain more than the minimum of two magnetometers required for analysis.

Figure 27. A histogram of the geodetic distances between each magnetometer and its two nearest neighbours (for the 41 substorms used in the main text). Separations of $> 1000\text{km}$ are rare.

R1 C5: 2) *There are a couple of aspects of the study that I think emphasize picking out large scale features while minimizing the small scale features the study wants to identify. First, the time normalization is a good approach for the superposed aspect of the study, but introduces a bias against short duration structures. For example, the 16-Mar-1997 example in the paper had onset at 05:41 UT, and going by the SML index peak expansion was about 06:30 UT. That's a 50 minute substorm, which was contracted into a 30 minute time-series. A wedgelet current system that may last only a few minutes in 'real' time, is contracted further, making it difficult to pull out in a 128 minute cross-correlation.*

Response: The cross correlation and the calculation of the network and its parameters is carried out first, in the 'real time' domain. The normalization to a uniform time-base is then applied to the time dependent network parameters.

The typical duration of a BBF associated wedgelet or streamer is 10-12 min (e.g. Angelopoulos et al., 1992, Gallardo-Lacourt et al. 2014). The time of maximum auroral bulge expansion, in the case of the example from the main text, is at 06:20 UT by Polar VIS images; therefore it a 40 minute time series. This is slightly above the average but provides good resolution for an example of the method (also see Gjerloev et al., 2007, Fig 4). We have clarified this with the text:

Lines 94-95

"The network and all associated parameters are calculated at the resolution of the magnetometer data (1 minute) before being rescaled onto the normalized time-base."

R1 C6: Relatedly, in a 128 minute cross correlation, I have a hard time believing that it would key off of a temporally short and importantly small relative amplitude signal of a wedgelet. More likely is it'll key off of the temporally and dynamically larger integrated SCW system. I'd like to see an example of what a cross-correlation looks like. Even in the Liu et al. scenario, how long could one reasonably expect a wedgelet system to remain intact, and would it be picked out by this technique? The authors need to make that case.

Response: As discussed above, we are able to resolve wedgelets with our analysis method. The (canonical) cross-correlation time window needs to be sufficiently long to ensure accuracy so that a 128 min window gives a 128 point cross-correlation. Importantly, this is a cross-correlation not a time average. It can still resolve changes on timescales much shorter than that of the window. The key signature in this study is onset, that is a sharp ramp in activity in time as the SCW forms. We have demonstrated that this can be time-resolved with model data in Dods et al 2017 (Figure B1) which we reproduce here.

Figure B1. The figure from top to bottom shows the average cross correlation between all test signals as a function of delay since the turning, the $\tanh(t)$ signal (exactly the same for all test signals), an example $R_i(t)$, and an example of the noise and the signal combined, $S_j(t)$.

Dods et al. uses a signal $S_i(t) = \tanh(t) + R_i(t)$, where $R(t)$ is randomized noise and i references the test stations. The tanh function is representative of substorm onset. The average correlation clearly resolves the time step.

Below is an example of canonical cross correlation for a test set of twelve time series. This figure has been added to the SI as Figure S28 with the caption

51 **Figure 28**

Two types of continuous functions are used to model the magnetic perturbations associated with storms. The first, $f(n)$ is a sharp linear decrease with an exponential decay. $f(n)$ is given by

$$f(t) = - \begin{cases} 0 & t \leq 0, \\ \frac{t}{4} & 0 \leq t \leq 4, \\ e^{-\frac{t-4}{300}} & 4 \leq t, \end{cases} \quad (1)$$

52 The function is used to approximate the north, B_N , east, B_E , and down, B_Z , components of the vector magnetic field
 53 perturbations observed at a magnetometer. Random variables are drawn from the standard normal distribution and multiplied
 54 such that noise with the maximum amplitude 30 nT is added to the function; noise is denoted by $N(t)$. The north component is
 55 scaled by 400 nT and the east and down components are scaled by 300 nT, e.g. in figure 28, $B_N(t) = N(t) + f_1(t) \times (400 \times 10^{-9})$.

56 We model the three component vector magnetic field perturbations expected as a response of twelve magnetometers to a
 57 substorm using this described method. For each magnetometer pair we then take a window, of length 128, linearly detrend
 58 the time series and calculated the canonical cross correlation. This is repeated for each magnetometer pair with a sliding window.
 59

Figure 28. Canonical cross correlation for a test set of twelve modelled vector time series. The top panel shows one time series of the modelled data which is the three component function, $f(t)$ (equation 1), with a sharp linear decrease, $f(t) = \frac{t}{4}$ and a slow exponential decay $f(t) = e^{-\frac{t-4}{300}}$. Noise is white with amplitude of 30nT. The second panel shows the mean canonical cross correlation across all 12 nodes. The correlation begins to increase within minutes of the sharp decrease. The third panel shows the normalized number of connections the network would have for different uniform cross correlation thresholds. The network responds as soon as the canonical cross correlation reaches the network threshold.

Further, in Figure 2 of Orr et al. 2019 (edited and attached here), we can see that within the expansion wedge (from polar VIS, region B in figure), the response to onset, as seen by the network,

is resolved in <10 minutes for a given single event and the mean of multiple events.

We have added text in the methods section on. To clarify:

Lines 111-117

“The 128 minute window is chosen to give sufficient accuracy in the computed cross-correlation function, whilst still capturing the large-scale spatio-temporal current system behavior. Dods et al. 2017 previously demonstrated, by use of model time series, that this window length resolves changes on much shorter timescales than that of the window length, specifically capturing a sharp ramp in activity, such as when the SCW forms. Figure S28 in the SI demonstrates likewise, using a 128 minute CCC window. In Dods et al. 2015, the technique was applied using zero-lag CCC and trialed to obtain the un-directed network for a small set of isolated substorm events. This was sufficient to capture the initial spatially coherent response at onset. “

R1 C7: 3) *The spatial coverage is key to making this technique work for the intended purpose. Figure 2b was really nicely done, but highlights the issue. Onset in auroral imagery is just post midnight, where there is 1 station. So the technique will not pick out a community until the coherence expands to the next station, which is ~2 hours of LT duskward, which is larger than the scale size of wedgelet systems. In other words, unless you had a dense network of magnetometers, this current study would miss the wedgelet. And the criterion of 2 stations in 3 MLT hours I think is insufficient (see e.g., any number of papers that have ‘imaged’ the current system of BBFs/wedgelets using inversion techniques).*

Response: As above, we have designed this methodology to discriminate when a large-scale spatially correlated structure is present, not to detect wedgelets. Perhaps it is worth highlighting here the quantitative statistical nature of this study which we feel is a particular strength. Our aim has been to construct a method that robustly tests a conjecture: [is a large-scale SCW present?] and then apply it across a large set of events. We can quantitatively confirm that a transition to such a large-scale structure is generally present in substorms, and indeed, propose a robust quantitative test that could be applied to simulations as well. Contrast this with the conjecture: [is a wedgelet present?], where indeed, if conditions are favourable, with sufficient stations, conjugate satellites and so forth, one may detect a wedgelet, if conditions are unfavourable, the wedgelet may be there but may not be detected. Given that arguably there will always be wedgelets that are not detectable, this

conjecture can only be tested on an event-by-event basis and cannot form the basis of a statistical study.

R1 C8: *Further, the study limits stations to $m\text{lat} > 65^\circ$, which explicitly limits measurements to the auroral zone – and often to well poleward of the auroral zone. The reasoning for this was not mentioned, but presumably this was because mid latitude stations would not observe local wedgelet systems. But auroral zone magnetometers during substorms are dominated by local variations, so it's not surprising to not get a 'community' until later in the substorm.*

Response:

All data is from stations between 60-75 degrees magnetic latitude, within the nightside. The low latitude limit ensures that sub-auroral stations are not included. If we move to lower latitudes the stations are located at very large distances from the auroral currents, in fact they measure the net FAC's linking Earth and near-space rather than the electrojets or even the FAC's themselves. Further, the spatial smearing prevents us from seeing much other than global scale currents. The high latitude boundary also ensures the stations are primarily from the auroral zone. Poleward of this the stations no longer measure the auroral electrojets but the net FAC's linking the auroral zone to the magnetospheric source (e.g. Laundal et al., 2015). Much like at mid-latitudes. Finally it should be mentioned that all substorms included in the MT have an onset latitude (as found from polar vis) within this region (also see Gjerloev et al., 2007, Fig 6).

Text has been added to clarify:

Lines 83-84

"Magnetometers within the nightside, from 18-6 hours of magnetic local time (MLT), between 60-75 degrees magnetic latitude (MLAT), are chosen to best observe the auroral electrojets (Laundal et al., 2015, Gjerloev, 2007)."

R1 C9: *4) Figure 2 also highlights that I think the technique is pulling out the WTS/auroral bulge, at least initially (Figure 2c). I think this explains why the Q factor drops 1/3 – 2/3 of the way into a substorm expansion. You don't get largescale coherence until the WTS forms. It also explains why, in Figure 1, you don't see much indication that the substorm has occurred until T=10.*

Response: Agreed we are detecting the formation of a large scale correlated system. Our understanding of these concepts is that the WTS is merely the westward part of the substorm bulge and it is associated with bright emissions and a strong upward FAC. The latter is essentially draining the westward electrojet current. This electrojet region is only a few degrees wide and located just inside the open-closed field-line boundary. The WTS FAC and this electrojet are part of the SCW that

widens in LT and moves poleward following the substorm onset (e.g. Gjerloev et al., 2007, Fig 15). This implies that the SCW has a small LT width at the onset and thus a small probability of having a ground station at that location. This is our interpretation of the 10 min delay.

R1 C10: *Finally, at several points the paper notes that the substorm starts off initially as non coherent/distinct communities that eventually merge into a single coherent community at substorm peak expansion. I think that is in agreement with the widespread understanding of how BBFs/flows accumulate to create and sustain the SCW, as I first described above. However, because of the issues listed above, I'm not sure the study would have reached any other conclusion. That is, because of the sparse ground network and the use of 128 minute windows, I think the study could only come to the conclusion that the SCW is not spatially coherent until expansion is well underway.*

Response: On the contrary, as we are using cross-correlation it is not sufficient that BBFS 'accumulate', or that wedgelets are averaged, or superimposed, to produce a spatially extended signature. If BBFs or wedgelets are the ultimate drivers of the substorm, our results clearly show that some physics mechanism is needed for them to cohere into a large-scale correlated spatially extended structure. It may indeed be the case that this is what is going on, in which case our results sharpen this scenario of substorm dynamics in a testable manner. We have added text in the introduction to emphasise this point:

Lines 53-58

"Multiple discrete current systems that are present before onset are found to progressively transition into a coherent SCW. Since our methodology quantifies cross-correlation between spatially separated stations, this transition is to a coherent large-scale spatially extended structure, rather than solely an accumulation of smaller scale wedgelets or flow bursts that are not coherent with each other. This transition occurs over 10-20 minutes following onset and characterizes the peak expansion phase of the substorm."

R1 C11: *Also, if the authors are really arguing specifically against a specific model of distinct wedgelets that remain intact, that should be made explicit in the beginning. And the authors should then detail what they would expect to see if this was really happening. I think the authors are arguing that because the SCW eventually becomes coherent – 2/3 of the way into the substorm – the model cannot be correct. But let's say that the model is correct. Would the model show a set of communities within the auroral bulge, each representing a distinct wedgelet? I think that's what the authors are arguing, but it isn't very clear. But, again, based on the issues above, I don't believe there are sufficient magnetometer points to see this, even if it was occurring.*

Response: We have designed methodology to specifically discern the presence or absence of a spatially extended coherent (i.e. correlated) structure and have amended the text to further clarify this. If 'the model' predicts such a structure then it is correct. If on the other hand 'the model' predicts the accumulation of smaller structures, which are uncorrelated with each other, then it is incorrect. Whether or not the wedgelets remain 'distinct and intact', the conclusion is the same, either they result in a spatially extended, correlated structure, or they do not.

We have also included a schematic of the current systems we would expect to correspond to low or high modularity in the SI, and added text to the MT to explain what we interpret modularity to mean in terms of currents:

Lines 166-171

"The change in the modularity as the substorm progresses identifies how these current systems are changing. The modularity will be maximum when spatially localized coherent perturbations cause a

group/community of magnetometers to be internally correlated, but not cross-correlated with other groups/communities of magnetometers. If all auroral latitude magnetometers are highly correlated due to a large-scale ionospheric current system overhead, the modularity will be near-zero. A schematic of the current system models we would expect from high or low modularity is contained in the SI, Figure S36.”

Figure 36. A schematic showing our interpretation of high or low modularity. Modularity will be near-zero when all auroral latitude magnetometers are highly correlated due to a large-scale ionospheric current system overhead, e.g.¹⁵. The modularity will be maximum when spatially localized coherent perturbations cause a group/community of magnetometers to be internally correlated, but not cross-correlated with other groups/communities of magnetometers e.g.¹⁶. If current systems were highly correlated and not spatially distinct e.g.¹⁷ they could appear as a single system (low modularity) under our analysis.

Reviewer #2

We are truly grateful for the detailed report provided by the reviewer. Both the extensive insight/interpretations, the historical aspects and the analysis.

We have spent considerable amount of time to respond to the report and realize that some of the dispute is merely due to a misunderstanding. In lines 70-79 the event selection and timing is briefly explained and the referenced paper provide robust support. The list is a subset of the substorm list compiled by Gjerloev et al. [2007] and used in a long list of published papers. Substorms were identified and their timing determined from global auroral images which is arguably the most convincing and accepted technique. In short – the substorms selected are indeed isolated events occurring during non-storm conditions, the timing is based on the widely accepted superior technique, and the list has been validated through a long list of published papers. Thus, inconsistencies with the indices discussed by the reviewer are due to the inherent limitations of those indices. We have clarified these points in detail below. We have additionally added a glossary

of the terms which the reviewer has highlighted as new to the field and attempted to add further explanation to the text where appropriate.

Below we discuss each point made by the reviewer.

R2 C1:

EVALUATION

I do not find fault with the mathematics or statistical procedures used although I was unable to follow their description of the analysis procedure for lack of detailed explanation of several concepts.

Response: We have added a 'glossary' to the SI which details these concepts, with diagrams where appropriate. We do appreciate that this is an interdisciplinary approach and we are aiming to bring concepts, whilst well established in network science, to a space physics problem and this is quite new. The glossary and relevant diagrams have been referenced throughout the text when introducing new terms.

R2 C2: *I also agree with their conclusions as I have also criticized the multiple and simultaneous wedgelet explanation of the current wedge.*

Instead I have advocated for a coherent source that is localized in space but produced by the accumulation of multiple flow bursts. My main objections to this paper involve the absence of a clear physical description of how the magnetic perturbations used in the analysis are produced by processes in the magnetotail, and how their method would detect them.

Response: Our aim has been to construct a method that robustly tests a clear conjecture: [are large-scale SCW present?] and then apply it across a large set of events. We can quantitatively confirm that a transition to such large-scale structures are generally present in substorms, and indeed, propose a robust quantitative test for it that could be applied to simulations as well. Our results confirm any physical description or model that predicts such a large-scale spatially extended coherent structure. Crucially, if the 'accumulation of multiple flow bursts' drives the formation of such a coherent structure then it is consistent with our results, however, if this model predicts the accumulation of smaller structures, which are uncorrelated with each other, then it is incorrect. We feel that this is a quite useful contribution to the understanding of the physics of substorms.

Lines 17-19

We have added text to the introduction to explain why we are using magnetometers:

"Ground-based magnetometers provide decades worth of time series, with near global-coverage, to address the spatio-temporal coupling of the ionosphere and magnetosphere (gjerloev, 2012) and have been used in a long list of publications to study the auroral electrojet system, e.g. (nishida, 1971, gjerloev, 2010)."

R2 C3: *I am also concerned with details of the substorm selection and onset timing. I am unable to determine what effects these details have on their analysis or conclusion.*

Response: We discuss this in detail below.

R2 C4: *Finally I do not feel the authors make the case that their approach disproves the wedgelet ideas. My recommendation is significant revisions are needed.*

Response: Our analysis has at its core cross-correlation between the signals at spatially separated ground-based magnetometer stations. It only gives a positive result in the presence of spatial

coherence across these signals. Structures on a smaller spatial scale will simply not give such a globally coherent signal. We carefully tested our results using random phase surrogates which destroy phase information whilst preserving all other aspects of the signals to discount the possibility that small scale structures below the spatial separation of the stations, or noise could give a positive result.

R2 C5: *HISTORICAL NOTES*

Response: We appreciate this nice summary and do apologize that the referencing is tilted towards the papers published by Gjerloev. This is not to dismiss other papers but simply because it is focused on the details of the study at hand and after all this is not a review paper, we are restricted in the number of citations.

Reviewer: *GENERAL COMMENTS*

R2 C6: *My first question is if you take account of past work why would you expect to find correlations between stations averaged over two hours (128 min) and the entire nightside auroral zone to reveal the presence of a wedgelet perturbation localized in local time (<1 hour), magnetic latitude, and in time (< 15 min)?*

Response: We have designed methodology to specifically discern the presence or absence of a spatially extended coherent (ie correlated) structure and have amended the text to further clarify this. Crucially this is not an 'average over two hours', we have demonstrated using modelling (published previously, but additional modelling included here in the revised SI) that the windowed cross-correlation can detect relatively fast changes on a timescale of 10 minutes or less. In our revision we also include some examples of our analysis method applied to known flow burst events and show that we can indeed detect them.

The (canonical) cross-correlation time window needs to be sufficiently long to ensure accuracy so that a 128 min window gives a 128 point cross-correlation. Importantly, this is a cross-correlation not a time average. It can still resolve changes on timescales much shorter than that of the window. The key signature in this study is onset, that is a sharp ramp in activity in time as the SCW forms. We have demonstrated that this can be time-resolved with model data in Dods et al 2017 (Figure B1)

which we reproduce here.

Figure B1. The figure from top to bottom shows the average cross correlation between all test signals as a function of delay since the turning, the $\tanh(t)$ signal (exactly the same for all test signals), an example $R_i(t)$, and an example of the noise and the signal combined, $S_j(t)$.

Dods et al. uses a signal $S_i(t) = \tanh(t) + R_i(t)$, where $R(t)$ is randomized noise and i references the test stations. The \tanh function is representative of substorm onset. The average correlation clearly resolves the time step.

Below is an example of canonical cross correlation for a test set of twelve time series. This figure has been added to the SI as Figure S28 with the caption

51 **Figure 28**

52 Two types of continuous functions are used to model the magnetic perturbations associated with storms. The first, $f(n)$ is a
 53 sharp linear decrease with an exponential decay. $f(n)$ is given by

$$f(t) = - \begin{cases} 0 & t \leq 0, \\ \frac{t}{4} & 0 \leq t \leq 4, \\ e^{-\frac{t-4}{300}} & 4 \leq t, \end{cases} \quad (1)$$

54 The function is used to approximate the north, B_N , east, B_E , and down, B_Z , components of the vector magnetic field
 55 perturbations observed at a magnetometer. Random variables are drawn from the standard normal distribution and multiplied
 56 such that noise with the maximum amplitude 30 nT is added to the function; noise is denoted by $N(t)$. The north component is
 57 scaled by 400 nT and the east and down components are scaled by 300 nT, e.g. in figure 28, $B_N(t) = N(t) + f_1(t) \times (400 \times 10^{-9})$.

58 We model the three component vector magnetic field perturbations expected as a response of twelve magnetometers to a
 59 substorm using this described method. For each magnetometer pair we then take a window, of length 128, linearly detrend
 the time series and calculated the canonical cross correlation. This is repeated for each magnetometer pair with a sliding window.

Figure 28. Canonical cross correlation for a test set of twelve modelled vector time series. The top panel shows one time series of the modelled data which is the three component function, $f(t)$ (equation 1), with a sharp linear decrease, $f(t) = \frac{t}{4}$ and a slow exponential decay $f(t) = e^{-\frac{t-4}{300}}$. Noise is white with amplitude of 30nT. The second panel shows the mean canonical cross correlation across all 12 nodes. The correlation begins to increase within minutes of the sharp decrease. The third panel shows the normalized number of connections the network would have for different uniform cross correlation thresholds. The network responds as soon as the canonical cross correlation reaches the network threshold.

Further, in Figure 2 of Orr et al. 2019 (edited and attached here), we can see that within the expansion wedge (from polar VIS, region B in figure), the response to onset, as seen by the network,

is resolved in <10 minutes for a given single event and the mean of multiple events.

We have added text in the methods section on. To clarify:

Lines 111-117

“The 128 minute window is chosen to give sufficient accuracy in the computed cross-correlation function, whilst still capturing the large-scale spatio-temporal current system behavior. Dods et al. 2017 previously demonstrated, by use of model time series, that this window length resolves changes on much shorter timescales than that of the window length, specifically capturing a sharp ramp in activity, such as when the SCW forms. Figure S27 in the SI demonstrates likewise, using a 128 minute CCC window. In dods et al. 2015, the technique was applied using zero-lag CCC and trialed to obtain the un-directed network for a small set of isolated substorm events. This was sufficient to capture the initial spatially coherent response at onset. “

R2 C7: *My second question is why use auroral zone data to study the current wedge? The current wedge is most clearly seen in midlatitude data where the systematic growth of specific patterns in X and Y components is seen over the entire nightside.*

Response: All data is from stations between 60-75 degrees magnetic latitude, within the nightside. The low latitude limit ensures that sub-auroral stations are not included. If we move to lower latitudes the stations are located at very large distances from the auroral currents, in fact they measure the net FAC's linking Earth and near-space rather than the electrojets or even the FAC's themselves. Further, the spatial smearing prevent us from seeing much other than global scale currents. The high latitude boundary also ensures the stations are primarily from the auroral zone. Poleward of this the stations no long measure the auroral electrojets but the net FAC's linking the auroral zone to the magnetospheric source (e.g. Laundal et al., 2015). Much like at mid-latitudes. Finally it should be mentioned that all substorms included in the MT have an onset latitude (as found from polar vis) within this region (also see Gjerloev et al., 2007, Fig 6).

Text has been added to clarify:
Lines 83-84

Magnetometers within the nightside, from 18-6 hours of magnetic local time (MLT), between 60-75 degrees magnetic latitude (MLAT), are chosen to best observe the auroral electrojets (Laundal et al., 2015, Gjerloev, 2007).

R2 C8: *My third concern is the absence of any discussion of the distinction between DP-2 and DP-1. DP-2 is present throughout the substorm driven by the R-1 & R-2 currents. It is globally coherent as shown by previous studies such as AMIE (assimilative modeling of ionospheric dynamics). DP-1 (the substorm current wedge) is an independent current added to the pre-existing currents. The analysis does not include a procedure for separation of these two systems so it is difficult to see how a network of correlations proves anything about how the current wedge is formed. See for example the use of principal component analysis to carry out this task (Sun et al., 1998).*

Response: Our method resolves spatially extended coherent structure and (by means of random phase surrogates) distinguishes it from large amplitude but phase incoherent signals (ie 'noise'). We resolve the spatially extended coherent structure that follows substorm onset (seen in auroral imaging) and is otherwise not present by setting the cross-correlation threshold for constructing the network to pick-out this signature.

Time changing global convection, ie DP2 can be picked up by a suitable choice of threshold. We demonstrated this in Dods et al (2017) where we used a similar analysis to study the effect of isolated northward and southward turnings under otherwise quiet conditions. A first step in our analysis it to remove the slow trend in the data on the timescale of the window which removes any constant background- as is essential for any Fourier based analysis. Therefore we have established that we are resolving the large scale current response to a substorm.

Furthermore we do not claim to 'prove anything about how the current wedge is formed', rather we have constructed methodology that unambiguously tests a clear hypothesis: is there a spatially extended coherent current system initiated generally (ie across many events) in substorms, or not? This adds to our understanding of physical mechanism in that it categorically excludes any model that does not predict the formation of a large scale coherent current system. Of course one can always decompose any spatial field into modes by methods such as PCA. One can indeed customize such analysis to pick out modes on particular spatial scales of interest such as those coincident with particular component current systems. This amounts to a formal categorization of features seen in the spatial field, in terms of already established terminology. Our analysis is carefully constructed to

avoid any such categorization so that our result is completely general, and does not rely on any such underlying categorization.

R2 C9: *A major concern on which I can comment is related to the list of substorm onsets provided in the Supporting Information. This list has 38 rows and two columns totaling 76 events. These were presumably timed using a combination of SML index records and Polar Satellite images. The authors say they used 41 events for their analysis but do not specify which events. When I examined the first event used in Figure 1 of this paper (05:41 on March 16, 1997) I found I did not agree with the timing of the event...*

Response: We have carefully reviewed the detailed analysis of the Reviewer and are able to clarify all of the points raised as follows:

A) Substorm Timing.

The reviewer discusses to great length the detailed timing of the substorm events. We have added the following text:

Lines 71-74

"We study a subset of events drawn from the list of isolated substorms between 1997-2001 established by Gjerloev et al. [2007] and also described in Gjerloev and Hoffman [2014]. They determined the substorm timing (onset and peak expansion) solely from global auroral images rather than magnetometer data or magnetic. The substorms were selected such that..."

The selection criteria and the reasons for these were carefully discussed and illustrated in the above paper and in a long list of subsequent papers.

Historically it is worth mentioning that the original substorm paper by Akasofu used all-sky images. The use of AL is convenient as it provides an uninterrupted time series that stretches many decades. The mid-latitude bay index that the reviewer mentions is a logical parameter in light of the ground-breaking McPherron substorm-current-wedge paper although this index, so far, has limited community usage.

Substorm onset identification:

- 1) Images. We argue that auroral images provide the most objective and clear onset identification. E.g. the THEMIS mission acknowledged this technique and we would argue that this is generally acknowledged as the most reliable and objective technique.
- 2) AL index. Using AL has been widely used historically – largely due to convenience. However, when compared to AL, Newell and Gjerloev [2011], showed that the 12 stations of AL are unlikely to be positioned under the onset location and hence AL will not show any significant bay until the substorm has expanded in size to cover one of these 12 stations. This implies a finite delay that is only zero in the lucky situation that the onset occurs over one of the 12 stations. To limit this inherent weakness (as well as others) Newell and Gjerloev developed the SML index which is essentially the same as AL but it uses all available stations. They showed that utilizing >100 stations limits the above mentioned delay.
- 3) MPB index. The reviewer states that *"The fact that the average delay is more than 10 minutes suggests that their topic of study, the substorm current wedge, does not start at the time assumed in their analysis."* The conclusion is based on the MPB index and as we explain below this is simply not correct. Using the mid-latitude bay index has very limited usage and thus it has not undergone any critical scrutiny (as far as we know). However, the index is based on mid-latitude stations which thus are located at large distances from the net-field-aligned currents that feed and drain the substorm-current wedge. The magnetic bay is

cleaner than at auroral latitudes due to the large distance which essentially works as a spatial low-pass filter. This, however, also has the unfortunate effect that the two net field-aligned current of the substorm current wedge must separate widely enough in local time before they can be detected at mid-latitudes. Gjerloev have provided this simple explanation in the past and the reviewer alludes to it. The substorm current wedge simply need to expand sufficiently before these stations can measure them. Further, the large distance leads to a smaller signal than at auroral latitudes which can lead to a delay in identification. Finally, adding to this delay are the oceans at mid-latitudes which lead to a probabilistic delay similar to the above discussed AL index delay.

To illustrate these points we use the example r1 provided by the reviewer.

Onset time for March 16, 1997:

Polar VIS: 05:41:00
 SML: 05:49:00 or 05:50:00 provided by reviewer
 AL: 05:51:00 or 05:52:00 depending on def.
 MPB: 05:52:00 provided by reviewer

The sequence of this substorm as seen in the Polar VIS images can be found on SuperMAG but here is a screenshot from the actual U. Iowa VIS website. The onset is clearly visible in the 05:41 image and the expansion does indeed develop from this onset:

The substorm timing used in this study is likely as good as it ever gets.

4) Other techniques to determine substorm onset timing. Other techniques include for example Pi2 waves observed on ground, magnetospheric dipolarizations and bursty bulk flows observed by S/C. Their acceptance as an indicator has been somewhat mixed due to a variety of reasons (e.g. Pi2 occurring at other than substorm onset times; less than optimal S/C location in the magnetosphere).

In short for the auroral substorm:

- Global auroral images provide exact onset timing

- SML onsets will on average have a delay but in events the delay could be zero
- AL onsets will on average have a delay (>SML) but in events the delay could be zero
- MPB will by definition have a delay
- Other techniques have been used with somewhat mixed acceptance

These systematic delays are also seen in the reviewers figures r1-r3.

Onset peak.

We again use the timing provided by the above mentioned Gjerloev and Hoffman list. They selected the peak solely from images. Gjerloev et al. [2007] showed that on average the substorm peak as identified from UV images did indeed coincide with the peak of AL.

B) Substorm Selection.

The reviewer points out concerns (as identified from SML) regarding the substorm selected. Selection was based primarily on the Polar VIS images ensuring that no disturbance was taking place but AL and Dst were also checked to ensure that this was not storm time conditions and that the electrojets were relatively quiet. A clear isolate onset and subsequent expansion to a bulge-type or Akasofu-type substorm was required. No consideration was made as to whether the event was associated with a positive bay or for that matter a negative bay.

The reviewer also refers to the following temporal evolution of the SML. The behavior of this index as a function of substorm time is complex due to several facts:

- The intensity of the westward electrojet system is dynamic and different parts of the system can evolve differently. Competing substorm current wedge models keep being proposed.
- The spatial location and extend of the westward electrojet system is also dynamic and ground stations which may place or remove stations from monitoring the electrojet.
- The non-uniform station location further complicates the above two issues.

We also refer to the paper by Gjerloev et al. [2004] in which they discussed the behavior of the electrojet indices as a function of substorm time.

C) Substorm normalization.

Figure r4 is based on an amplitude and onset timing analysis. It unfortunately does not include an important normalization of the duration of the expansion phase. Gjerloev and Hoffman [2008] showed that without this significant smearing of the feature in mind will occur. The duration of the expansion phase vary widely (e.g. Gjerloev and Hoffman, 2007; and several preceding papers). In the present paper we do follow the conclusions of that paper.

Reviewer: *CONCLUSIONS*

Response: We have addressed these points above

Reviewer: *SPECIFIC COMMENTS*

R2 C10: Abstract

The paper appears to be directed to the magnetospheric research community as it begins with a discussion of the substorm current wedge and the question of the structure of the ionospheric currents at the end of the current wedge. However, the bulk of the presentation emphasizes the technique of network structure and barely discusses the geophysical issues. I am not convinced that the technique used is capable of answering the geophysical question posed in the abstract.

Response: As stated above in our general comments, we have edited the end of our abstract to read:

“All 40+ substorms analysed exhibit robust structural change from many small, uncorrelated current systems before substorm onset, to a large spatially-extended coherent system, ~10 minutes after onset. We interpret this as strong indication that the auroral electrojet system during substorm expansions is inherently a large-scale phenomenon and is not solely due to many meso-scale wedgelets.”

This is our key result. It relies on quantitatively establishing that in all of these events there is the formation of an extended spatial region encompassing several ground based magnetometers, all of which are spatially and temporally correlated. The technique has cross correlation at its core- it can determine whether the signals seen at two spatially separated stations are coherent or not. We have performed formal tests (construction of random phase surrogates) to confirm that this is a real physical result.

R2 C11: *I note that the abstract reads more like an Introduction complete with citations to the geophysical literature.*

Response: We have edited the abstract to now be in the style of this journal.

R2 C12: *Section 1: At the end of this section it is claimed that there are multiple discrete current systems present prior to expansion onset, i.e. during the substorm growth phase. I would expect that this would be the time when the global DP-2 system is present. Studies of this system show that this is globally coherent.*

Response: Our method resolves spatially extended coherent structure and (by means of random phase surrogates) distinguishes it from large amplitude but phase incoherent signals (ie ‘noise’). We resolve the spatially extended coherent structure that follows substorm onset (seen in auroral imaging) and is otherwise not present by setting the cross-correlation threshold for constructing the network to pick-out this signature. Changes to global convection would be generally be present in the data and can be picked up by a suitable choice of threshold. We demonstrated this in Dods et al (2017) where we used a similar analysis to study the effect of isolated northward and southward turnings under otherwise quiet conditions. A first step in our analysis it to remove the slow trend in the data on the timescale of the window which removes any constant background- as is essential for any Fourier based analysis. Therefore we have established that we are resolving the large scale current response to a substorm.

The DP2 current system has undergone a change as a concept since its introduction by Obayashi and Nishida [1968]. From a basic 2-cell convection pattern this seem logical but as Gjerloev et al. [2010] discussed it is unclear how current continuity is maintained across the terminator. They found that the darkness westward electrojet essentially did not respond to a southward turning of the IMF. In other words –the recent introduction of vastly improved measurement coverage have allowed testing the DP2 concept.

R2 C13: *That this does not show up in the analysis suggests that its magnetic perturbations are swamped by localized noise which is correlated only over a few stations. If there are localized communities of connections at this time then one should look at the local magnetic perturbations to determine if they have an obvious physical cause. After the current wedge forms I would expect its signal to grow to the point it overcomes the local noise and a single coherent system develops. What is the evidence that these merge rather than just disappear?*

Response: First, we can quantify what ‘local noise’ is if this means no phase coherence (ie extended structures) spanning the stations as we have constructed a random phase surrogate in which all

information is retained except the signal phases. This establishes a ‘noise floor’ which can be seen on our plots. Second, agreed we do not know if the smaller scale currents merge or disappear, what we do know is that they are replaced by a single large scale spatially extended system. To clarify this point we have removed all mentions of ‘merge’ from the text.

R2 C14: *Section 2: This section introduces a number of mathematical/statistical concepts that are unlikely to be familiar to most space physicists. More description of these are needed. These include:*

- Network
- Node
- Canonical correlation
- Direction of propagation
- Community detection
- Edge betweenness
- Modularity
- Normalized number of connections
- Adjacency matrix
- Time series normalization

Response: We have added a glossary of terms in the SI

R2 C15: *Line 84: Which substorms? The list provided contains two columns with 38 rows or a total of 76 events. These events are not in any obvious order. Also the time for the end of each expansion is not included.*

The list has been amended (and copied here) to clarify which substorms are included in the main text and other examples. It clearly states that polar VIS timings of onset and maximum auroral bulge expansion were used for the analysis. The method of finding the exact timings is detailed in Gjerloev, 2007, the timings are determined directly from Polar VIS images (freely available on SuperMAG).

Extremely quiet before onset (41 events used in MT)	Almost quiet before onset (11 events used in figure S21)	Not quiet before onset (23 events used in figure S22)
07-Jan-1997, 03:00	22-Jan-1997, 05:00	30-Sep-1997, 08:30
07-Jan-1997, 17:30	03-Mar-1997, 16:30	16-Dec-1997, 21:30
27-Jan-1997, 10:00	06-Feb-1998, 16:30	10-Jan-1998, 16:00
31-Jan-1997, 20:30	26-Nov-1998, 08:00	29-Jan-1998, 16:00
02-Feb-1997, 16:00	18-Jan-1999, 13:00	21-Feb-1998, 20:00
24-Feb-1997, 08:30	26-Mar-1999, 09:00	15-Sep-1998, 06:30
27-Feb-1997, 14:30	21-Oct-1999, 16:00	16-Nov-1998, 02:30
02-Mar-1997, 03:30	02-Nov-1999, 00:00	16-Nov-1998, 21:30
16-Mar-1997, 05:30	29-Dec-1999, 15:30	26-Nov-1998, 16:30
24-Mar-1997, 10:30	07-Jan-2000, 16:00	05-Dec-1998, 14:00
06-Sep-1997, 05:00	31-Oct-2001, 07:30	16-Dec-1998, 10:30
06-Nov-1997, 04:00		01-Jan-1999, 05:00
02-Jan-1998, 13:30		08-Jan-1999, 20:00
04-Jan-1998, 12:30		03-Feb-1999, 22:00
06-Jan-1998, 03:00		15-Feb-1999, 02:00
11-Jan-1998, 07:00		03-Jan-2000, 03:00
20-Jan-1998, 03:30		03-Jan-2000, 19:30
17-Mar-1998, 11:00		09-Feb-2000, 16:30
02-Nov-1998, 19:00		16-Feb-2000, 12:00
03-Dec-1998, 07:00		19-Feb-2000, 10:00
05-Dec-1998, 11:30		25-Feb-2000, 12:00
07-Dec-1998, 11:30		29-Feb-2000, 07:30
07-Dec-1998, 16:00		03-Jan-2001, 12:30
12-Dec-1998, 20:30		
20-Dec-1998, 04:00		
10-Jan-1999, 09:00		
29-Oct-1999, 19:00		
06-Nov-1999, 11:30		
07-Nov-1999, 02:00		
25-Dec-1999, 05:30		
08-Jan-2000, 10:30		
09-Jan-2000, 21:30		
25-Jan-2000, 17:30		
20-Feb-2000, 07:00		
26-Feb-2000, 06:00		
06-Mar-2000, 05:00		
17-Mar-2000, 09:30		
18-Dec-2000, 16:00		
29-Dec-2000, 03:30		
06-Sep-2001, 06:30		
21-Oct-2001, 07:30		

All times are given to the nearest thirty minutes of onset. The list is a subset of events drawn from the list of isolated substorms between 1997-2001 established by Gjerloev et. al, 2007 and also described in Gjerloev et. al, 2014. They determined the substorm timing used with this letter and SI (onset and maximum auroral bulge expansion) solely from global auroral images rather than magnetometer data or magnetic indices.

R2 C16: *Line 113: The procedure for canonical correlation is not described. A search of the space physics literature shows only a few papers that describe this technique. I suspect that this procedure obtains two pseudo time series, one for each station, that is some linear combination of the three field components that have maximum correlation between stations. A fuller explanation is needed here if anyone is going to reproduce this work.*

A glossary has been provided.

R2 C17: *Line 126: The typical duration of a substorm is 150-180 minutes. The duration of a current wedge is less than 70 minutes. It appears that the correlation function cannot change much once the full pulse is inside the moving window of two-hour length. I can imagine that the correlation would not change much for at least one hour as the window passes over the pulse related to current wedge. Please explain how a 2-hour window is able to detect short-term temporal variations. In my experience the correlation functions tends to be dominated by the largest disturbance present in the window so I don't see how it could detect a small and short term perturbation at two stations if a global system is also present.*

Because the substorms are isolated events, specifically chosen to be quiet before onset, the 128 minute window does pick up changes on short time scales. We have demonstrated this with modelling as described in detail above and changes added to the text to clarify.

R2 C18: *Line 132: The concept of "directed" is defined here as delays due to propagation. In the study of current patterns I would think that the sign of the correlation coefficient is also important. In the current wedge the east component (Y or D) is positive pre-midnight and negative post-midnight. It is not clear that this would be true if components are being combined in a canonical correlation. It is true that this requires knowledge of the actual pattern of the magnetic perturbations to prove that the correlations have the correct sign.*

The canonical correlation coefficient does not have a sign as it finds the linear combination of the two time series which are maximally correlated (included in glossary). It does contain direction information in its basis vectors which could be used in a further study but this is not required to test the hypothesis of a large scale correlated spatial structure which is the topic of this paper. The "directed" part of the network here captures the time lag between correlation between magnetometers' vector time series. Here we are mainly using the directed network so that magnetometers which are correlated in time will be included in the network. We have explained canonical correlation and direction in the glossary, along with a schematic showing the method of calculating a directed network connection from two magnetometers vector magnetic field perturbations. We have edited the paragraph describing directed networks, referenced by the reviewer, to clarify

Lines 118-128

"Importantly, to recover the full dynamics that occurs on multiple timescales, the CCC at all time lags are needed. Orr et al. (2019) pioneered this approach, obtaining the first directed networks for the response to substorms seen in the full SuperMAG data set. Including time lagged correlations means that a magnetometer pair which were not 'connected' to the network (i.e. correlated above the threshold) at zero lag may be connected at non-zero time lag. The directed network is formed by using the peak value of the CCC (considered for lags $-15 \leq \tau_c \leq 15$ minutes) between two magnetometers to determine whether there is a network connection between them. A station pair is 'connected' in the network if the peak CCC value at time lag τ_c exceeds the connection threshold for the station pair. A non-zero lag, τ_c , indicates temporal information flow, so that the connection now has an associated direction (indicated by the sign of τ_c) and timescale (magnitude of τ_c) of

propagation/expansion of the observed signal between the two magnetometers. A schematic showing how a magnetometer pairs' magnetic field is converted into a directed connection is included in the SI, figure S34. We use this method to calculate the raw time-varying directed networks for this study."

R2 C19: *Line 164: I have no idea of what the "edge-betweenness" algorithm does. References to the statistical literature are not sufficient for the typical space physics reader. Please give some simple summary of what is done in this process.*

(Previously) Lines 159-166 already held a simple summary of edge betweenness, as well as reference to the code used (Csardi & Nepusz, 2006). This standard code is available in C, Python and R. However we have added this term to the glossary and attempted to simplify further.

Lines 139-148

"The edge betweenness algorithm from the igraph package (Csardi & Nepusz, 2006) ... The edge betweenness community detection algorithm may be summarized as follows: consider a network comprised of two communities (A and B), each containing highly connected nodes, with only one connection (a bridge) between the two communities. If we consider all possible shortest paths needed to travel between the nodes in A and the nodes in B, the edge that connects the communities will always be crossed, therefore the bridge edge will have the highest edge-betweenness. The same logic corresponds to multiple communities; the edges between the communities will carry the majority of shortest paths and therefore have the highest edge betweenness. The edge-betweenness algorithm (Newman & Girvan, 2004) identifies and successively removes these edges which have the highest edge betweenness, leaving behind sub-networks that are the individual communities."

R2 C20: *Section 2.3: The description of event selection is inadequate as I discuss at length above. I am concerned that timing errors in the selection of expansion and recovery onset times has introduced significant errors in the analysis. Also it cannot be determined if all the substorms used are truly isolated.*

As stated above, the selection criteria and the reasons for the chosen substorms were carefully discussed and illustrated in Gjerloev, 2007 and in a long list of subsequent papers. The list of events provided has been amended to clarify which substorms are included in the main text. Gjerloev, 2007, and our methods description clearly states that all substorms are "(i) they are isolated single events optically and magnetically" (line 74) and further that we "require that the nightside is quiet for at least one CCC window (127 minutes) before the substorm onset so that the network calculated at the time of substorm onset is not contaminated with previous activity, rejecting events where the SML index (Newell & Gjerloev, 2011) exceeded ~25% of its maximum value (at the peak of the substorm) in the 127 minutes before the start of the substorm." (lines 80-83)

We only use SML as a guide to how quiet the substorm was before onset (from polar VIS), but as a response to the reviewers figure r2 we here plot the superposed epoch of the SML for the 41

substorms from the main text.

We have further emphasised this with added text:

Lines 71-74

"We study a subset of events drawn from the list of isolated substorms between 1997-2001 established by Gjerloev et al. [2007] and also described in Gjerloev and Hoffman [2014]. They determined the substorm timing (onset and peak expansion) solely from global auroral images rather than magnetometer data or magnetic indices. The substorms were selected such that..."

R2 C21: *Line 274: In Figure 1 there is a proliferation of small communities with few connections until 10 minutes into expansion. This is likely noise as they vanish as the wedge develops. There is a more important issue here. I showed above that the timing used in the study is likely erroneous. The stated onset is 05:41, but my scaling of the SML index shows it is actually 05:50 in the magnetometer data, and the MPB index shows the current wedge did not start until 05:52. Thus the first 10 minutes do not appear to be related to the actual wedge formation.*

The issue of timings is discussed at length above. Polar VIS images (which can be seen in figure 2) clearly show a brightening at 05:41.

R2 C22: *Line 319: There are only 4 dashed lines at 10 min intervals in Figure 1. Figure 2 uses 5-in intervals.*

Vertical lines make panels 3 and 4 make it difficult to see the magnetometers in panel 3 and 4 of figure 1. Figure 2 is clearly labelled with the relevant times. We have edited the line referenced to read:

Line 236-237

"Figure 2 contains eight snapshots of the same substorm, corresponding to the times in Figure 1."

R2 C23: *Figure 1: Can you explain why you used theta for magnetic local time and phi for magnetic latitude? This is the opposite of the usual definitions of spherical coordinates.*

Phi and theta tend to be used interchangeably in spherical coordinates literature and tend to vary per source. In our letter they are clearly defined and mainly used to show the difference between the mean/centroid magnetic coordinates of a community and the individual magnetic coordinates of a magnetometer.

R2 C24: RECOMMENDATION

I believe this manuscript requires major revisions to address the issues I have raised above.

Response: We have made major changes as described above.

R2 C25: *Frankly I don't believe that better timing will change any of the results in a significant way.*

Response: As detailed above, we have not sought to implement 'better timing' as our current methodology provides a robust and consistent method to compare across multiple events. We feel that this critique arises from a misunderstanding of our methodology as discussed above.

R2 C26: *I am not convinced that the technique used really establishes the non-existence of wedgelet signatures.*

Response: As above, we are testing for the existence of large-scale spatial coherence, not the non-existence of small scale wedgelets. We have now included examples of our analysis method applied to some previously identified flow bursts and these can be seen not be clearly distinct from the signature of large-scale spatial coherence. We have determined a 'noise floor' to quantify what would be seen when the signals are large amplitude but spatially incoherent. This is already addressed in text which we have modified as follows:

Lines 176-182:

"The random phase surrogate provides an estimate of the network properties that could arise 'by chance' in a given set of data. For each event, the time-series from each magnetometer are Fourier transformed, the phase spectrum is randomized whilst preserving the amplitude spectrum and this is then inverse Fourier transformed to give a surrogate time-series with the same power spectrum as the original signals, but with no time correlation.... The surrogate time series of each magnetometer is then used to calculate the random phase surrogate of the network parameters using the method as described in section 2.3 and 2.4."

R2 C27: *I am not even convinced that they have studied the current wedge (DP-1) as distinct from DP-2 (global convection). They co-exist during a substorm and are both global and coherent. The paper must first establish that they are studying DP-1.*

Response: We have resolved the spatially extended coherent structure that follows substorm onset (seen in auroral imaging) and is otherwise not present. We have set the cross-correlation threshold for constructing the network to pick-out this signature. Changes in global convection can be picked up by a suitable choice of threshold. We demonstrated this in Dods et al (2017) where we used a similar analysis to study the effect of isolated northward and southward turnings under otherwise quiet conditions. A constant background is excluded from our analysis by background trend subtraction. We only see the large scale current system in direct response to onset and not at other times. Therefore we have established that we are resolving the large scale current response to a substorm, ie DP-1.

R2 C28: *Second they must show what their technique can respond to wedgelet signatures. I am not sure what these are in the SuperMag data. My guess is that wedgelets cause a small isolated negative bay at one or two stations. Typically I would expect these to occur at high latitudes and in only one station. The signature should move equatorward. I can't see how these could appear in the network analysis when two other global current systems are present.*

Response: We have performed our analysis on two previously published flow burst events and this shows that our analysis method can indeed detect them, this is now provided in the SI and described above. However our main result- that a large-scale spatially extended current system invariably follows substorm onset- does not require the detection of wedgelets.

Reviewer #3:

Reviewer: *The paper presents an interesting piece of research whereby isolated substorms were analysed collectively using network science. Magnetic properties of the substorms were used to construct networks and community detection was applied and benchmarked with null models. Results demonstrated that the previously unconnected substorms merged into a large scale SCW.*

Reviewer: *1) Results presented here are very interesting. With regards to the network analysis, two ares would affect the overall results.*

R3 C1: *Firstly, it is dependent on how the networks of constructed. The networks were constructed using readings from over 100 ground-based magnetometers whereby the magnetometers (physical locations) are nodes and a link exists between two locations when the cross-correlation of their vector magnetic field exceeds a pre-defined values - i.e. when two locations exhibit similar magnetic patterns. Given that the networks were conjectured based on pre-defined assumptions, it would be also be useful to say something about what is actually forming the communities. While the results show that a large-scale structure has emerged in the “constructed” network, what does this actually mean in terms of the behaviour of substorms. Magnetic activities from different locations become more similar/correlated/synchronous?*

Response: We have amended the text throughout to clarify what the results tell us in terms of the ionospheric behaviour. We have edited the abstract to clearly state our interpretation:

Abstract

“All 40+ substorms analysed exhibit robust structural change from many small, uncorrelated current systems before substorm onset, to a large spatially-extended coherent system, ~10 minutes after onset. We interpret this as strong indication that the auroral electrojet system during substorm expansions is inherently a large-scale phenomenon and is not solely due to many meso-scale wedgelets.”

We have also included a schematic of the current systems we would expect to correspond to low or high modularity in the SI, and added text to the MT to explain what we interpret modularity to mean in terms of currents:

Lines 166-171

“The change in the modularity as the substorm progresses identifies how these current systems are changing. The modularity will be maximum when spatially localized coherent perturbations cause a group/community of magnetometers to be internally correlated, but not cross-correlated with other groups/communities of magnetometers. If all auroral latitude magnetometers are highly correlated due to a large-scale ionospheric current system overhead, the modularity will be near-zero. A schematic of the current system models we would expect from high our low modularity is contained in the SI, Figure S36.”

Figure 36. A schematic showing our interpretation of high or low modularity. Modularity will be near-zero when all auroral latitude magnetometers are highly correlated due to a large-scale ionospheric current system overhead, e.g.¹⁵. The modularity will be maximum when spatially localized coherent perturbations cause a group/community of magnetometers to be internally correlated, but not cross-correlated with other groups/communities of magnetometers e.g.¹⁶. If current systems were highly correlated and not spatially distinct e.g.¹⁷ they could appear as a single system (low modularity) under our analysis.

R3 C2: *The authors have assigned directions to links in the construction of the networks (page 3 line 65 and ln 401). The original edge-betweenness algorithm (Newman & Girvan, 2004) ignores the direction and treats networks as undirected graphs. Is this also the case here? If so, this needs to be made clear. If the direction is included in the actual analysis, then please clarify why and how this has been taken into account when the algorithm was applied.*

Response: The edge-betweenness algorithm used from the igraphs package does consider direction, which the Newman & Girvan, 2004, describes as a “trivial variation”. However the modularity, as well as all algorithms used in the SI, do treat all networks as undirected graphs. We will explicitly state this with the text:

Lines 139-141

“The edge betweenness algorithm from the igraph package (Csardi & Nepusz, 2006) considers the direction of the edge when dividing into communities. All other algorithms tested treat the network as undirected.”

Line 154:

“The modularity treats all edges as undirected..”

Including lagged corrections in the context of this letter is mainly to capture the full dynamics of the expanding auroral bulge. We have emphasised this with the text:

Lines 118-121

“Importantly, to recover the full dynamics that occurs on multiple timescales, the CCC at all time lags are needed. Orr et al. (2019) pioneered this approach, obtaining the first directed networks for the response to substorms seen in the full SuperMAG data set. Including time lagged correlations means that a magnetometer pair which were not ‘connected’ to the network (i.e. correlated above the threshold) at zero lag may be connected at non-zero time lag.”

R3 C3: *Secondly, for the null comparison, the randomisation is done on the phase spectrum. Could you provide more information why this was adopted? A common way to perform null comparison is to preserve certain property of the network, e.g. degree sequence, and generate a large number of ensemble networks (e.g. 100 or 1000). Also, the empirical networks are benchmarked against 10 randomly generated networks. It is not clear why 10 networks would be sufficient.*

Response: We are generating our networks from physical data which has multiple issues around band-pass filtering, finite size windowing, instrument response and data fidelity. It is well known that Fourier based methods applied to finite length band- pass filtered timeseries can, if applied to coloured noise (a random signal with a non-white spectrum) generate pseudo-periodicities. There are standard methods in signal processing to deal with this, specifically to reconstruct what the analysis would produce for a random signal with the same non-white power spectrum. We have added text to clarify that the random phase surrogate is calculated for the magnetometer vector time series in

Lines 177-182:

“For each event, the time-series from each magnetometer are Fourier transformed, the phase spectrum is randomized whilst preserving the amplitude spectrum and this is then inverse Fourier transformed to give a surrogate time-series with the same power spectrum as the original signals, but with no time correlation.... The surrogate time series of each magnetometer is then used to calculate the random phase surrogate of the network parameters using the method as described in section 2.3 and 2.4.”

We have additionally compared the modularity to a null comparison by preserving the number of edges per number of nodes and calculating 100 random networks per combination (948000 random networks in total). This plot appears in the SI figure S25. We have elaborated on this by adding the following text to the MT:

Lines 184-185

“We also compared the modularity with that of random networks generated according to the Erdős-Rényi model (Erdős & Rényi, 1959), constructed with the same (time dependent) number of nodes, this is shown in Fig S25.”

39 **Figure 25**

40 Figure 25 shows how the modularity scales with the number of connections and the number of nodes. The modularity from a
 41 random network with n nodes and m connections is overplotted with the modularity from the magnetometer network. The
 42 random networks are generated according to the Erdős-Rényi model⁶ and using the igraph package⁷. For each number of nodes,
 43 n , and each number of connections, m , 100 random networks were calculated. The modularity of the community structure was
 44 then calculated for each network and plotted against the number of connections.

Figure 25. Modularity, Q , plotted versus the number of connections, m . Each panel selects times from all 41 events when the network is comprised of a specific number of nodes (normalized time indicated by color). The edge betweenness algorithm has been used for community detection as in main text. The red circles are the modularity obtained from randomly generated networks. The plot shows random networks have a range of modularities (a measure of how separated the communities are) and there is a threshold in Q, m space. The modularity from the networks derived from the substorm events shows more structure (higher Q) compared to the random networks. The modularity from the networks derived from the substorm events explores a broader range of modularity values per number of connections. There is a clear upper bound to the value of Q per m but it is much higher than that of random networks. The patterns of decreasing modularity observed throughout the MT are not simply a result of the increasing numbers of connections.

R3 C4: *In addition, I have a number of suggestions on the manuscript and hope they would help improve the overall readability.*

3) It would be useful to emphasise more on the contribution of the research in various parts of the manuscript. In the abstract, In 15-17, “imply different and conflicting scenarios” - it would be useful to be more precise about the research question/problem being address here and therefore help highlight the novelty of the work. There are also points in various parts of the manuscript that need to be linked to make the argument here more coherent.

See response to **R3 C1**.

R3 C5: *Ln 47 - suggests large-scale structure has its weaknesses and then in Ln 433 recommends a model for correlated large-scale SCW is needed. It seems contradictory?*

We have reworded the line (previously 47) to clarify:

Line 22

“There have been many variations of this classic scenario (Kamide & Kokubun, 1996; Gjerloev & Hoffman, 2014; Sergeev et al., 2011) with most focusing on the large-scale structure.”

R3 C6: *Are the issues being substorms have only been studied as isolated object? ... i.e. in Ln 20 “a series of mesoscale wedgelets”? or been only focusing on the large scale structure (Ln 47). does that mean that this study is the FIRST to examine SCWs and their time varying properties collectively? If this is the case, I don't think it has been made 100% clear in the manuscript. Also, if this is the case, the use of network science is a mean to achieve this goal, suggest to emphasise more on the objective.*

The issue is a long standing debate with many different models for the current systems, often with static diagrams and analysis to pick out modes on particular spatial scales of interest such as those coincident with particular component current systems. This amounts to a formal categorization of features seen in the spatial field, in terms of already established terminology. Our analysis is carefully constructed to avoid any such categorization so that our result is completely general, and does not rely on any such underlying categorization.

We have added text to emphasise why network science is a good tool for this problem:

Lines 43-46

“Network analysis does not introduce spatial correlation, require any a priori assumptions for variation in ground conductivity or formal categorization of features seen in the spatial field, but allows for quantification of spatiotemporal patterns across sets of multiple, spatially distributed observations.”

R3 C7: *3) In the methods section, suggest you move 2.3 and 2.4 to the beginning and introduce your data before describing the methods.*

We have followed the reviewers advice and swapped these sections.

R3 C8: *4) Ln 255 - 271 - this paragraph is repetitive of the caption, suggest the authors to make this more concise or use this space to help readers to interpret panel 1 and panel 2 of figure 1. At present, these subfigures (as well as the rest of the subfigures) contain a lot of information, it would be help to provide further description and explanation to help the readers.*

We have cut down this paragraph to remove repetition.

Line 195-202

“Panels 1 and 2 of Figure 1 visualize the overall importance in the network, and physical location, of the network communities as a function of time. The edge betweenness algorithm does not pre-define how many communities there should be so the number of communities (plotted as circles) at each time is completely unconstrained and changes throughout the substorm. The network has been constructed by identifying connections using the CCC lag at which the CCC between each pair of magnetometers is at its peak so that each connection has an associated lag, τ_c . The magnitude of the lag $|\tau_c|$ indicates whether connections within a given community are formed rapidly (zero CCC lag i.e. $\tau_c=0$ (gray)) or whether they are associated with propagation and/or expansion (non-zero CCC lags from 1-15 minutes (blue-red)) (orr et. al.2019).”

R3 C9: *In Panel 6 (fig 1), it is not so easy to see the results from the surrogate - presumably there should be two lines close to zero there.*

Indeed. This has been added to the figure caption to emphasis:

“Panel 6 plots the normalized number of connections, $\alpha(t')$, both within the nightside and within the SCW, as well as their surrogates (both near zero throughout).”

- J. Dods, S. C. Chapman, J. W. Gjerloev, Characterising the Ionospheric Current Pattern Response to Southward and Northward IMF Turnings with Dynamical SuperMAG Correlation Networks, *J. Geophys. Res.*, 122, doi:10.1002/2016JA023686. (2017)
- Newell, P. T., and J. W. Gjerloev (2011), Evaluation of SuperMAG auroral electrojet indices as indicators of substorms and auroral power, *J. Geophys. Res.*, 116, A12211, doi:10.1029/2011JA016779.
- Gjerloev, J. W., and R. A. Hoffman (2014), The large-scale current system during auroral substorms, *J. Geophys. Res. Space Physics*, 119, 4591–4606, doi:10.1002/2013JA019176.
- Gjerloev, J. W., R. A. Hoffman, J. B. Sigwarth, L. A. Frank, and J. B. H. Baker (2008), Typical auroral substorm: A bifurcated oval, *J. Geophys. Res.*, 113, A03211, doi:10.1029/2007JA012431.
- Gjerloev, J. W., R. A. Hoffman, J. B. Sigwarth, and L. A. Frank (2007), Statistical description of the bulge-type auroral substorm in the far ultraviolet, *J. Geophys. Res.*, 112, A07213, doi:10.1029/2006JA012189.
- Gjerloev, J. W., R. A. Hoffman, M. Friel, L. A. Frank, and J. B. Sigwarth, Substorm behavior of the auroral electrojet indices, *Ann. Geophys.*, 22, 2135-2149, 2004.
- Gallardo-Lacourt, B., Nishimura, Y., Lyons, L. R., Zou, S., Angelopoulos, V., Donovan, E., ... & Nishitani, N. (2014). Coordinated SuperDARN THEMIS ASI observations of mesoscale flow bursts associated with auroral streamers. *Journal of Geophysical Research: Space Physics*, 119(1), 142-150.
- Grocott, A., Yeoman, T. K., Nakamura, R., Cowley, S. W. H., Frey, H. U., Reme, H., & Klecker, B. (2004). Multi-instrument observations of the ionospheric counterpart of a bursty bulk flow in the near-Earth plasma sheet.

REVIEWER COMMENTS

Reviewer #1 (Remarks to the Author):

The authors have performed a large amount of additional work, and I greatly appreciate their detailed responses. The sharpened focus of the paper - is a large scale SCW present? Yes or no? - that is stated explicitly and clearly now, really help. I believe this paper confirms the generally accepted view of how the SCW forms and evolves, but does so with a new technique, and in a way that I think adds to our body of knowledge. On this narrowly focused question, I think it can (and does) provide an answer. As I mentioned in my previous review, bringing in a new technique to space physics is useful, with extensive potential. Applying it to a question that the majority of the field understands I think will help disseminate the concept. Given this, I think the paper is acceptable for publication. I have just a few minor comments (and I don't need to see the paper again).

Figure 26 helpfully shows the LT and Lat of the stations at onset. While it assuages my previous concern about potentially poor coverage, it does highlight a couple of very poor events around pre-midnight. Event 13 (light blue), 14 (dark red), 33 (green), 38 (yellow). The authors should take a look at those to see if they should continue to be included. But otherwise there is good LT coverage. A minor nit - Figure 27 shows the minimum separation between magnetometers, but that isn't really the concern. The concern is minimum coverage around the pre-midnight region, where the substorm initially forms and expands. To make a really simple example, imagine a station at 21, 24, 3 MLT, then clusters at 18 and 6. This satisfies 2 stations within 3 hours MLT, and has stations very close together. But it would be very bad for monitoring the SCW. But the dataset has only 4 events that are questionable (and not as severe as this example).

Figure 36 is a good summary of the results and how modularity differentiates between the models. An immediate question that arises - can the technique resolve between the single McPherron type SCW, or the 2 system Gjerloev/Hoffman picture? It seems like the answer is no, but if the authors can say anything here that would be helpful.

Reviewer #2 (Remarks to the Author):

Referee 2 Second Review

Recommendation

The authors have performed a valiant effort to answer the questions raised by three referees, 37 pages of response! I am convinced that they have correctly performed very complex analysis analysis. I also agree with the conclusion that the substorm current wedge (SCW) is a global current system in the auroral zone. They also appear to have abandoned the idea that the method can distinguish between various wedgelet models for the generation of the current system. Once the global network is established the addition of the perturbations caused by the arrival of a single dipolarizing flux bundle in the outer magnetosphere must have negligible effect on the network. I believe their result was a foregone conclusion. The SCW was originally postulated to explain systematic midlatitude magnetic perturbations across the night side during substorms. In the auroral zone Rostoker and all of his students and colleagues studied the substorm with the assumption that the global SCW system was the cause of their observations. All of the efforts to map the time development of the pattern of ionospheric currents (e.g. AMIE) were based on the assumption of a global pattern, and their maps prove the conjecture.

In my second review I have first reviewed the author's response to my concerns. They have been given labels of the form R2 CN, where N runs from 1 to 28. Below I provide further comments on some of the Responses that are not clear or not fully answered. If a label is missing it is because I have no issue with the author's response. I have also read the new manuscript with its multitude of changes. I have no new suggestions for revisions. I would like the authors to consider by new comments to their response. If they see an easy way to embed an answer in their manuscript it would be good. If they simply answer my new questions in a second response it will be acceptable.

The only change that I insist must be done is to include a table divided in three parts corresponding to their different subsets of events used in different parts of the analysis. This table should include both the times of the onset and end of auroral expansion derived from Polar images. In their response I only find the manner in which they identified subsets of the full list. There is still no list of expansion end times. Without the list of end times it is impossible for anyone to validate their results.

I am still convinced there are issues with the timing of events and their suggestion that there is a 10-minute interval after auroral onset in which small communities of stations coalesce into one large system. In my original Figure r3 I demonstrated that the average difference between my interactive SML onset and the authors' auroral onset list is 6 minutes, with the auroral onsets earlier than the SML onsets. For the MPB list the average difference is 13.5 minutes. These averages are larger than the expected error in interactively selecting the onsets from indices. One may argue the problem is the indices, but as I show just below there was a physical reason for the 10 minute delay in the author's prime example, Figure 1. They originally explained this delay as coalescence of small scale current systems into one large system. I show below it is more likely a pseudo breakup/man onset issue.

I don't believe that these timing issue have caused any significant problems in their network analysis, and slightly different times would not change their main conclusion that a global system forms. Thus with the addition of the timing table and answers to my new questions I recommend publication.

Timing of March 16, 1997 Substorm

One of the major issues I had with the original draft was the question of substorm onset timing. We all know timing is a major problem in substorm research. Some researchers have argued that no substorm is properly timed unless it is done with auroral observations. The authors of this paper apparently accept this view as they say all timing was done with Polar images. This choice is a little surprising as they are studying the behavior of ionospheric currents through magnetic perturbations. The issue arose when I examined the magnetic data for their Figure 1. I showed that the SML onset was 9 minutes later than their auroral onset and the MPB onset was 11 minutes later. The network results summarized in Figure 1 indicate that the globally coherent system started 10 minutes after their auroral onset. Within the resolution of the data this is the same time as the SML and MPB onsets. However, the authors interpreted this 10 minute delay as an interval in which several local current systems merge into the global system. They argued that the midlatitude perturbations do not start until the current wedge opens up enough to produce a significant ground signature. It seems this should also be true for the auroral network as an initially small wedge will have a much localized ionospheric closure and hence will not appear to be a globally coherent current system. The authors responded by removing from the paper the idea that local systems merge into a global system. But the question of the onset delay remains unanswered.

In their response the authors stated that the Polar images are available from the SuperMAG website. My investigation of this discovered that the images have been projected onto polar plots showing station locations and perturbation vectors. I have downloaded four of these images and combined them in Figure 1 presented below. The top row shows the image one minute before the auroral onset time of 05:41 UT that is shown in the right image. It is obvious that there was a weak activation of midnight aurora. The bottom row shows the image at 05:52 UT which is virtually identical to the 05:41 image. In fact all images between 05:41 and 05:52 UT are nearly identical. My conclusion is that the time chosen for the auroral substorm onset is a pseudo breakup as there is no azimuthal or poleward expansion of the original bright spot. The 05:53 image shows a radical change with additional brightening and azimuthal expansion. Subsequent images display more azimuthal and poleward expansion with an obvious westward traveling surge reaching 20:00 local time. The surge brightened at 06:21 and moved further west. Finally at 06:30 UT the western aurora begins to fade. Since I don't have a list of end of expansion times I can't determine when the authors decided the expansion ended. The details of this westward extension are apparent in the SML index plot as an enhancement of the negative bay at about these times.

In the current draft the authors describe this initial 10-minute interval as follows:

"From before onset until approximately 10 normalized minutes after there are 5 small communities. ... These communities spread throughout the nightside at all MLT."

In the figure I see five horizontal lines of gray dots present from the start of the interval to the time of SML and MPB onsets. They are spread across the nightside but there is no evidence of "spreading", nor is there evidence of "merging". It is likely that these communities are created by inadequate removal of the background DP-2 system and their connections disappear once the intense currents of the wedge begin to flow at 05:51 UT.

New questions on old comments

Comment R2 C2: I sked for a physical description of the expected effects of different possible causes of the current wedge and whether they produce different magnetic patterns on the ground that could be distinguished by the methods introduced in this paper. My list includes:

- Instability referred to as current disruption
- Multiple flow channels at different local times arriving at the same time, each channel producing wedgelets that adds up to a big wedge.
- Multiple flow channels at different local times arriving at different times but depositing flux resulting in current wedge
- Multiple flux bundles in time sequence along a single channel depositing magnetic flux that is accumulated to produce the distortion necessary to divert tail current.

Would the methods of this paper find any difference between these four? I don't see how it would. Each of the above produces a large scale current with correlations between distant stations. The conclusion that the current wedge is a large coherent current system was obvious from the start as it was discovered and studied from systematic perturbations over the entire night side.

Comment R2 C4: It seem likely that the current wedge is produced by the flux in at most one or two distinct BBFs(10-15 min duration) each containing multiple bundles of magnetic flux (of 1-2 min duration). The deceleration and deflection of flow bursts distorts the field creating field-aligned current. The flux deposited by these bursts accumulates and drives additional currents through the ionosphere by distortion of pressure and flux tube volume gradients. Even if the flux all arrive at the same time the low pass filter effect of the self-inductance and resistance of the current path dictate a relatively slow rise time. Could the technique distinguish between the arrival of a single large flux bundle and multiple small bundles?

Comment R2 C6: This seems to be a semantic argument. The authors responded with the statement: "Crucially this is not an 'average over two hours". The definition of the biased cross correlation is $R(m) = (1/N * \sum_{n=0:m-1} [x(n+m) * \text{conj}(y(n))]$. This is the average of the number in []. In a running analysis one can fix the 128 sample on a segment of y, and take a 128 sample segment of x shifted by m, and the correlation will be unbiased as all 128 cross products are available. If you use Matlab "xcorr" and use the same interval of x and y then normalize with $1/(N - |m|)$ to compensate for missing cross products in the sum. It is the normalized average cross product.

In the author's example they appear to have assigned the correlation to the time of the leading edge of the window. Assuming all series in the ensemble have an identical offsets at the same time, then the running cross correlation will begin to respond when the leading edge of the window hits the offset. The correlation will grow until the offset is entirely inside the window. Once the offset is completely inside the window the correlation should be constant apart from fluctuations caused by background noise. The correlation dies away as the trailing edge of the window passes beyond the offset. Thus the interval of response has a duration equal to the sum of the width of the window and twice the width of the disturbance. Since substorms generally lasts longer than 128 minutes the

correlation between two stations will never vanish unless it is completely isolated with more than 2 hours of quiet before and after the event.

I do not completely understand the author's example shown in Figure B1 of response. It seems that their signal occupies 5 ample points (they do not specify the sample rate). Also the duration of the response is about 50 minutes, not 128 minutes. Was a shorter window used in this example?

Figure 28 is a better simulation of the analysis and it is good they have added it to supplementary information. However, there is no SI file in the materials made available for review.

This simulation assumes all stations simultaneously experience a rapid (4 minutes) and large (300 or 400 nT) decrease in all three components of the magnetic field. Given this assumption they make the case that the average canonical correlation for 12 stations responds completely in 5 minutes. This simulation is unrealistic in terms of what must be happening in the real analysis. In general the wave forms of the D and Z differ very much from the H component and are dependent on the location of the two stations relative to the center of the current wedge in local time and latitude. The canonical correlation procedure mixes the three components at the paired stations in a manner difficult to imagine. It would be better to show happens when two stations in different locations relative to center are paired and correlated. I would like to see the combined wave forms that are correlated. See this paper for electrojet perturbation patterns.

Kisabeth, J.L., and G. Rostoker (1977), Modeling Of 3-Dimensional Current Systems Associated With Magnetospheric Substorms, *Geophysical Journal of the Royal Astronomical Society*, 49(3), 655-683. doi:10.1111/j.1365-246X.1977.tb01310.x

Comment R2 C8: My concern is how the author's method distinguishes between two distinct modes of high latitude response DP-1 and DP-2. The author's say the following: "A first step in our analysis it to remove the slow trend in the data on the timescale of the window which removes any constant background- as is essential for any Fourier based analysis" This basically means they have "high pass filtered" the observations to select rapid variations of the substorm expansion. I don't remember this statement in the original draft and I can't find it in the current draft, but I may have missed it. It needs to be emphasized as it removes many of my objections.

The authors continue with "Furthermore we do not claim to 'prove anything about how the current wedge is formed" I though the objective was to determine whether individual wedgelets played a role in its formation. They say they have "Our analysis is carefully constructed to avoid any such categorization". However, DP-1 is simply another name for substorm current wedge which is the response mode they have chosen to study.

Comment R2 C9: I find later in the response three lists of onset times for the events used in this study. Their response did not include the times selected as the end of the expansion. Perhaps this list is in the Supporting Information that is not available at the website.

This comment identified a major concern, in particular why the authors did not specify which 41 events in the list of 76 events they used in their analysis? They have not provided this essential information. Furthermore, at the end of the following paragraph in my first review I said "The

authors do not describe their timing procedure or list the times determined in the scaling of their time series.” Their response begins with the following:

Lines 71-74 “We study a subset of events drawn from the list of isolated substorms between 1997-2001 established by Gjerloev et al. [2007] and also described in Gjerloev and Hoffman [2014]. They determined the substorm timing (onset and peak expansion) solely from global auroral images rather than magnetometer data or magnetic indices.

I agree that the two papers mentioned do a good job of describing what was done to select events. Gjerloev et al. [2007] mention 116 substorms used in the analysis of the early paper. The Gjerloev and Hoffman [2014] paper also mentions a list of 116 substorms. However, neither of these papers provides a list of expansion or recovery onsets. Presumably the 76 events in the supporting information submitted with the first draft of this paper corresponds to some of the onsets in the original Gjerloev work.

The authors have not provided a list of times of the ends of the auroral expansion - They need to explain what additional constraints were applied to the original 116 events to reduce it to 76, and then why only 41 of these were used in this paper. As I point out in my first review, at least half of the 76 events are not truly isolated or single onset events as they were supposed to be.

In their extensive response to my concern they mention in item C) Substorm normalization:

“Figure r4 is based on an amplitude and onset timing analysis. It unfortunately does not include an important normalization of the duration of the expansion phase.”

My emphasis in bold is because the information necessary to do this normalization has not been provided by the authors in either draft, nor in the earlier papers. Without this information it is impossible for a referee to discuss the issue of substorm timing, or determine whether they have properly scaled the individual events before performing their analysis.

The authors have responded to this comment with a lengthy discussion of substorm timing. They argue that analysis of auroral images is the only good way to time substorm expansions. They provide a satellite image to counter the argument I made with my Figure r1. My reaction to this specific image is “good luck”. I am very dubious of timing derived from this image with such bad viewing geometry and such grainy display. [However, the projections of these images on polar plots are much better and allow one to time gross auroral features on a one minute scale.] Perhaps other substorms have better image sequences, but I don’t know. My question is why reject the magnetic timing of these events when you are studying the development of a current system? The authors then point out that the standard 12-station AL index has too few stations to do accurate timing, but in my review I used the SML index with over 100 stations. Finally the authors reject the use of the MPB index because: “Using the mid-latitude bay index has very limited usage and thus it has not undergone any critical scrutiny (as far as we know)”. The MPB index is simply an implementation of a long-standing procedure for determining the midlatitude positive bay onset that has been used since the beginning of substorm research. In fact it was first used in 1974 in a program called the “midlatitude bay finder”. See the Appendix to the following paper.

Caan, M.N., R.L. McPherron, and C.T. Russell (1978), The statistical magnetic signature of magnetospheric substorms, *Planetary and Space Science*, 26(3), 269-79.

I also note that Table 1 in the following paper compares the MPB onset times to both IMAGE, and Polar satellite image onsets. The peak of the time delay between them and MPB for IMAGE is – 1 min, and for Polar it is -3 min. The SML onset (71,056 events) is peaked -4 min ahead of MPB onsets. The conclusion is that on average the SML and MPB onsets are close to satellite image onsets.

McPherron, R.L., and X. Chu (2018), The Midlatitude Positive Bay Index and the Statistics of Substorm Occurrence, *Journal of Geophysical Research: Space Physics*, 123(4), 2831-2850. doi:10.1002/2017JA024766

The authors suggest that the MPB onset is delayed relative to other onsets because of time to pen the wedge enough to create a significant midlatitude perturbation. I agree this is possible, but why doesn't this apply to the global network? At first a narrow wedge will only cause nearby stations to be correlated. As the wedge grows in size more stations enter the network of correlated stations.

Comment R2 C12:

My concern was about the separation of DP-2 (global convection) and DP-1 (current wedge). I must have missed the discussion of detrending the data so that time variations longer than window length were removed from the data. I still can't find this statement in the current draft.

The authors mention a study by Gjerloev et al. [2010] that concludes the DP-2 current system during quiet times does not exist on the night side. They argue the whole idea of a global DP-2 current system has been changed by this work. In the paper they mention 73 substorms selected for a step function change from constant north Bz to constant south Bz. They were timed by Polar auroral images. This set of substorms is very unique because the selection criteria virtually guarantee that there is no nightside electrical conductivity until the expansion onset. Thus the application of a convection electric field drives no current. These probably differ from more typical substorms in which electrons injected by previous substorms circle the Earth and precipitate causing conductivity in the oval.

Comment R2 C15:

This modified substorm onset list is part of what I needed to see. Note there is no new Supplementary Information file at the Journal Website. Where is the list giving the end of expansion times? The authors say "the timings are determined directly from Polar VIS images (freely available on SuperMAG)."

The SuperMAG web site does not list Polar VIS Images as an available product. I don't want the original data, I want to see the list someone created from the images. I don't want to repeat this laborious project.

Data Products (from website)

Line Plots and Data Download

Polar Plots

Magnetic Indices

Substorm Event List

Movies

Inventory

The polar plots have overlays of VIS images. It is not obvious that one can get just the images. (Finally managed this by turning off many options.) The substorm onset list at the website only gives the times of negative bay onset determined from SML index, and does not include the end of expansion. These are not the VIS times needed for proper evaluation.

Comment R2 C16: Does your glossary contain explanations of the correlation procedure? The SI file is not available at the journal website. I am curious how the CCA combines the components of the field when stations are located in distinctly different parts of the perturbation pattern.

Comment R2 C17: The CCA must grow slowly as the leading edge of the moving window passes over the sudden onset of the expansion. It must take as long as the duration of the expansion to reach a maximum value. Is this true?

Comment R2 C20: Where is the list of the end of expansion times?

Comment R2 C21: This question was concerned with the explanation of the 10 minute delay, as well as the timing issue. You pointed out that that the MPB index may be delayed by opening of the current wedge sufficiently that the ground perturbation exceeds the noise. Why is this not true for the CCA in the auroral zone? As the moving window first encounters the station perturbations the signal occupies only one sample out of 128. You have thresholds for acceptable signal strength and do not establish a connection until this happens.

Comment R2 C27: The authors seem to use the word substorm to refer only to the current wedge. "...large scale current response to a substorm, i.e. DP-1. R2". The growth phase preceding the expansion phase is part of the substorm. You have said in several places in the response that you detrended the data so that variations on the time scale of DP-2 are removed from the data. DP-2 is the manifestation of the changes in magnetospheric configuration that makes the expansion phase possible so this is also part of the substorm. Please add this information to the manuscript.

Reviewer #4 (Remarks to the Author):

This revised version of the network-based study of magnetospheric substorms is very clear and indeed exciting. Therefore, I recommend acceptance of this paper now.

Reviewer #5 (Remarks to the Author):

This paper's results are definitely of publishable quality; I have only a few comments, mostly on methodologies rather than the conclusions on geomagnetic substorms. As at places, the paper appears to be over-hyping the tools without stating the pitfalls.

Authors need to motivate why community detection is a better algorithm than, say, an EOF analysis. Please see the paper: <https://arxiv.org/abs/1305.6634>. Please discuss the drawbacks of community detection, artifacts, and caveats while analyzing spatiotemporal data.

These community structures are depended on the thresholds, i.e., connection threshold for the station pairs. Have authors made some tests on networks obtained without making subjective choices on thresholds, for example, using statistical significance for deciding on a link's existence? How would these results change if one uses a nonlinear metric for correlation, for example, mutual information? Similarly, several metrics are available that explicitly calculate information flow between grids, for example, transfer entropy. Generally, when directed networks are inferred from data, one uses a more sophisticated metric for information flow and not just time delays. I am raising this issue because these results should not be just artifacts of metrics and thresholds employed.

The glossary of network terminology and diagrams is useful, but this paper needs a simple diagram/illustrating explaining how exactly these networks were constructed.

The comparison with Erdos-Renyi is technically flawed. The comparison should be with configuration models, random networks with the same degree distribution as the empirical networks.

A critical comment about the visualization in this paper: apart from figure captions being incomplete and doing a poor job explaining the figures and their significance, the visualizations are incredibly

complex (lines of different types, colors, markers, left-right axis, multiple panels, and more). Please consider simplifying some of the figures and rewriting captions. A test that authors can use is to write captions that are standalone; that is, a reader can understand most of the figure without going into text back and forth, looking for details of each variable and acronym.

Are the authors claiming that the area that falls, say, between a community formed by three nodes (a triangle) is somehow an area of spatially coherent activity? I am not able to understand these shapes in Fig. 2 (and other similar figures). From community structure how do you estimate the shape of region with coherent activity?

Minor points:

In the paper, the figures from supplementary material are referred to as S1-S35; however, there is no "S" in figure numbering in the supplementary material.

REVIEWER COMMENTS

Reviewer #1:

Reviewer: The authors have performed a large amount of additional work, and I greatly appreciate their detailed responses. The sharpened focus of the paper - is a large scale SCW present? Yes or no? - that is stated explicitly and clearly now, really help. I believe this paper confirms the generally accepted view of how the SCW forms and evolves, but does so with a new technique, and in a way that I think adds to our body of knowledge. On this narrowly focused question, I think it can (and does) provide an answer. As I mentioned in my previous review, bringing in a new technique to space physics is useful, with extensive potential. Applying it to a question that the majority of the field understands I think will help disseminate the concept. Given this, **I think the paper is acceptable for publication**. I have just a few minor comments (and I don't need to see the paper again).

Response: We are grateful to the Reviewer for their careful reading of our manuscript and we have addressed all of these minor points with modifications to the manuscript as detailed below.

Reviewer: Figure 26 helpfully shows the LT and Lat of the stations at onset. While it assuages my previous concern about potentially poor coverage, it does highlight a couple of very poor events around pre-midnight. Event 13 (light blue), 14 (dark red), 33 (green), 38 (yellow). The authors should take a look at those to see if they should continue to be included. But otherwise there is good LT coverage. A minor nit - Figure 27 shows the minimum separation between magnetometers, but that isn't really the concern. The concern is minimum coverage around the pre-midnight region, where the substorm initially forms and expands. To make a really simple example, imagine a station at 21, 24, 3 MLT, then clusters at 18 and 6. This satisfies 2 stations within 3 hours MLT, and has stations very close together. But it would be very bad for monitoring the SCW. But the dataset has only 4 events that are questionable (and not as severe as this example).

Response: We thank the reviewer for their positive comments and raising their concerns about these events. We have double checked each of these events and all four do show the change from high modularity to low modularity during the expansion phase. When dealing with real data coverage is always a challenge but for a statistical study it is important to pre-define a criteria for inclusion of each event into the statistical sample, as hand picking events would immediately create a bias. It is encouraging that even the minimum criteria highlighted by the Reviewer of 2 stations within 3 hours MLT still captures the changing dynamics described in this paper. We have added text to the methods section to emphasis this point:

Lines 92-95

"Figure 26 shows that the vast majority of the chosen substorms exceed this minimum criteria for spatial distribution and have good local time coverage. We find the inclusion of substorms with lower coverage does not alter our result and their inclusion avoids bias that could arise by hand picking events."

Reviewer: Figure 36 is a good summary of the results and how modularity differentiates between the models. An immediate question that arises - can the technique resolve between the single McPherron type SCW, or the 2 system Gjerloev/Hoffman picture? It seems like the answer is no, but if the authors can say anything here that would be helpful.

Response: We agree with the Reviewer; this is a fundamental issue that generally arises in interpreting the network results. Where ground magnetic perturbations are seen over a wide area, the network

modularity can distinguish between (i) multiple smaller regions with spatially correlated structure within each region, and weak or no correlation between regions and (ii) strong correlation over a single, spatially extended area. However, there is always ambiguity between (ii) and the situation where we have (iii) spatially overlapping correlated structures. Where region A is correlated with region B, and region B is correlated with region C, the network cannot distinguish this from the case where A is directly correlated with C (as both are correlated with B).

These SCW models differ in the spatial distribution of the electrojet currents but not necessarily in their temporal evolution. The electrojet solutions primarily differ in is the midnight region where, unfortunately, the ground dB signature of the electrojets is very similar (at least for the classical McPherron and the Gjerloev/Hoffman models) as shown in the Gjerloev and Hoffman paper (figure 6).

The problem this paper is trying to solve is, however, different. Wedgelets and SCW differ significantly in both lifetime (some 10 min vs ~30 min) and longitudinal scale size (1h MLT vs ~6h MLT). The technique used here is well suited for differentiating wedgelet from a large-scale electrojet.

We have reworded text in the conclusion to clarify:

Lines 323-326

"We cannot rule out the scenario in which there are two or more individually correlated spatial structures that result in perturbations on the ground that are spatially overlapping. A two component system where region A is correlated with region B, and region B is correlated with region C, cannot be distinguished from a single component system where A is directly correlated with C (as both are correlated with B)."

Reviewer #2

Recommendation

The authors have performed a valiant effort to answer the questions raised by three referees, 37 pages of response! I am convinced that they have correctly performed very complex analysis. I also agree with the conclusion that the substorm current wedge (SCW) is a global current system in the auroral zone.

Response: We appreciate the kind words from the Reviewer. That said the response was in line with the likewise careful and comprehensive report by Reviewer.

Reviewer: They also appear to have abandoned the idea that the method can distinguish between various wedgelet models for the generation of the current system.

Response: We appreciate that our clarification is well received.

Reviewer: Once the global network is established the addition of the perturbations caused by the arrival of a single dipolarizing flux bundle in the outer magnetosphere must have negligible effect on the network. I believe their result was a foregone conclusion. The SCW was originally postulated to explain systematic midlatitude magnetic perturbations across the night side during substorms. In the auroral zone Rostoker and all of his students and colleagues studied the substorm with the assumption that the global SCW system was the cause of their observations. All of the efforts to map the time development of the pattern of ionospheric currents (e.g. AMIE) were based on the assumption of a global pattern, and their maps prove the conjecture.

Response: The wedgelet scenario has gained a lot of popularity in the community. It is appealing as it points to a fundamental property of magnetospheric convection in that BBFs (or “bubbles”) are widely considered as a means of resolving the so-called pressure balance inconsistency in the magnetotail [Pontius & Wolf 1990]. Global MHD provides a causal relationship between injections or flows and ionospheric currents. Thus, our paper does indeed address contrasting views which are being widely discussed. As such the paper is written in response to the recent promotion of the wedgelet scenario. One of the authors (Gjerloev) had the same expectations regarding the outcome as the referee. That said it never hurts to check and the dataset and technique does provide a powerful tool to test this scenario.

Reviewer: In my second review I have first reviewed the author’s response to my concerns. They have been given labels of the form R2 CN, where N runs from 1 to 28. Below I provide further comments on some of the Responses that are not clear or not fully answered. If a label is missing it is because I have no issue with the author’s response. I have also read the new manuscript with its multitude of changes. **I have no new suggestions for revisions.** I would like the authors to consider by new comments to their response. If they see an easy way to embed an answer in their manuscript it would be good. If they simply answer my new questions in a second response it will be acceptable.

Response: We have addressed all of these comments as detailed point-by-point below.

Reviewer: The only change that I **insist** must be done is to include a table divided in three parts corresponding to their different subsets of events used in different parts of the analysis. This table should include both the times of the onset and end of auroral expansion derived from Polar images. In their response I only find the manner in which they identified subsets of the full list. There is still no list of expansion end times. Without the list of end times it is impossible for anyone to validate their results.

Response: We have included these timings as requested.

Reviewer: I am still convinced there are issues with the timing of events and their suggestion that there is a 10-minute interval after auroral onset in which small communities of stations coalesce into one large system. In my original Figure r3 I demonstrated that the average difference between my interactive SML onset and the authors’ auroral onset list is 6 minutes, with the auroral onsets earlier than the SML onsets. For the MPB list the average difference is 13.5 minutes. These averages are larger than the expected error in interactively selecting the onsets from indices.

Response: It seems logical that the visible onset precedes the SML onset which precedes the AL onset which again precedes the MPB onset. The logic was outlined in our previous response and illustrated by the March 16, 1997 event and sequence of Polar VIS images (both Earth camera and low res camera were shown). The ‘expected errors’ are really not ‘errors’ but merely delays that are a result of probabilities of observing the signal (emissions or dB). MPB as mentioned has the embedded issue of monitoring the currents at large distances which implies a delay. Estimating precisely what these delays should be requires a careful statistical study.

The substorm list used in this study was developed by Gjerloev and Hoffman. They identified events and onsets independently, then cross checked and discussed. With this new acquired knowledge they redid the identification and through discussion consolidated the two new lists. It was a comprehensive and highly time consuming exercise. Visual inspection identified the peak of the substorm and by back-tracking in time the onset was identified. This avoids pseudo-onsets.

Reviewer: One may argue the problem is the indices, but as I show just below there was a physical reason for the 10 minute delay in the author's prime example, Figure 1. They originally explained this delay as coalescence of small scale current systems into one large system. I show below it is more likely a pseudo breakup/man onset issue.

Response: In our previous 37 page response we showed all the images available for the event. There is a gap in imaging from 5:40 UT to 5:50 UT. The 5:40 Earth Camera image shows a spot while the 5:50 Low Res Camera image show considerable expansion which likely is why we selected the 05:40 as the onset. Note that on SuperMAG the Low Res Camera images are not include as they did not have any means for geo-location. In cases like these Gjerloev and Hoffman writes:

"Since our event selection is biased around the winter solstice when the Earth camera was sharing the VIS telemetry allocation with the visible imagers, the temporal resolution was 1–5 minutes. To obtain higher time resolution data especially for the onset time, the Earth camera data were supplemented with images from the visible imaging cameras and the 1-minute image data were supplemented with images from the Ultraviolet imager (UVI) on Polar [Torr et al., 1995]. These additional data also enabled the elimination of pseudo-onsets and the determination that the substorm developed continuously out of the identified onset location."

Checking our onset list the notes stated that UVI was used and the onset is classified as 'excellent'.

We have added a line of text to the methods section to clarify the method for onset timings.

Lines 73-75

"Earth camera data was supplemented with additional data (visible imaging cameras and Ultraviolet imager (UVI) (Torr et al. 1995) to eliminate pseudo-onset."

Reviewer: I don't believe that these timing issue have caused any significant problems in their network analysis, and slightly different times would not change their main conclusion that a global system forms. Thus with the addition of the timing table and answers to my new questions **I recommend publication.**

Response: We are grateful to the Reviewer for their careful reading and thorough consideration of our manuscript.

Reviewer: Timing of March 16, 1997 Substorm

One of the major issues I had with the original draft was the question of substorm onset timing. We all know timing is a major problem in substorm research. Some researchers have argued that no substorm is properly timed unless it is done with auroral observations.

Response: Indeed, we maintain that the imaging is the most objective and clear means for identifying the onset. The reason is partially that imaging provides 2D coverage while for example ground magnetometers do not have global coverage and must wait for the current system to expand over a ground station. This latter point is why SML onsets will precede AL onsets. What is 'proper onset timing' depend on the study. For many studies the use of ground magnetometers will have sufficiently precise timing. We do recommend the study:

Gjerloev, J. W., R. A. Hoffman, J. B. Sigwarth, and L. A. Frank (2007), Statistical description of the bulge-type auroral substorm in the far ultraviolet, *J. Geophys. Res.*, 112, A07213, doi:10.1029/2006JA012189. Please see their Figure 3 which relates SML/SMU (axis says AL/AU as this paper was written before these indices were named but it is indeed SML/SMU) and the 116 substorms. A more careful study of the timing using the various techniques may indeed be informative and would be the topic of another paper; as the Reviewer points out 'I don't believe that these timing issue would not change their main conclusion'.

Reviewer: The authors of this paper apparently accept this view as they say all timing was done with Polar images. This choice is a little surprising as they are studying the behavior of ionospheric currents through magnetic perturbations. The issue arose when I examined the magnetic data for their Figure 1. I showed that the SML onset was 9 minutes later than their auroral onset and the MPB onset was 11 minutes later. The network results summarized in Figure 1 indicate that the globally coherent system started 10 minutes after their auroral onset. Within the resolution of the data this is the same time as the SML and MPB onsets. However, the authors interpreted this 10 minute delay as an interval in which several local current systems merge into the global system. They argued that the midlatitude perturbations do not start until the current wedge opens up enough to produce a significant ground signature. It seems this should also be true for the auroral network as an initially small wedge will have a much localized ionospheric closures and hence will not appear to be a globally coherent current system. The authors responded by removing from the paper the idea that local systems merge into a global system. But the question of the onset delay remains unanswered.

Response: As discussed above we maintain that global auroral imaging is the most objective and clear means for identifying the onset. We have carefully checked the timings of the event in question. We would also add that the event highlighted in figure 1 is an example which clearly illustrates our method. The main conclusion of our paper is not concerned with the details of individual events but is a statistical study of the common behavior of many events. We clarify this point with text added to the results section:

Line 200

"We present this example event as an illustration of the methodology that we will use to compare multiple events."

Reviewer: In their response the authors stated that the Polar images are available from the SuperMAG website.

Response: As discussed above, only the Polar Vis Earth Camera images are available on SuperMAG not the Polar Vis Low Res Camera. Polar VIS is now explicitly mentioned in the acknowledgements section.

Reviewer: My investigation of this discovered that the images have been projected onto polar plots showing station locations and perturbation vectors. I have downloaded four of these images and combined them in Figure 1 presented below. The top row shows the image one minute before the auroral onset time of 05:41 UT that is shown in the right image. It is obvious that there was a weak activation of midnight aurora. The bottom row shows the image at 05:52 UY which is virtually identical to the 05:41 image. In fact all images between 05:41 and 05:52 UT are nearly identical. My conclusion is that the time chosen for the auroral substorm onset is a pseudo breakup as there is no azimuthal or poleward expansion of the original bright spot. The 05:53 image shows a radical change with additional brightening and azimuthal expansion.

Response: The reason for this is simply that the Earth Camera did not provide any images in the 5:41-5:52 time range. The Low Res Camera had images 5:50 and 5:51 UT as mentioned above and UVI provided images throughout.

Reviewer: Subsequent images display more azimuthal and poleward expansion with an obvious westward traveling surge reaching 20:00 local time. The surge brightened at 06:21 and moved further west. Finally at 06:30 UT the western aurora begins to fade. Since I don't have a list of end of expansion times I can't determine when the authors decided the expansion ended. The details of this westward extension are apparent in the SML index plot as an enhancement of the negative bay at about these times.

Response: Gjerloev and Hoffman identified onset (location and timing) and peak (boundaries and timing). Timing information is used in the study and will be made available with the revised manuscript.

Reviewer: In the current draft the authors describe this initial 10-minute interval as follows: "From before onset until approximately 10 normalized minutes after there are 5 small communities. ... These communities spread throughout the nightside at all MLT." In the figure I see five horizontal lines of gray dots present from the start of the interval to the time of SML and MPB onsets. They are spread across the nightside but there is no evidence of "spreading", nor is there evidence of "merging". It is likely that these communities are created by inadequate removal of the background DP-2 system and their connections disappear once the intense currents of the wedge begin to flow at 05:51 UT.

Response: We have modified the line referenced by the reviewer to now read: "*These communities are spread spatially throughout the nightside at all MLT.*" Figure 2 provides evidence of the change in structure of coherent signals after onset at 05:41 UT. The removal of the DP-2 system will be discussed extensively in comments below.

Reviewer: New questions on old comments

Comment R2 C2: I asked for a physical description of the expected effects of different possible causes of the current wedge and whether they produce different magnetic patterns on the ground that could be distinguished by the methods introduced in this paper. My list includes:

- Instability referred to as current disruption

Response: We are not aware of any study predicting what the ground dB signature would be of current disruption.

Reviewer: • Multiple flow channels at different local times arriving at the same time, each channel producing wedgelets that adds up to a big wedge.

Response: As discussed in our manuscript, multiple flow channels producing wedgelets that are uncorrelated with each other will not 'add up to a single big wedge' under our network analysis. Our analysis tests for spatial coherence within and between structures. Global MHD simulations show dB signatures of BBF driven wedgelets. Assuming a number of identical wedgelets are aligned in local time this could be used to directly test this scenario. Data analysis, however, has shown that BBFs are sporadic in nature and this scenario seem exceedingly unlikely.

Reviewer: • Multiple flow channels at different local times arriving at different times but depositing flux resulting in current wedge

Response: If the deposited flux forms a single spatially coherent structure then our network analysis will identify it as a single SCW. If on the other hand the deposited flux remains contained within many distinct uncorrelated structures then our network analysis will identify it as such.

Reviewer: • Multiple flux bundles in time sequence along a single channel depositing magnetic flux that is accumulated to produce the distortion necessary to divert tail current.

Response: We are not aware of any studies that can provide the observational constraints required. What dB signature would this actually produce? Again, the key discriminator is whether the accumulated flux results in a single spatially coherent structure or not.

Reviewer: Would the methods of this paper find any difference between these four? I don't see how it would. Each of the above produces a large scale current with correlations between distant stations. The conclusion that the current wedge is a large coherent current system was obvious from the start as it was discovered and studied from systematic perturbations over the entire night side.

Response: As discussed above and shown in our schematic in figure 36 we are discriminating two well-defined scenarios where (i) the SCW is a large-scale spatially correlated system and (ii) where there is no large-scale spatially correlated current, instead the system is comprised of spatially distinct, uncorrelated wedgelets.

If the individual flow channels and flux bundles are not highly correlated with each other they will not appear as one big wedge under our analysis. Our method is based on correlation, not summation. We have endeavored to make this clearer in the introduction with the text:

Lines 54-56

"Since our methodology quantifies cross-correlation between spatially separated magnetometers, this transition is to a coherent large-scale spatially extended structure, rather than solely a flux accumulation of small-scale wedgelets or flow bursts that are not coherent with each other."

Reviewer: Comment R2 C4: It seems likely that the current wedge is produced by the flux in at most one or two distinct BBFs (10-15 min duration) each containing multiple bundles of magnetic flux (of 1-2 min duration). The deceleration and deflection of flow bursts distorts the field creating field-aligned current. The flux deposited by these bursts accumulates and drives additional currents through the ionosphere by distortion of pressure and flux tube volume gradients. Even if the flux all arrive at the same time the low pass filter effect of the self-inductance and resistance of the current path dictate a relatively slow rise time. Could the technique distinguish between the arrival of a single large flux bundle and multiple small bundles?

Response: Yes, if the single large flux bundle is in fact a single, spatially correlated structure, our network analysis will discriminate it from multiple small flux bundles that are uncorrelated with each other. However there is a definition concern here: What is a 'large' and a 'small' flux bundle and how would we know which is the case? The auroral signature of BBFs (streamers or north-south structures) have local time widths of ~1 h MLT which does not seem to fit the width of the SCW. We wish to point

out to the referee that the above described scenario is far from accepted. It implies several inconsistencies one being the fact that the total integrated energy provided by BBF's vs. the total deposited energy in the ionosphere differ by a factor of 100 or so. That said we again wish to point out that the current paper is not attempting to test various magnetospheric causes for the ionospheric electrojet system. We do not have magnetospheric observations to constrain any speculations – as appealing as they may appear.

Reviewer: Comment R2 C6: This seems to be a semantic argument. The authors responded with the statement: "Crucially this is not an 'average over two hours". The definition of the biased cross correlation is $R(m) = (1/N * \sum_{n=0:m-1} [x(n+m) * conj(y(n))]$. This is the average of the number in []. In a running analysis one can fix the 128 sample on a segment of y, and take a 128 sample segment of x shifted by m, and the correlation will be unbiased as all 128 cross products are available. If you use Matlab "xcorr" and use the same interval of x and y then normalize with $1/(N - |m|)$ to compensate for missing cross products in the sum. It is the normalized average cross product.

Response: Indeed this may be an issue of semantics as we agree that the cross correlation is not an average because it is based on the (normalized) product of signal x with time lagged signal y. We have verified our results by considering different thresholds as detailed in the manuscript and supporting information (figures 23-24) and with a simple example substorm in the SI (figure 28). Lines 119-124 discuss the window length with reference to figure 28 and dods et al 2015, 2017.

Lines 119-124

"The 128 minute window is chosen to give sufficient accuracy in the computed cross-correlation function, whilst still capturing the large-scale spatio-temporal current system behavior. Dods et al. 2017 previously demonstrated, by use of model time series, that this window length resolves changes on much shorter timescales than that of the window length, specifically capturing a sharp ramp in activity, such as when the SCW forms. Figure 28 in the SI demonstrates likewise, using a 128 minute CCC window. In dods et al 2015, the technique was applied using zero-lag CCC and trialled to obtain the un-directed network for a small set of isolated substorm events. This was sufficient to capture the initial spatially coherent response at onset."

Reviewer: In the author's example they appear to have assigned the correlation to the time of the leading edge of the window. Assuming all series in the ensemble have an identical offsets at the same time, then the running cross correlation will begin to respond when the leading edge of the window hits the offset. The correlation will grow until the offset is entirely inside the window. Once the offset is completely inside the window the correlation should be constant apart from fluctuations caused by background noise. The correlation dies away as the trailing edge of the window passes beyond the offset. Thus the interval of response has a duration equal to the sum of the width of the window and twice the width of the disturbance. Since substorms generally lasts longer than 128 minutes the correlation between two stations will never vanish unless it is completely isolated with more than 2 hours of quiet before and after the event.

Response: Yes, the referee is correct, time refers to the window leading edge as our study is focused on the onset of the event. We do not attempt to analyze more complex events with multiple onsets or the recovery phase of substorms. We have used isolated substorms which are particularly quiet before onset (exact definition is given in the manuscript) and only considered the expansion phases of the substorm.

Reviewer: I do not completely understand the author's example shown in Figure B1 of response. It seems that their signal occupies 5 sample points (they do not specify the sample rate). Also the duration of the response is about 50 minutes, not 128 minutes. Was a shorter window used in this example?

Response: Figure B1 is a published figure (in Dods et al 2017) to illustrate the cross correlation for a simple test model (a tanh function plus noise) and in that paper we used a 48 min window, hence the ~50 min response noted by the Reviewer.

Reviewer: Figure 28 is a better simulation of the analysis and it is good they have added it to supplementary information. However, there is no SI file in the materials made available for review. This simulation assumes all stations simultaneously experience a rapid (4 minutes) and large (300 or 400 nT) decrease in all three components of the magnetic field. Given this assumption they make the case that the average canonical correlation for 12 stations responds completely in 5 minutes. This simulation is unrealistic in terms of what must be happening in the real analysis. In general, the wave forms of the D and Z differ very much from the H component and are dependent on the location of the two stations relative to the center of the current wedge in local time and latitude. The canonical correlation procedure mixes the three components at the paired stations in a manner difficult to imagine. It would be better to show happens when two stations in different locations relative to center are paired and correlated. I would like to see the combined wave forms that are correlated. See this paper for electrojet perturbation patterns.

Kisabeth, J.L., and G. Rostoker (1977), Modeling Of 3-Dimensional Current Systems Associated With Magnetospheric Substorms, *Geophysical Journal of the Royal Astronomical Society*, 49(3), 655-683. doi:10.1111/j.1365-246X.1977.tb01310.x

Response: It is unfortunate that our revised SI was not made available to the Reviewer. We have modified figure 28 from the SI to address the points made here by R2 and to address concerns about the DP-1 and DP-2 currents. A model that is simple enough to be illustrative will not encompass all the detail mentioned by the Reviewer (D & Z differing w/r to the relative location of the current system) so we also include figure 34 which shows a real example of two magnetometers. We have expanded this schematic to include the detrended and 'rotated' vectors- i.e. the linear combination of the vector time series that are maximally correlated.

Figure 28. Canonical cross correlation for a test set of twelve modelled vector time series. The top panel shows three example time series of the modelled data. DP1 is represented by a sharp exponential increase, and a slower exponential decrease (equation 1). DP2 is represented by a slowly (500 time units) varying curve (equation 2). The second panel shows all three components of the combined DP1 and DP2 currents with noise and scaling. The third panel shows the mean canonical cross correlation across all 12 nodes for the individual currents and the combined current. DP2 shows no increase in correlation throughout whilst the correlation begins to increase within minutes of the sharp decrease for DP1 and the combined current, both of which have a very similar response. The fourth panel shows the normalized number of connections the network would have for different uniform cross correlation thresholds. The network responds as soon as the canonical cross correlation reaches the network threshold.

Figure 34. A schematic of how a connection/edge between two magnetometers is identified. 1) Take two magnetometer vector magnetic field perturbation time series'. 2) Take a running 128 minute window of each time series, linearly detrended and canonically cross-correlated across positive and negative lags to calculate the linear combination of the two which are maximally correlated (U_1, V_1). 3) If the maximum canonical correlation component (correlation of U_1 and V_1) exceeds the station and event specific threshold¹¹, the magnetometers are connected in the network. The cross correlation lag τ_c of that peak value of cross correlation is identified with this network connection. 4) Given the geographical coordinates of the magnetometers we know the direction of propagation/expansion, in time, of the correlated signal between the two magnetometers¹².

Reviewer: Comment R2 C8: My concern is how the author's method distinguishes between two distinct modes of high latitude response DP-1 and DP-2. The author's say the following: "A first step in our analysis it to remove the slow trend in the data on the timescale of the window which removes any constant background- as is essential for any Fourier based analysis" This basically means they have "high pass filtered" the observations to select rapid variations of the substorm expansion. I don't remember this statement in the original draft and I can't find it in the current draft, but I may have missed it. It needs to be emphasized as it removes many of my objections.

Response: We have added text and updated our toy model (figure 28) to address this comment, the revised text reads:

Lines 117-120

"For each windowed time series we linearly detrend to remove any slow trends such as seasonal trends or modes associated with enhanced DP-2 current. Figure 28, SI, models the typical DP-1 and DP-2 responses and shows our detrending distinguishes between the two currents."

Reviewer: The authors continue with "Furthermore we do not claim to 'prove anything about how the

current wedge is formed” I though the objective was to determine whether individual wedgelets played a role in its formation. They say they have “Our analysis is carefully constructed to avoid any such categorization”. However, DP-1 is simply another name for substorm current wedge which is the response mode they have chosen to study.

Response: We understand the Reviewer’s point. However, we analyze ground based magnetometers and while the properties we determine provide constrains on the possible M-I current systems it is not the same as stating that these measurement and the analysis will explain how the SCW is formed. That is after all a magnetospheric process.

Reviewer: Comment R2 C9: I find later in the response three lists of onset times for the events used in this study. Their response did not include the times selected as the end of the expansion. Perhaps this list is in the Supporting Information that is not available at the website.

This comment identified a major concern, in particular why the authors did not specify which 41 events in the list of 76 events they used in their analysis? They have not provided this essential information. Furthermore, at the end of the following paragraph in my first review I said “The authors do not describe their timing procedure or list the times determined in the scaling of their time series.” Their response begins with the following:

Lines 71-74 “We study a subset of events drawn from the list of isolated substorms between 1997-2001 established by Gjerloev et al. [2007] and also described in Gjerloev and Hoffman [2014]. They determined the substorm timing (onset and peak expansion) solely from global auroral images rather than magnetometer data or magnetic indices.

I agree that the two papers mentioned do a good job of describing what was done to select events. Gjerloev et al. [2007] mention 116 substorms used in the analysis of the early paper. The Gjerloev and Hoffman [2014] paper also mentions a list of 116 substorms. However, neither of these papers provides a list of expansion or recovery onsets. Presumably the 76 events in the supporting information submitted with the first draft of this paper corresponds to some of the onsets in the original Gjerloev work.

Response: The full list with associated timings is provided in the current version of the manuscript.

Reviewer: The authors have not provided a list of times of the ends of the auroral expansion - They need to explain what additional constraints were applied to the original 116 events to reduce it to 76, and then why only 41 of these were used in this paper. As I point out in my first review, at least half of the 76 events are not truly isolated or single onset events as they were supposed to be.

Response: All substorms used in this paper are a subset of those 116 events used by Gjerloev and Hoffman. The 75 used within the SI are a subset of the 116 which have good coverage of the nightside (criteria described in methods). The 41 used in the main text are a subset of the 75 which have good coverage and are suitable quiet before onset (isolated). The SI further has a plot (21) containing 11 substorms ‘that are “almost” quiet before onset (SML>25% of that at the time of peak expansion during the 127 minute window before onset but <50%)’, as well as a plot (22) containing 23 substorms which are not quiet before onset. We have endeavoured to make the distinctions obvious in the associated timings list and have rewritten section 2.1:

Lines 82-90

“To maintain good magnetometer coverage of the nightside during the substorm we require two or more magnetometers in each three hour window of the nightside (e.g. ≥ 2 magnetometers in each segment of

18–21, 21–24, 00–03 and 03–06 hours of local time). A subset of 75 substorms fulfil this criteria. We also require that the nightside is quiet for at least one canonical cross-correlation (CCC) window (127 minutes) before the substorm onset so that the network calculated at the time of substorm onset is not contaminated with previous activity, rejecting events where the SML index exceeded ~25% of its maximum value (at the peak of the substorm) in the 127 minutes before the start of the substorm. 41 substorms fulfil all specified criteria and are analysed in this paper. Substorms with more activity before onset, from the list of 75 events are analysed in the SI (20-22) for comparison.”

Reviewer: In their extensive response to my concern they mention in item C) Substorm normalization: "Figure r4 is based on an amplitude and onset timing analysis. It unfortunately does not include an important normalization of the duration of the expansion phase."

My emphasis in bold is because the information necessary to do this normalization has not been provided by the authors in either draft, nor in the earlier papers. Without this information it is impossible for a referee to discuss the issue of substorm timing, or determine whether they have properly scaled the individual events before performing their analysis.

Response: The full list with associated timings is provided in the current version of the manuscript. Section 2.2 specifies the methodology for timing normalization.

We refer to the string of papers by Gjerloev and Hoffman. E.g. Gjerloev, J. W., R. A. Hoffman, J. B. Sigwarth, and L. A. Frank (2007), Statistical description of the bulge-type auroral substorm in the far ultraviolet, *J. Geophys. Res.*, 112, A07213, doi:10.1029/2006JA012189.

Reviewer: The authors have responded to this comment with a lengthy discussion of substorm timing. They argue that analysis of auroral images is the only good way to time substorm expansions.

Response: Please see the discussion of this point above. We do not recall stating that this is the “only good way” but rather that it is widely accepted as the most robust and objective method.

Reviewer: They provide a satellite image to counter the argument I made with my Figure r1. My reaction to this specific image is “good luck”. I am very dubious of timing derived from this image with such bad viewing geometry and such grainy display. [However, the projections of these images on polar plots are much better and allow one to time gross auroral features on a one minute scale.] Perhaps other substorms have better image sequences, but I don’t know. My question is why reject the magnetic timing of these events when you are studying the development of a current system?

Response: We could indeed have performed the study using SML for onset timing but using the imaging we ensure that all events are indeed bulge-type auroral substorms and minimize uncertainty in timing. As Gjerloev and Hoffman concluded “the AL (based on more than 100 ground stations included in the SuperMAG initiative) on average does minimize at the image maximum (see paragraph 2 for our definition), although individual events can deviate”. We would further point out that the onset identification was not done using the SuperMAG site which did not exist at the time of the first Gjerloev and Hoffman papers. They used the calibrated images with the viewing geometry provided by the S/C.

Reviewer: The authors then point out that the standard 12-station AL index has too few stations to do accurate timing, but in my review I used the SML index with over 100 stations. Finally the authors reject the use of the MPB index because:

“Using the mid-latitude bay index has very limited usage and thus it has not undergone any critical scrutiny (as far as we know).”. The MPB index is simply an implementation of a long-standing procedure

for determining the midlatitude positive bay onset that has been used since the beginning of substorm research. In fact it was first used in 1974 in a program called the “midlatitude bay finder”. See the Appendix to the following paper.

Caan, M.N., R.L. McPherron, and C.T. Russell (1978), The statistical magnetic signature of magnetospheric substorms, *Planetary and Space Science*, 26(3), 269-79.

I also note that Table 1 in the following paper compares the MPB onset times to both IMAGE, and Polar satellite image onsets. The peak of the time delay between them and MPB for IMAGE is – 1 min, and for Polar it is -3 min. The SML onset (71,056 events) is peaked -4 min ahead of MPB onsets. The conclusion is that on average the SML and MPB onsets are close to satellite image onsets.

McPherron, R.L., and X. Chu (2018), The Midlatitude Positive Bay Index and the Statistics of Substorm Occurrence, *Journal of Geophysical Research: Space Physics*, 123(4), 2831-2850.

doi:10.1002/2017JA024766

The authors suggest that the MPB onset is delayed relative to other onsets because of time to pen the wedge enough to create a significant midlatitude perturbation. I agree this is possible, but why doesn't this apply to the global network? At first a narrow wedge will only cause nearby stations to be correlated. As the wedge grows in size more stations enter the network of correlated stations.

Response: We certainly did not intend to reject the MPB index. Our comment was merely that the MPB has a built in delay compared to the actual onset as we stated in the previous report. This does not imply that the index is not suitable for analysis or event selection. Neither did we argue that AL can't be used but simply that the fewer station used to identify the onset the longer the average delay will be. It's simple probability.

Reviewer: Comment R2 C12:

My concern was about the separation of DP-2 (global convection) and DP-1 (current wedge). I must have missed the discussion of detrending the data so that time variations longer than window length were removed from the data. I still can't find this statement in the current draft.

Response: As detailed above this has now been added to the method section and is shown in the modeling of SI figure 28.

Reviewer: The authors mention a study by Gjerloev et al. [2010] that concludes the DP-2 current system during quiet times does not exist on the night side. They argue the whole idea of a global DP-2 current system has been changed by this work. In the paper they mention 73 substorms selected for a step function change from constant north Bz to constant south Bz. They were timed by Polar auroral images. This set of substorms is very unique because the selection criteria virtually guarantee that there is no nightside electrical conductivity until the expansion onset. Thus the application of a convection electric field drives no current. These probably differ from more typical substorms in which electrons injected by previous substorms circle the Earth and precipitate causing conductivity in the oval.

Response: This is a very interesting comment. The DP2 current system is indeed a 'current system' not convection. As such that paper simply asked how the DP2 current system responds to southward turnings of the IMF. Not the convection but the currents. The conclusion was simple and exactly what the referee refers to – the DP2 requires more than a southward turning, it also require precipitation that is not necessarily a direct consequence of a southward turning. This seems exceedingly simple but it is nevertheless an assumption of the DP2 current system.

Reviewer: Comment R2 C15:

This modified substorm onset list is part of what I needed to see. Note there is no new Supplementary Information file at the Journal Website. Where is the list giving the end of expansion times? The authors say “the timings are determined directly from Polar VIS images (freely available on SuperMAG).” The SuperMAG web site does not list Polar VIS Images as an available product. I don’t want the original data, I want to see the list someone created from the images. I don’t want to repeat this laborious project.

Data Products (from website)

Line Plots and Data Download

Polar Plots

Magnetic Indices

Substorm Event List

Movies

Inventory

The polar plots have overlays of VIS images. It is not obvious that one can get just the images. (Finally managed this by turning off many options.) The substorm onset list at the website only gives the times of negative bay onset determined from SML index, and does not include the end of expansion. These are not the VIS times needed for proper evaluation.

Response: The list of all times is included in the current version of the manuscript as requested by the Reviewer. The substorm list on SuperMAG is merely a quick attempt to identify and time substorm onsets. It is not suitable for this study.

Reviewer: Comment R2 C16: Does your glossary contain explanations of the correlation procedure? The SI file is not available at the journal website. I am curious how the CCA combines the components of the field when stations are located in distinctly different parts of the perturbation pattern.

Response: Yes the glossary contains a canonical correlation description and we have included an example of the procedure for a real pair of magnetometers in figure 34.

Reviewer: Comment R2 C17: The CCA must grow slowly as the leading edge of the moving window passes over the sudden onset of the expansion. It must take as long as the duration of the expansion to reach a maximum value. Is this true?

Response: For our model example (figure 28) maximum correlation is reached within several minutes of the ‘onset’, but for real examples we rarely have near zero activity before onset, even for an isolated substorm. For real examples it takes longer but is still usually within the expansion phase and often within twenty normalized minutes. The panel 6 of the individual substorm plots (fig 1, SI 2, 5, 8, 11) show what proportion of the magnetometer pairs are correlated above their threshold (connected to the network). In figure 1 maximum correlation within the auroral bulge is reached approximately half way through the expansion phase (18 normalized minutes).

Reviewer: Comment R2 C20: Where is the list of the end of expansion times?

Response: The list of all times is included in the current version of the manuscript as per request.

Reviewer: Comment R2 C21: This question was concerned with the explanation of the 10 minute delay, as well as the timing issue. You pointed out that that the MPB index may be delayed by opening of the current wedge sufficiently that the ground perturbation exceeds the noise. Why is this not true for the

CCA in the auroral zone? As the moving window first encounters the station perturbations the signal occupies only one sample out of 128. You have thresholds for acceptable signal strength and do not establish a connection until this happens.

Response: As part of our analysis we obtain random phase surrogates from the data and feed these through our analysis to directly determine the 'noise floor' in the cross correlation and in our derived network parameters. Thus we are able to identify from the data when a spatially coherent pattern is seen across multiple stations that is above the noise.

Reviewer: Comment R2 C27: The authors seem to use the word substorm to refer only to the current wedge. "...large scale current response to a substorm, i.e. DP-1. R2". The growth phase preceding the expansion phase is part of the substorm. You have said in several places in the response that you detrended the data so that variations on the time scale of DP-2 are removed from the data. DP-2 is the manifestation of the changes in magnetospheric configuration that makes the expansion phase possible so this is also part of the substorm. Please add this information to the manuscript.

Response: We agree with the referee and have added text:

Line 51-52

"We perform this analysis across 41 isolated substorm events at one minute temporal resolution, focusing on the spatial and temporal characteristics of the SCW and we find a robust and consistent configuration of ionospheric substorm currents evolving as the substorm progresses."

Reviewer #4:

Reviewer: This revised version of the network-based study of magnetospheric substorms is very clear and indeed exciting. Therefore, I recommend acceptance of this paper now.

Response: we thank the Reviewer for their careful and constructive reading of our manuscript.

Reviewer #5:

Reviewer: This paper's results are **definitely of publishable quality**; I have only a few comments, mostly on methodologies rather than the conclusions on geomagnetic substorms. As at places, the paper appears to be over-hyping the tools without stating the pitfalls.

Response: We have added text in the conclusions to highlight limitations of our approach:

'Our network analysis is built on linear cross correlation and as such does not identify non-linear relationships. Communities are a relatively non-formal way (despite unbiased estimates) of quantifying interactions. This could be addressed in future work using more advanced methods, provided there is sufficient observations to make this viable; an essential limitation here is that the substorm timescale is a few hours and the data are at 1 minute time resolution.'

Reviewer: Authors need to motivate why community detection is a better algorithm than, say, an EOF analysis. Please see the paper: <https://arxiv.org/abs/1305.6634>. Please discuss the drawbacks of community detection, artifacts, and caveats while analyzing spatiotemporal data.

Response: We note that the Donges et al preprint referred to by the Reviewer is a study applied to climate data which are spatial fields already available mapped onto a regular spatial grid. Here we are dealing with 100+ magnetometer stations that are non-uniformly distributed spatially. We can directly obtain the network and its community structure without first imposing a coordinate system or a k-space partitioning. We do not need to apply any interpolation or krigging to map the data onto a regular grid and this would be undesirable as it would introduce ‘artificial’ spatial correlation. One can in principle apply a decomposition such as that referred to by the referee. However this would essentially be a fit of EOF modes across non-uniformly distributed spatial samples which can result in artefacts.

Community detection can be biased but we have tested whether these structures are more naturally likely than expected from a random configuration by (i) temporally, using the data to generate random phase surrogates and then feeding these through the analysis to generate a noise floor for the network modularity and (ii) spatially, by comparing the modularity with that found from a random network.

We have added text to discuss some relevant drawbacks to community detection:

Lines 147-149

“The ability to validate community structure is limited Porter et. al, 2009 but we aim to perform a statistical study without focusing on individual details within individual substorms, hence our conclusions should be robust against detection algorithm.”

Lines 192-196

“This is plotted in figure 25 which shows there is a clear upper bound to the value of modularity per degree distribution observed in random networks. However the modularity patterns observed throughout this text exceed those observed in a random network and therefore are not simply a result of the increasing numbers of connections.”

Reviewer: These community structures are depended on the thresholds, i.e., connection threshold for the station pairs. Have authors made some tests on networks obtained without making subjective choices on thresholds, for example, using statistical significance for deciding on a link's existence?

Response: The connection threshold is determined per station pair and per event to meet a criterion that all nodes are on average equally statistically likely to be connected to the network. This overall criterion (the average degree threshold) has then been varied to check the robustness of our modularity results from the manuscript. This procedure is essential as the magnetometer stations have different response functions and the response also varies with season and with local geography- it depends on the local ground conductivity. We use each event to determine directly from the data what the connection thresholds are for each station pair.

Reviewer: How would these results change if one uses a nonlinear metric for correlation, for example, mutual information? Similarly, several metrics are available that explicitly calculate information flow between grids, for example, transfer entropy. Generally, when directed networks are inferred from data, one uses a more sophisticated metric for information flow and not just time delays. I am raising this issue because these results should not be just artifacts of metrics and thresholds employed.

Response: Bearing in mind that a substorm is of a few hours in duration, and that we have 1 min cadence observations, we have found that we simply do not have enough data for MI or transfer entropy. We have typically 100 stations that are irregularly distributed in space. We do not have a regular grid. It is not necessarily the case that these more sophisticated methods will outperform linear cross-correlation, for example Song et al. 2012 tested various correlation methods against MI as methods of association between network nodes when dealing with gene expression data. They found that correlation outperformed MI in measuring association between pairwise genes.

In our SI we detail a range of studies that we have performed to verify that our results are robust when we vary the threshold, and that they can be clearly discriminated from the analogous random network or coloured noise signals. Figures 14-24 present just some of the variations of the input that we have tried. In particular I would refer to figures 23-24 which vary the threshold frequency. We reemphasize these figures in the main text with:

Lines 295-304

“We find the same overall behavior looking over a wider set of events (see figures 19-20 SI), however more active conditions before the substorm make the pattern less clear. If we only consider substorms with large values of SML in the 127 minutes preceding onset, we can no longer isolate such a clear transition in the modularity distribution (see SI Figure 22). Figures 15-19 show the same figure but using five alternative algorithms for community detection. The same pattern is repeated in all cases. Further, the coherent behavior is repeated across a range of cross-correlation thresholds (see SI figures 23-24). If the threshold is set too high (<1% of magnetometers ‘connected’ within a month) there would not be enough network connections to see any pattern (hence figure 23 contains more outliers). Figure 25 in the SI shows that the modularity, Q, derived from the substorm events has more structure (higher Q) than random networks and for a given degree distribution, the data explores a broad range of modularity values, Q, and vice versa. We have included these figures in the SI to show our results are not simply an artefact of choice of algorithm or threshold.”

Reviewer: The glossary of network terminology and diagrams is useful, but this paper needs a simple diagram/illustrating explaining how exactly these networks were constructed.

Response: We have expanded figure 34 to explain how the network is constructed. Further diagrams are available in Dods et al. 2015. This is clarified with new text:

Lines 107-108

The details of the underpinning methodology for forming the raw network is detailed in Dods et al. 2015, Orr et. al. 2019, and we summarize it here.

Lines 134-135

“A schematic showing how the network is constructed by means of a pair of magnetometers is included in the SI, figure 34.”

Figure 34. A schematic of how a connection/edge between two magnetometers is identified. 1) Take two magnetometer vector magnetic field perturbation time series'. 2) Take a running 128 minute window of each time series, linearly detrended and canonically cross-correlated across positive and negative lags to calculate the linear combination of the two which are maximally correlated (U_1, V_1). 3) If the maximum canonical correlation component (correlation of U_1 and V_1 exceeds the station and event specific threshold¹¹, the magnetometers are connected in the network. The cross correlation lag τ_c of that peak value of cross correlation is identified with this network connection. 4) Given the geographical coordinates of the magnetometers we know the direction of propagation/ expansion, in time, of the correlated signal between the two magnetometers¹².

Reviewer: The comparison with Erdos-Renyi is technically flawed. The comparison should be with configuration models, random networks with the same degree distribution as the empirical networks.

We thank the reviewer for pointing this out. We have now compared the network derived from the substorm events to a random network with the same distribution. We reach the same conclusions. We have replaced the plot in the SI as attached:

40 **Figure 25**

41 Figure 25 shows how the modularity scales compared to that of a random network with the same degree distribution. The
42 network at each time is randomly rewired while preserving the original graph's degree distribution, using the igraph package⁶.
43 To make a random network the algorithm arbitrarily chooses two edges, (a,b) and (c,d) and substitutes them for (a,d) and (c,b).
44 This is iterated $100 \times N$ times, where N is the number of nodes, to make a random network with the same degree distribution as
45 as the original network. We calculate 100 random networks per time point for all 41 substorms, and the modularity is calculated
according to the edge betweenness algorithm as per the method in the main text.

Figure 25. Modularity, Q , plotted versus the normalized time, t' , for each of the 41 substorms in the main paper. The edge betweenness algorithm has been used for community detection as in main text. The modularity from the networks derived from the observed substorm events is plotted with colour representing the normalized number of connections, or edges, $\alpha(t')$. The red circles plot the modularity obtained from sets of randomly generated networks which have the same degree distribution and number of edges as the observed network at each time. As expected, the random networks explore a range of modularity values (a measure of how separated the communities are) but this range has a clear upper bound. The networks derived from the substorm events explores a broad range of modularity values and this systematically exceeds that of the corresponding random networks that share the same degree distribution and number of edges. Our main result, that the modularity transitions from a high to a low value during substorms, hence does not simply arise as a result of increasing number of network edges with time.

And edited the text in the main text to:

Lines 192-196

“We also compared the modularity with that of random networks generated with the same total number of connections and degree distributions as the networks derived from the substorms which varies with time. This is plotted in figure 25 which shows there is a clear upper bound to the value of modularity per degree distribution observed in random networks. However the modularity patterns observed throughout this text exceed those observed in a random network and therefore are not simply a result of the increasing numbers of connections.

Lines 301-304

“Figure 25 in the SI shows that the modularity, Q , derived from the substorm events has more structure (higher Q) than random networks and for a given degree distribution, the data explores a broad range of modularity values, Q , and vice versa. We have included these figures in the SI to show our results are not simply an artefact of choice of algorithm or threshold.

Reviewer: A critical comment about the visualization in this paper: apart from figure captions being incomplete and doing a poor job explaining the figures and their significance, the visualizations are incredibly complex (lines of different types, colors, markers, left-right axis, multiple panels, and more). Please consider simplifying some of the figures and rewriting captions. A test that authors can use is to write captions that are standalone; that is, a reader can understand most of the figure without going into text back and forth, looking for details of each variable and acronym.

Response: We have clarified the figure descriptions:

Figure 1. Community structure of an example substorm. The community structure of a substorm on the 16/03/1997. The abscissa of all panels is normalized time ($t' = 0$ is onset (green line) and $t' = 30$ (purple line) is the time of maximum auroral bulge expansion). Panels a-b plots individual communities as circles where the size of the circle reflects the number of connections within the community. The ordinate plots the mean magnetic local time/ latitude (MLT/MLAT) of the community, $\bar{\theta}_x(t')$ and $\bar{\phi}_x(t')$, and the color indicates the proportion of connections with each time lag, $|\tau_c|$. The dashed lines overlotted are the edges of the auroral bulge (MLT) and the onset location (MLAT), found from auroral images. Panels c-d show the spatial extent of each community, where the dots are the magnetometer locations and the shading is the extent. Color represents the mean MLT of the stations contained within each community, $\bar{\theta}_x(t')$. Panel e plots the modularity, Q , (blue line) and the random phase surrogate (black line). Panel f plots the normalized number of connections, $\alpha(t')$, both within the nightside and within the SCW, as well as their surrogates (both near zero throughout). The right ordinate plots (negative) SML. 6/13

Figure 2. Community structure snapshots of an example substorm. The magnetic field perturbation vectors (North and east components, $B_{N,E}$) are colored by the the magnetic local time (MLT) of the centroid ($\bar{\theta}(t')$) of each community for a substorm on 16/03/1997. The colors match those of panels c-d in figure 1. Each subplot represents a snapshot of the nightside community structure in five normalized minutes from before onset to the time of maximum expansion, corresponding to the times in figure 1. The circles represent ground magnetometers in MLT with the line representing the $B_{N,E}$ vector. Black magnetometers are not connected to the network. The dashed lines the locations of the auroral bulge found from auroral images.

Figure 3. Community structure of multiple substorm events. The normalized modularity, Q_N , of the set of 41 substorms that have 2 or more magnetometers in four even local time sectors of the nightside and are quiet before onset. Panels a-b have normalized time as the abscissa. The panel a ordinate bins Q_N at each normalized time and the color indicates the probability (count of substorms with Q_N /total number of substorms). Panel b plots Q_N of all substorms as a function of normalized time with the median overlotted in black and the 25% and 75% quantiles in gray. Panel c plots the normalized histograms of Q_N of the events aggregated over 10 minute intervals as time progresses. The median is overlotted.

Reviewer: Are the authors claiming that the area that falls, say, between a community formed by three nodes (a triangle) is somehow an area of spatially coherent activity? I am not able to understand these shapes in Fig. 2 (and other similar figures). From community structure how do you estimate the shape of region with coherent activity?

Response: Our example substorms have been presented as an example of how community structure changes from small localized areas of correlation to a spatially-extended correlated system. We are not

attempting to deduce individual details of these structures using this method as we do not have the spatial resolution.

We have emphasized this point with text:

Line 200

"We present this example event as an illustration of the methodology that we will use to compare multiple events."

Reviewer: Minor points:

In the paper, the figures from supplementary material are referred as S1-S35; however, there is no "S" in figure numbering in the supplementary material.

Response: We will change the figure numbering to reflect nature communications template.

REVIEWERS' COMMENTS

Reviewer #1 (Remarks to the Author):

The authors have performed an extraordinary amount of work in response to the referee comments. I have read through the lengthy report and the equally lengthy reply, and have no further suggestions. The authors have responded well to the comments, and I believe the paper is ready for publication.

I have also reviewed the Reviewer #2's comments and the authors response, and find that all major questions raised by referee #2 were responded to adequately by the authors, with changes to the text as needed, and even the minor recommendations were given considerable attention. Again, I commend the authors on such a thorough response, and applaud the authors for bringing (and demonstrating the applicability of) a valuable new technique to our discipline.

Reviewer #1 (Remarks to the Author):

The authors have performed an extraordinary amount of work in response to the referee comments. I have read through the lengthy report and the equally lengthy reply, and have no further suggestions. The authors have responded well to the comments, and I believe the paper is ready for publication.

I have also reviewed the Reviewer #2's comments and the authors response, and find that all major questions raised by referee #2 were responded to adequately by the authors, with changes to the text as needed, and even the minor recommendations were given considerable attention. Again, I commend the authors on such a thorough response, and applaud the authors for bringing (and demonstrating the applicability of) a valuable new technique to our discipline.

Reviewer #5 (Remarks to the Author):

Authors response: We thank all reviewers for their comments and taking the time to read and respond to our reports.